# On the Recall Scaling Laws in Mamba: A Theoretical and Mechanistic Study via Hashing

## Abstract

Associative Recall (AR) is the cognitive ability to learn and retrieve links between items in memory. In NLP, AR is used as a benchmark for evaluating the in-context memory capacity of architectures such as Mamba, and has been found to strongly correlate with language modeling performance. This paper explores AR from the perspective of mechanistic interpretability, aiming to reverse-engineer the exact internal algorithm used by Mamba to perform recall. Our key insight is that Mamba performs recall by implicitly learning linear hash functions, and we identify the low-level circuit that enables this behavior. Building on these findings and inspired by theoretical tools in similarity-preserving hashing, such as the Johnson–Lindenstrauss (JL) lemma, we develop a theoretical framework for analyzing AR, which we term recall scaling laws. For example, given the model embedding dimension and state dimension, our theory predicts an upper bound on the vocabulary size and the number of facts that it can perfectly recall. Empirical results show that this bound is tight and predictive, offering insights into how AR capacity scales with vocabulary, state, embedding size, and architecture.

## 1 Introduction

Although Transformers have achieved remarkable success in sequence modeling, they suffer from a key inefficiency when handling long sequences: during decoding, memory complexity grows linearly with sequence length due to key-value caching. Recent architectures such as Mamba (Gu & Dao, 2023), RWKV (Peng et al., 2023), and other linear RNNs (De et al., 2024) address this limitation by utilizing a fixed-size recurrent state, enabling constant memory complexity.

However, this fixed-size state is a double-edged sword: from an information perspective, compressing the entire context into a single vector imposes inherent limitations and may lead to information loss (Wen et al., 2024; Ben-Kish et al., 2025). Hence, improving memory utilization in recurrent architectures has become an active area of research (Parnichkun et al., 2025; Arora et al., 2024).

To better understand the memory bottlenecks of sequence models, researchers have proposed synthetic benchmarks such as Associative Recall (AR) (Ba et al., 2016), Multi-Query Associative Recall (MQAR) (Arora et al., 2023) and others (Arora et al., 2025; You et al., 2024), which are highly associated with language modeling capabilities (Arora et al., 2023). These tasks are designed to diagnose a model's recall abilities in controlled environments and have been widely studied across various architectures (Wang et al., 2025; Bick et al., 2025), including Transformers (Olsson et al., 2022), linear RNNs (Fu et al., 2022; Trockman et al., 2024), and others (Poli et al., 2023).

In this paper, we take a step toward a deeper understanding of Mamba's recall abilities by adopting a mechanistic interpretability perspective (Elhage et al., 2021; Olah et al., 2020), aiming to reverse-engineer the internal algorithm employed by linear RNNs to solve AR tasks. Through empirical analysis, we identify the circuit-level mechanism that enables Mamba to perform AR and interpret it as implicitly computing a similarity-preserving linear hash function (Andoni & Indyk, 2008; Datar et al., 2004; Salakhutdinov & Hinton, 2009). Building on this connection, and drawing on theoretical tools from linear hashing such as the Johnson–Lindenstrauss (JL) lemma (Johnson et al., 1984), we develop a theoretical framework that we term "recall scaling laws". This framework theoretically analyzes how the state size and embedding dimension must scale with the vocabulary size to achieve perfect recall, and empirical analysis shows that these scaling laws are relatively tight and reflect the model's behavior in practice.

**Our main contributions** encompass the following three aspects: (i) We study the associative recall capabilities of Mamba through the lens of mechanistic interpretability. By progressively removing components from the model, we isolate the circuit-level mechanism responsible for performing recall. (ii) Motivated by this finding, we interpret the learned weights as implicitly computing similarity-preserving linear hash functions, and the entire circuit as exhibiting hash-like behavior. This interpretation allows us to leverage the JL lemma to better understand the theoretical capacity of Mamba's recall mechanism. Finally, (iii) we propose a theoretical framework, which we term "recall scaling laws", that provides an upper bound on the state size, context length and embedding size required to perform near-perfect recall for a given vocabulary size. We empirically validate that these scaling laws hold in practice and extend them to multi-head and multi-layer regimes.

## 2 BACKGROUND & RELATED WORK

**Associative Recall.** The AR task was first introduced by Ba et al. (2016), and has proven to be an effective tool for evaluating recall and retrieval capabilities. More recently, AR has gained popularity due to its strong correlation with language modeling performance (Arora et al., 2023). A growing body of work has leveraged AR to guide architectural choices (Poli et al., 2023; Fu et al., 2022; Arora et al., 2024; Lutati et al., 2023), improve initialization strategies (Trockman et al., 2024), analyze optimization dynamics (Okpekpe & Orvieto, 2025), and provide both theoretical and empirical insights into the limitations of recurrent LLMs (Wen et al., 2024; Ben-Kish et al., 2025).

The work most closely related to ours is by Bick et al. (2025), who use mechanistic interpretability to analyze in-context retrieval through a gather-and-aggregate mechanism. However, it does not study AR or provide theoretical analysis. Jelassi et al. (2024) investigate the copying capabilities of Transformers and SSMs, offering theoretical insights into these mechanisms. While relevant, their work does not directly examine AR and does not employ mechanistic interpretability to understand what models learn in practice. Finally, Huang et al. (2025) explores the recall capabilities of Mamba models, leveraging the JL lemma to derive theoretical bounds. However, their work does not adopt a mechanistic interpretability perspective and does not investigate the mechanisms learned in practice. In contrast, our theoretical and empirical analysis is more comprehensive, culminating in recall scaling laws that demonstrate how both the state size and embedding size must scale with the vocabulary size to enable effective recall. Furthermore, our study provides an understanding of recall circuits in *multi-layer* and *multi-head* Mamba architectures, in addition to the plain single-layer mechanism. (Also see detailed comparison in App. I.)

**Multi-Query Associative Recall (MQAR).** MQAR by Arora et al. (2023) is an extension of the classical AR. It is constructed by concatenating a factual context section with a multiple queries section to form a single prompt. The vocabulary $\mathcal{V}$ is partitioned into two disjoint subsets: the key vocabulary $\mathcal{V}_k$ and the value vocabulary $\mathcal{V}_v$ (see Table 1, Appendix). The vocabulary size is $V = |\mathcal{V}|$, which is assumed to be even so that half of the tokens correspond to keys, and the other half correspond to values ($V_k = |\mathcal{V}_k| = V_v = |\mathcal{V}_v| = \frac{V}{2}$). Each prompt has a total sequence length $L$. The context is composed of $N_f$ non-repeating key–value pairs $(k_i, v_i)$, resulting in $2N_f$ tokens. The remaining $L - 2N_f$ tokens form the query section, which includes $N_f$ query tokens $q_i$, each duplicating a key token that appears in the context. The rest of the query section is filled with padding. Padding tokens may be set to zero or chosen at random, and following the original implementation, we assume they are randomly sampled from the full vocabulary $\mathcal{V}$.

For each query $q_i$, the model's task is to retrieve and output the value $v^*$ corresponding to the fact $(k^*, v^*)$ with a matching key $k^* = q_i$. An example input $x$ and ground-truth output $y$ sequences can be found below, with highlighted key, value, query and padding tokens:

$$x \quad \text{A} \; 6 \; \text{B} \; 3 \; \text{C} \; 7 \; \text{B} \; 2 \; 5 \; 0 \; 9 \; \text{C} \; 4 \; \text{A} \; 8 \; 1$$

$$y \quad \star \; \star \; \star \; \star \; \star \; \star \; 3 \; \star \; \star \; \star \; \star \; 7 \; \star \; 6 \; \star \; \star$$

where $\star$ denotes an ignored output token (whose classification does not contribute to the loss). Success on MQAR demonstrates a model's capacity for dynamic in-context learning and retrieval.

**Mamba.** The recently presented *Selective SSM* (Gu & Dao, 2023), known as S6, outperforms previous SSMs and various other architectures in NLP (Anthony et al., 2024; Wang et al., 2024b), vision (Liu et al., 2024; Zhu et al., 2024), graph classification (Wang et al., 2024a), and more. S6 incorporates a dynamic input-dependent form of the discrete matrices $\bar{A}, \bar{B}$, and $C$, such that for

every time-step the SSM employs a different recurrent rule. This technique differs from the previous state-space layers, which use the same set of matrices for each timestep.

A *Mamba* model of embedding size $D$ and state size $N$ is composed of $\Lambda$ stacked Mamba blocks. The *Mamba block* combines the S6 state space model (SSM), a Conv1D layer, and other elementwise operators; it borrows elements from Gated MLP and similar architectures. We denote the expanded dimension by $D_{\text{in}} = \text{expand} \times D$ (where $\text{expand} \in \mathbb{N}$), and the convolution filter size by $D_{\text{conv}}$.

Given an embedded input sequence $x^e = (x_0^e, \ldots, x_{L-1}^e) \in \mathbb{R}^{D \times L}$, the Mamba block operation is as follows (see Tab. 2, 3 for notation and dimension details):

$$\hat{x} = \text{SiLU}(\text{Conv1D}(\text{Linear}(x^e))), \quad \hat{z} = \text{SiLU}(\text{Linear}(x^e)), \quad \hat{y} = \text{SelectiveSSM}(\hat{x}) \odot \hat{z}, \quad y^e = \text{Linear}(\hat{y})$$

where $\odot$ denotes element-wise multiplication, and we omit normalization layers for simplification. For each timestep $t$, the SSM vectors $B_t, C_t \in \mathbb{R}^N$ and discrete matrices $\bar{A}_t, \bar{B}_t \in \mathbb{R}^{D_{\text{in}} \times N}$ are computed as:

$$B_t = S_B \hat{x}_t, \quad C_t = S_C \hat{x}_t, \quad \Delta_t = \text{SoftPlus}(S_\Delta \hat{x}_t), \quad \bar{A}_t = \exp(\Delta_t A), \quad \bar{B}_t = \Delta_t B_t \tag{1}$$

where $S_B, S_C, S_\Delta$ are linear projection matrices, SoftPlus is a smooth approximation of ReLU, and biases are omitted for simplicity. Then, each SSM channel $d = 1, \ldots, D_{\text{in}}$ transforms an input $\hat{x}_t^d \in \mathbb{R}$ into an output $\hat{y}_t^d \in \mathbb{R}$, through updating a recurrent hidden state vector $\mathbf{h}_t^d \in \mathbb{R}^N$:

$$\mathbf{h}_t^d = \bar{\mathbf{A}}_t^d \odot \mathbf{h}_{t-1}^d + \hat{x}_t^d \bar{\mathbf{B}}_t^d, \quad \hat{y}_t^d = \mathbf{h}_t^d \cdot \mathbf{C}_t \tag{2}$$

where $\bar{\mathbf{A}}_t^d, \bar{\mathbf{B}}_t^d, \mathbf{C}_t \in \mathbb{R}^N$, $\odot$ denotes an element-wise product, and $\cdot$ is a dot product. In total, all channels form the SSM hidden state matrix, $h_t \in \mathbb{R}^{D_{\text{in}} \times N}$. (Also see App. A.3 for details.)

The usage of time-variant layers adds to the expressivity of the layer (Cohen-Karlik et al., 2025), allowing it to adapt to the input, and potentially captures more complex dependencies. While other input-dependent time-variant mechanisms have been proposed in previous works through gated RNNs, S6 also presents an efficient IO-aware implementation, which is parallelized on GPUs via work-efficient parallel scanners (Blelloch, 1990; Smith et al., 2022), making it well-suited for long-context processing (Ben-Kish et al., 2024; Ye et al., 2025).

**Mechanistic Interpretability.** Mechanistic interpretability seeks to reverse-engineer deep learning models so they can be understood, trusted, and controlled (Elhage et al., 2021; Olah et al., 2020). The approach works by linking observable behaviors to specific neural components, often by assigning semantic functions to individual neurons or weights and identifying compact "circuits" within the network. Representative examples include the induction-heads (Olsson et al., 2022) and arithmetic mechanisms (Kantamneni & Tegmark, 2025; Zhong et al., 2023; Chughtai et al., 2023), in-context learning via task-vectors (Hendel et al., 2023) and thinking progress via progress-vectors (Eisenstadt et al., 2025). Our work follows this line of research and aims to uncover the intrinsic mechanisms responsible for recall in Mamba models and linear RNNs.

# 3 REVERSE ENGINEERING MAMBA ON MQAR

We aim to reverse-engineer the underlying algorithm that enables Mamba to perform MQAR. To do so, in Sec. 3.1, we present a minimal yet efficient Mamba model that successfully solves MQAR. Next, in Sec. 3.2, through mechanistic interpretability analysis, we verify that our suggested mechanism is indeed the one learned in practice.

## 3.1 A MAMBA CIRCUIT FOR MQAR

We describe a simplified, purely-linear, single-layer Mamba model, and show theoretically and empirically that it is able to solve MQAR with arbitrarily high probability. A schematic overview of how the minimal model performs recall is presented in Fig. 1.

**Minimal model and simplified SSM.** In Alg. 1 we present a minimal single-layer Mamba model architecture, where gating, discretization, nonlinearities, biases, normalization layers, and residual connections are all removed. Dimensions are listed in Tab. 3, Appendix. The model is wrapped with embedding and unembedding layers of a dimension $D$, which expands to

---

**Algorithm 1:** Simplified Single-Layer Mamba

**Input:** $x \in \mathbb{R}^{V \times L}$ , **Output:** $y \in \mathbb{R}^{V \times L}$

Parameters: $E, \ P_{\text{in}}, \ P_{\text{out}}, \ W_{\text{conv}}, \ S_B, \ S_C$

$x_{\text{e}} \leftarrow E x$

**Mixer:**

$\quad x_{\text{in}} \leftarrow P_{\text{in}} x_{\text{e}}$
$\quad \hat{x} \leftarrow \text{Conv1D}(x_{\text{in}}, W_{\text{conv}})$
$\quad B, \ C \leftarrow S_B \hat{x}, \ S_C \hat{x}$
$\quad \hat{y} \leftarrow \text{SSM}(\hat{x}, B, C)$
$\quad y_{\text{out}} \leftarrow P_{\text{out}} \hat{y}$

$y \leftarrow E^\top y_{\text{out}}$
**return** $\underline{y}$

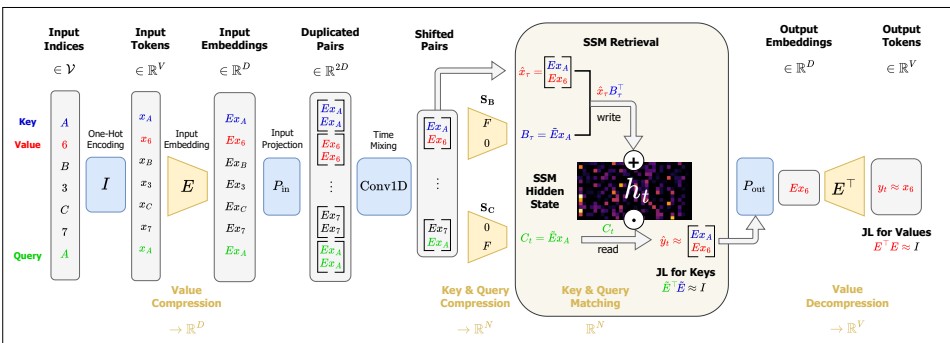

Figure 1: **Minimal recall circuit.** An overview of the Mamba recall circuit in Thm. 2. Left to right: (1) Tokens are embedded using $E \in \mathbb{R}^{D \times V}$. (2) $P_{\text{in}}$ and Conv1D apply duplication, copy and shift, forming pairs $\left(\begin{smallmatrix} Ex_{t-1} \\ Ex_t \end{smallmatrix}\right)$. (3) SSM projections $S_B, S_C$ extract keys and queries, then further compress them via $\tilde{E} = FE \in \mathbb{R}^{N \times D}$. (4) Embedding pairs are stored in the SSM using $h_t = \sum_\tau^t \hat{x}_\tau B_\tau^\top$. (5) JL matrix $\tilde{E}$ enables retrieval through key-query matching, $h_t C_t = \sum_\tau^t \hat{x}_\tau (B_\tau^\top C_t) \approx \hat{x}_\tau$. (6) $P_{\text{out}}$ selects the value half $Ex_\tau$. (7) JL matrix $E$ enables value reconstruction, $y_t = E^\top E x_\tau \approx x_\tau$.

size $D_{\text{in}} = \text{expand} \times D$ in the input projection. We further note that without discretization, the SSM recurrent update step described in Eq. 2 becomes rather simple. If we additionally set $A$ to identity, the SSM operation can now be written as (see App. A.3 for details):

$$h_t = h_{t-1} + \hat{x}_t B_t^\top, \quad \hat{y}_t = h_t C_t. \tag{3}$$

**Mamba recall circuits.** We first describe an ideal (dimensionally inefficient) non-compressive circuit that enables Mamba to solve the MQAR task. Next, we present a rather realistic, compressive circuit as a feasible approximation of the ideal one.

**Theorem 1** (**Perfect non-compressive recall circuit**). *Given a vocabulary size $V$, a single-layer simplified Mamba with dimensions $D = V$, $N = V$, expand = 2, $D_{\text{conv}} = 2$ can **perfectly solve** an MQAR task, i.e. with recall probability = 1.*

*Proof sketch.* (Full proof can be found in App. B.) We set the model weights as follows:

$$E = I_V, \quad P_{\text{in}} = \left(\begin{smallmatrix} I_V \\ I_V \end{smallmatrix}\right), \quad P_{\text{out}} = (0 \mid I_V),$$
$$W_{\text{conv}} = \left(\begin{smallmatrix} 1_V & \mid & 0_V \\ 0_V & \mid & 1_V \end{smallmatrix}\right), \quad S_B = (I_V \mid 0), \quad S_C = (0 \mid I_V), \tag{4}$$

where $\mid$ denotes concatenation. $P_{\text{in}}$ duplicates the input, $x_t^p = P_{\text{in}} E x_t = \left(\begin{smallmatrix} x_t \\ x_t \end{smallmatrix}\right)$. The Conv1D performs copy and shift; the SSM input is $\hat{x}_t = \text{Conv1D}\left((x_{t-1}^p, x_t^p), W_{\text{conv}}\right) = \left(\begin{smallmatrix} x_{t-1} \\ x_t \end{smallmatrix}\right)$. Inside the SSM, we have $B_t = S_B \hat{x}_t = x_{t-1}$ and $C_t = S_C \hat{x}_t = x_t$. After unrolling the expression in Eq. 3 and plugging in the above values, the SSM operation becomes $\hat{y}_t = h_t x_t = \left(\sum_{\tau=0}^t \hat{x}_\tau B_\tau^\top\right) x_t = \sum_{\tau=0}^t \left(\begin{smallmatrix} x_{\tau-1} \\ x_\tau \end{smallmatrix}\right) x_{\tau-1}^\top x_t$. Finally, given a query $x_t \equiv q_t \in \mathcal{V}_k$, the output is:

$$y_t = E^\top P_{\text{out}} \hat{y}_t = \sum_{\tau=0}^t x_\tau \langle x_{\tau-1}, q_t \rangle. \tag{5}$$

If the context contains a fact $(x_{\tau-1}, x_\tau) = (k^*, v^*)$ such that $q_t = k^*$, and if we assume $k^*$ is unique in the context, the output becomes exactly $y_t = v^*$, since all input vectors $\{x_t\}$ are one-hot, hence orthonormal. Thus, the model perfectly solves the associative recall task. $\qquad \square$

The circuit described in Theorem 1 is impractical, primarily due to its reliance on large state and model dimensions (increases linearly with $V$), which are unrealistic in practice. Thus, the following theorem presents a more efficient circuit, based on an approximate solution rather than a perfect one.

**Theorem 2** (**Efficient compressive recall circuit**). *Given a vocabulary size $V$, a single-layer simplified Mamba with dimensions $D$, $N = O(\log V)$, expand = 2, $D_{\text{conv}} = 2$ can solve an MQAR task **with high probability**. (See Thm. 3 for quantitive analysis.)*

*Proof sketch.* (Full proof can be found in App. B.) We apply the adapt the circuit from Theorem 1 to enable compression. Given embedding matrices $E \in \mathbb{R}^{D \times V}$ and $F \in \mathbb{R}^{N \times D}$, we now choose:

$$P_{\text{in}} = \left(\begin{smallmatrix} I_D \\ I_D \end{smallmatrix}\right), \quad P_{\text{out}} = (0 \mid I_D), \quad W_{\text{conv}} = \left(\begin{smallmatrix} 1_D & \mid & 0_D \\ 0_D & \mid & 1_D \end{smallmatrix}\right), \quad S_B = (F \mid 0), \quad S_C = (0 \mid F) \tag{6}$$

These weights compress both SSM input $\hat{x}_t = \begin{pmatrix} E\,x_{t-1} \\ E\,x_t \end{pmatrix}$ and operators $B_t = \tilde{E}\,x_{t-1}$, $C_t = \tilde{E}\,x_t$, where $\tilde{E} \equiv FE \in \mathbb{R}^{N \times V}$. Similarly to Eq. 5, the overall model output is now:

$$y_t = E^\top P_{\text{out}}\,\hat{y}_t = \sum_{\tau=0}^{t} E^\top (E\,x_\tau)\,(\tilde{E}\,x_{\tau-1})^\top (\tilde{E}\,x_t) \tag{7}$$

Applying ***Johnson–Lindenstrauss lemma*** (Johnson et al., 1984) twice, for arbitrarily small $0 < \varepsilon_k,\ \varepsilon_v < 1$, given sufficiently large model dimensions $D = O\big(\frac{\log V}{\varepsilon_v^2}\big)$, $N = O\big(\frac{\log V}{\varepsilon_k^2}\big)$ we can construct compression matrices $E,\ \tilde{E}$ such that $(\tilde{E}\,x_{\tau-1})^\top (\tilde{E}\,x_t) = \langle x_{\tau-1}, x_t \rangle + O(\varepsilon_k)$ and $E^\top (E\,x_{\tau-1}) = x_{\tau-1} + O(\varepsilon_v)$. Hence, given a unique context fact $(k^*, v^*)$ and a query $x_t = k^*$, the model output is $y_t = v^* + O(\varepsilon_v + \varepsilon_k)$. This implies the model solves MQAR up to an arbitrarily small reconstruction error, which depends on model dimensions. □

**MQAR with repetition.** In App. G we study two generalizations of MQAR, where either the keys or the queries repeat. Interestingly, we find that while single-layer Mamba models struggle with these harder tasks, two-layer models can successfully solve them.

### 3.2 Validating the Circuit: A Mechanistic Interpretability Perspective

**Equivalent recall circuits.** When conducting a mechanistic examination of Mamba solution to MQAR, we are faced with a problem: even if the circuit from Thm. 2 is indeed the exact learned one, it is rather unlikely to find these specific weights naturally within a trained model, since the *solution space* is wider. For example, if we suspect that weights $E$ and $P_{\text{in}}$ transform $x_t$ to $P_{\text{in}} E\,x_t$ through a specific operation, an equivalent operation would occur with $P'_{\text{in}} = P_{\text{in}} Q$ and $E' = Q^\top E$ for any orthogonal matrix $Q$. The evidence we seek for the circuit must be *invariant* under these orthogonal linear transformations. Such an inviraint operator is presented next.

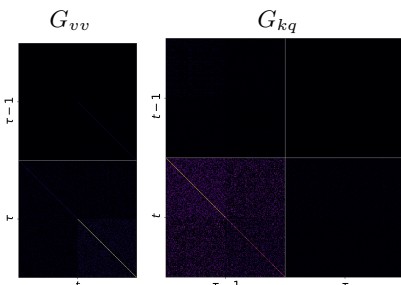

Figure 2: **Invariant operators.** Model inner mechanism is revealed when applying appropriate transformations. As predicted by Eq. 10, model core operation compares a query $q_t = x_t$ to a key $k_\tau = x_{\tau-1}$, then outputs the corresponding value $x_\tau = v_\tau$ into $y_t$. See detailed analysis in App. C.1.

Given the simplified model from Alg. 1, without any assumption on the weights, we define the input operator $\hat{E}_{\text{in}} \equiv \big( \operatorname{diag}(W_{\text{conv}}^0)\,P_{\text{in}}\,E \mid \operatorname{diag}(W_{\text{conv}}^1)\,P_{\text{in}}\,E \big)$, where we denote $W_{\text{conv}} = \big( W_{\text{conv}}^0 \mid W_{\text{conv}}^1 \big)$. Then, let us define the following operators:

$$\Pi_{v,\text{in}} = \hat{E}_{\text{in}}\,,\quad \Pi_{v,\text{out}} = E^\top P_{\text{out}}\,,\quad \Pi_{k,\text{in}} = S_B\,\hat{E}_{\text{in}}\,,\quad \Pi_{q,\text{in}} = S_C\,\hat{E}_{\text{in}}\,, \tag{8}$$

$$G_{vv} = \Pi_{v,\text{out}}\,\Pi_{v,\text{in}}\,,\quad G_{kq} = \Pi_{k,\text{in}}^\top\,\Pi_{q,\text{in}}\,, \tag{9}$$

such that model output is $y_t = \sum_{\tau=0}^{t} G_{vv}\,\xi_\tau\,\xi_\tau^\top\,G_{kq}\,\xi_t$, given input pairs $\xi_t \equiv \begin{pmatrix} x_{t-1} \\ x_t \end{pmatrix}$. If we modify model weights such that $G_{vv},\ G_{kq}$ are unchanged, we construct an *equivalent recall circuit* with the same performance. Therefore, if the solutions learned in practice are equivalent to the one presented in Theorem 2, we predict that the model weights will satisfy:

$$G_{vv} \approx \begin{pmatrix} 0 \\ I_V \end{pmatrix}\,,\quad G_{kq} \approx \begin{pmatrix} 0 & 0 \\ I_V & 0 \end{pmatrix} \tag{10}$$

matching $G_{vv},\ G_{kq}$ computed for the designed circuit weights. In Fig. 2 we can see that indeed, for a linear Mamba trained on MQAR (see App. J for implementation details), the solution weights satisfy this structure almost precisely. Specifically, we see that indeed $G_{kq}$ attends only to the query $x_t$ and the stored key $x_{\tau-1}$, while $G_{vv}$ attends only to the stored values $x_\tau$; all other matrix entries are $\approx 0$. However, we note that the nonzero blocks $G_{vv}^{t,\tau}$ and $G_{kq}^{\tau-1,t}$ do present more complex patterns than $\approx I_V$. As we discuss in App. D, these patterns are responsible for key and value selectivity.

**Hidden state as a hash table.** Intuitively, for a perfect recall, the SSM hidden state must encode the full context *information*. This fact-pairs information can be thought of as a simple key-value array: $\{\,1 : v_1,\ 2 : v_2,\ 3 : 0,\ \ldots,\ n : v_n,\ \ldots V_k : \ldots\,\}$, where each key entry contains the corresponding fact value $v_\tau$ if exists, else $0$. The ideal, non-realistic, circuit of Thm. 1 suggests such storage via an outer product key-value memory (see definitions and dimensions in App. C.2):

$$H_t \equiv P_{\text{out}}\,h_t = \sum_{n=1}^{N_f} v_n k_n^\top\,,\quad y_t = H_t\,q_t = \sum_{n=1}^{N_f} v_n k_n^\top q_t\,. \tag{11}$$

This is a $V \times V$ table, quite similar to the array presented above, though with values represented as one-hot vectors. A visualization can be found in Fig. 3 (left). $H_t$ is fairly interpretable, as entries are simply $1$ where facts exist and $0$ elsewhere. Can we observe such interpretable patterns within trained models? Let us consider the compressive SSM update from Eq. 7. It can be rewritten as

$$H_t' = \sum_{n=1}^{N_f} v_n' k_n'^\top, \quad y_t = E^\top \sum_{n=1}^{N_f} v_n' k_n'^\top q_t',$$

(12)

with the compressed tokens $v_n' = E\, v_n$, $k_n' = \tilde{E}\, k_n$, $q_t' = \tilde{E}\, q_t$.

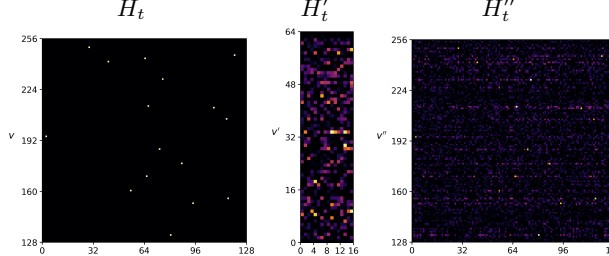

The hidden state $H_t'$ can be viewed as a **_hash table_**, for several reasons: (i) both mechanisms rely on a hash function. In our case, the model uses JL linear projections, which is by definition similarity-preserving hash function; (ii) in both cases, the internal representation is not directly interpretable (see Fig. 3, middle), while the final output is fairly interpretable (Fig. 3, right); (iii) efficiency is achieved by design through approximation, where the error is bounded with high probability due to the properties of the hash function; (iv) the interface closely follows that of a hash table:

Figure 3: **Interpreting the hidden state.** The context information (left) is compressed by the model into a hidden state (middle), which lacks clear interpretability. Projection onto vocabulary space (right) reveals interpretable pattern: $H_t'' \approx H_t$, indicating a compression-decompression scheme is learned.

First, Keys and queries are first hashed via $\text{hash}(x) = \tilde{E}x$. Next, while processing a fact, the model *stores* it into the table via $H_t = H_{t-1} + \text{encode}(v_n)\,\text{hash}(k_n)^\top$. Finally, when handling a query, the model *matches* $\text{hash}(q_t)$ with $\text{hash}(k_i)$ for all entries $i$, using inner products. If there exists a key $k_n$ such that $\text{hash}(k_n) \approx \text{hash}(q_t)$, the corresponding value is approximately *retrieved*.

**Inspecting the hash table.** We notice that $H_t'$ stores the full $V \times V$ information in a compressed $N \times D$ table, in a lossy yet decompressable way. To verify this mechanism in practice, we attempt to *decompress* the whole table, thus project it back onto vocabulary space. Let us define the decompressed tokens, $v_n'' = E^\top E\, v_n \approx v_n$ and $k_n'' = \tilde{E}^\top \tilde{E}\, k_n \approx k_n$. Using them, we can re-formulate the model operation as (see detailed definition in App. C.2):

$$H_t'' = \sum_{n=1}^{N_f} v_n'' k_n''^\top, \quad y_t = H_t'' q_t = \sum_{n=1}^{N_f} v_n'' k_n''^\top q_t$$

(13)

By this interpretation, the model is thought as using the full, sparse $V \times V$ table, which is an approximation of the non-compressive one: $H_t'' = E^\top H_t' \tilde{E} \approx H_t$.

Unfortunately, due to the non-uniqueness of solutions as discussed in Sec. 3.2, these actual operators do not necessarily exist as-is (see App. C.3 for details). Instead, one has to project the hidden state via $H_t'' = \Pi_{v,\text{out}} H_t' \Pi_{q,\text{in}}$. This way, we can verify that indeed, as seen in Fig. 3, the learned solution does leverage a hash-like mechanism. The actual SSM hidden state $H_t'$ is a random-looking efficiently compressed storage, which, when projected back to vocabulary space, seem to preserve the information stored in the context, as $H_t'' \approx H_t$.

**Noisy attention maps.** In addition to the hash-like recurrent view of the recall mechanism, we can alternatively view it through the lens of implicit attention maps. Following Ali et al. (2024); Dao & Gu (2024), for our simplified model, the attention matrix of the ideal circuit becomes $\alpha_{\tau t} = B_\tau^\top C_t = x_{\tau-1}^\top x_t$, such that $y_t = \sum_{\tau=0}^{t} x_\tau \alpha_{\tau t}$, equivalently to Eq. 5. Simply put, this formulation implies that given a query $x_t = q_t$, the model attends to previous tokens that satisfy $x_{\tau-1} = q_t$. For the realistic, compressive circuit, we have, based on Eq. 7,
$\alpha_{\tau t}' = B_\tau^\top C_t = (\tilde{E}\, x_{\tau-1})^\top (\tilde{E}\, x_t) = x_{\tau-1}^\top \tilde{E}^\top \tilde{E}\, x_t \approx$

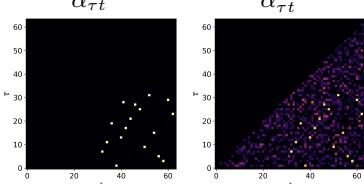

Figure 4: **Implicit attention maps.** The ideal attention map of a theoretical non-compressive model (left) is well approximated by a trained real-world model (right).

$x_{\tau-1}^\top x_t$, hence $\alpha_{\tau t}' \approx \alpha_{\tau t}$. (In equivalent circuits, the map is more complicated, $\alpha_{\tau t}' = \xi_\tau^\top G_{kq}\, \xi_t$, yet still $\approx \alpha_{\tau t}$). Indeed, computing these attention maps for both ideal and trained compressive

models, we verify in Fig. 4 that the learned map $\alpha'_{\tau t}$ functions as an approximation to the ideal one $\alpha_{\tau t}$, revealing circuit-level resemblance between regimes.

**Conv1D as a shift & copy.** An important claim in Thm. 2 is that the Conv1D applies copy and shift operations on the input sequence. Unfortunately, in practice the Conv1D weights cannot be observed directly, as they are mixed with other linear transformations. However, we can find evidence to its operation within $G_{vv}$ and $G_{kq}$. If the Conv1D performs copy and shift, we expect $G_{vv}\xi_\tau \approx x_\tau$ and $G_{kq}^\top \xi_\tau \approx \xi_{\tau-1}$. This can be verified given an MQAR sequence $x_t$, as visualized in Fig. 5. As expected, $x_t\, G_{vv}\, \xi_\tau \approx \delta_{t,\tau}$ and $\xi_t\, G_{kq}^\top \xi_\tau \approx \delta_{t,\tau-1}$. Lastly, in Tab. 10 (Appendix), we validate the importance

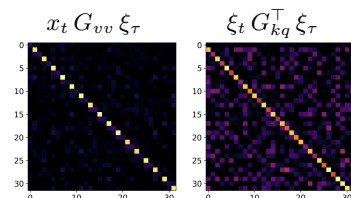

$x_t\, G_{vv}\, \xi_\tau \qquad \xi_t\, G_{kq}^\top\, \xi_\tau$

Figure 5: **Conv1D copy & shift**. Values are copied (left), as indicated by the dominant main diagonal. Keys are shifted (right), as indicated by the dominant second diagonal.

of the shifting operation, as well as the entire minimal circuit, through an ablation test. We find that a simplified Mamba block containing only a SSM and a Conv1d of length 2 achieves nearly perfect recall, but fails completely once the Conv1d is removed.

## 4 MAMBA RECALL SCALING LAWS

The analysis in the Sec. 3 identifies the recall circuit in Mamba and validates it through mechanistic experiments. However, it leaves us with an open question: how do recall capabilities vary as we change the model scale? In this section, we attempt to analytically answer this core question.

In Thm. 2 we suggest a mechanism that enables Mamba to approximately perform associative recall. The conclusion though, lacks a quantitative aspect. The following theorem addresses the question what model dimensions $D$, $N$ are required to solve an MQAR task of parameters $V$, $N_f$, $L$.

**Intuition.** The proof builds on the interpretation of the hidden state as information derived from a collection of approximately orthogonal hashed representations. These representations are constructed through two successive applications of the JL lemma. First, keys, queries and values in vocabulary space $\mathbb{R}^V$ are embedded into $\mathbb{R}^D$. Then, key and queries are further compressed into the state space $\mathbb{R}^N$. Finally, for sufficiently large state and embedding dimensions, successful retrieval with high probability is ensured through a statistical analysis that bounds the error introduced at each compression step. This analysis tracks how compression errors propagate through the circuit, and guarantees that the signal-to-noise ratio remains sufficiently high.

**Lemma 1 (JL-Based AR Scaling Law).** *Given a vocabulary size $V$, a single-layer simplified Mamba with dimensions $N < D < V$,* expand $= 2$, $D_{\mathrm{conv}} = 2$ *can **perfectly solve** an MQAR task if model dimensions satisfy $\varepsilon_v < 1$, $\varepsilon_k < 1$ and $\varepsilon_v + \varepsilon_k + N_f\, \varepsilon_v\, \varepsilon_k < \frac{1}{2}$, where*

$$\varepsilon_v = \sqrt{\frac{c \log V}{D}},\ \varepsilon_k = \sqrt{\frac{c \log V}{N}},\ c = 4.$$

***Proof sketch***. (Full proof can be found in App. D.) Assume weights are as in Thm. 2. Given a query $x_t = q_t = k_m$, the model output $y_t$ has entries $p$ given by $y_t^p = e_p^\top y_t = \sum_{\tau=0}^{t}\left(e_p^\top G_E\, x_\tau\right)\left(x_{\tau-1}^\top G_{\tilde E}\, k_m\right)$, where $e_p$ is the $p$ basis vector, $G_E = E^\top E$ and $G_{\tilde E} = \tilde E^\top \tilde E$. Since $E, \tilde E$ are both JL embeddings, we have $e_p^\top (G_E - I_V)\, x_\tau \in (-\varepsilon_v, \varepsilon_v)$ and $x_{\tau-1}^\top (G_{\tilde E} - I_V)\, k_m \in (-\varepsilon_k, \varepsilon_k)$. Assuming orthogonal key-value embeddings, as discussed in D (Appendix), we bound the *correct*, *wrong* and *empty* entries $i$, $j$, $l$ respectively by:

$$y_t^i > 1 - \varepsilon_v - \varepsilon_k - N_f\, \varepsilon_v\, \varepsilon_k, \quad y_t^j < \varepsilon_v + \varepsilon_k + N_f\, \varepsilon_v\, \varepsilon_k, \quad y_t^l < N_f\, \varepsilon_v\, \varepsilon_k < y_t^j \tag{14}$$

For perfect retrieval, we need $y_t^i = \max_p\, y_t^p$, thus $y_t^i - y_t^j \approx 1 - 2(\varepsilon_v + \varepsilon_k + N_f\, \varepsilon_v\, \varepsilon_k) > 0$. $\quad\square$

**Scaling laws for recall probability.** The JL-based scaling law developed in Lemma 1 provides a useful bound, quantifying model scales $D$, $N$ required for perfect recall. However, a linear summation of the $\varepsilon_k\, \varepsilon_v$ terms (mismatching-facts noise) might look too strict. In reality, the more *probable* case is that some of these noise term are of opposite signs, thus cancel each other. This insight leads us to the following, more precise probabilistic scaling law.

**Theorem 3** (**Probabilistic AR Scaling Law**). *A single-layer simplified Mamba can solve an MQAR task with a probability approximated by*

$$p_{\text{success}} \approx \left( \Phi \left( \sqrt{\frac{2ND}{N_f}} \right) \right)^{\frac{V}{2}} \tag{15}$$

*in the limit of $N_f >> N, D$, where $\Phi$ is the standard Normal CDF.*

***Proof sketch***. (Full proof can be found in App. D.) We view the distortion terms in Eq. 14 as random variables rather than tight bounds. Using the distortion matrices $\epsilon_v \equiv G_E - I_V$ and $\epsilon_k \equiv G_{\tilde{E}} - I_V$, given a query $q_t = k_m$, we have for the *correct*, *wrong* and *empty* value entries $i, j, l$, respectively:

$$y_t^i \approx 1 + \epsilon_v^{ii} + \epsilon_k^{mm} + \sum_p^{N_f} \epsilon_v^{ip} \epsilon_k^{pm}, \qquad y_t^j \approx \epsilon_k^{nm} + \sum_p^{N_f} \epsilon_v^{jp} \epsilon_k^{pm}, \qquad y_t^l \approx \sum_p^{N_f} \epsilon_v^{jp} \epsilon_k^{pm} \tag{16}$$

For JL embeddings, the off-diagonal elements have zero mean and variances $\approx \frac{1}{D}, \frac{1}{N}$ respectively. Each non-matching fact has an i.i.d contribution of variance $\sigma_f^2 \approx \frac{1}{ND}$. For $N_f >> N, D$, the $\epsilon_v^{ii}, \epsilon_k^{mm}, \epsilon_k^{nm}$ terms are negligible, and by CLT, the sum distributes as $\sim \mathcal{N}(0, \frac{N_f}{ND})$, hence $(y_t^i - y_t^j)$, $(y_t^i - y_t^l) \sim \mathcal{N}\left(1, \frac{2N_f}{DN}\right)$. Considering the total $V_v$ value entries, we obtain Eq. 15. $\square$

**Remark 1** (**Scaling Laws for model dimensions**). An important consequence of Theorem 3 is the following remark. In the large-$N_f$ approximation regime, a single-layer simplified Mamba model can solve an MQAR task with probability $> 1 - \delta$ if model dimensions satisfy $D, N \gtrsim \log V$ (JL condition), and $DN \gtrsim 2 N_f n_\sigma^2$, with $n_\sigma = \Phi^{-1}(1 - \frac{2\delta}{V})$. The condition above can be further approximated for $\delta << 1$. In this regime, approximating $\Phi$, the dimension requirement becomes $DN \gtrsim 4 N_f \log\left(\frac{V}{2\delta}\right)$, hence overall:

$$DN = \Omega\left( N_f \log\left(\frac{V}{\delta}\right) \right) \tag{17}$$

This concludes the scaling laws derivation. Given MQAR parameters $V$, $N_f$ and $L$, we have, up to constant factors, simple conditions on model dimensions required to solve MQAR with probability $> 1 - \delta$, expressed in Eq. 17.

**Multi-layer, multi-head and Mamba-2 scaling laws.** Further theoretical analysis, extending the results of this section to multi-layer and multi-head modles, is provided in App. E and App. F.

## 5 Scaling Laws Experiments

### 5.1 Single-layer Model: dimension scaling laws

In this section, we empirically verify that our proposed scaling laws for AR in Mamba hold in practice. We vary both model and state dimensions $(D, N)$ in multiple MQAR settings (namely, different combinations of $V, N_f$, and $L$), and measure the resulting MQAR accuracy. We test both linear and full single-layer Mamba models. To isolate aspects of generalization, each model is trained with 10 different seeds, and we report the maximum accuracy to avoid optimization artifacts (also see App. J.5). A very large training and test set is used to ensure that the results are representative and not affected by generalization issues. For additional technical details, see Appendix J.

**Validating consistency under our proposed simplification.** Our theory was developed in the reduced scope of a simplified linear model; therefore, it is important to compare it to a full model which includes nonlinearities, discretization, gating, etc. The results are reported in Fig. 6. We first focus on the third column, which corresponds to the simplified 1-layer linear model described in Alg. 1, and the fourth column, which shows the full 1-layer Mamba model. The main observation is that the full model produces recall scaling laws that closely match those of the simplified model. In particular, the regions in the $N, D$ space that achieve near-perfect recall are almost identical for both models. This result demonstrates that the simplifications made for the theoretical analysis did not distort the core operation of Mamba's recall circuit, further motivating our approach. Additional comparison between models performance can be found in Figure 12 (Appendix).

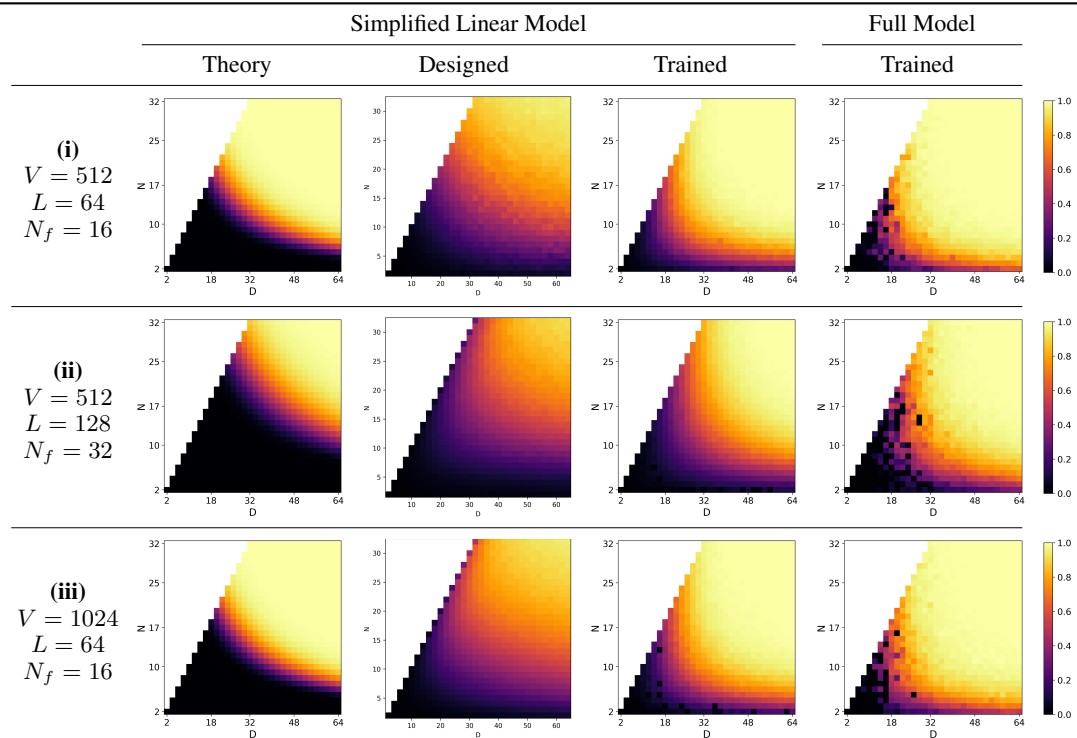

Figure 6: **Empirical versus theoretical recall scaling laws.** Left to right: (a) Theoretical approximate law as depicted in Eq. 15; (b) Simplified linear model with fixed weights, designed as presented in Thm. 2; (c) Trained linear model. (d) Trained full Mamba model. Pixel color represents accuracy. Each row represents a different regime of MQAR parameters.

**Comparison to theoretical predictions.** We turn to the first two columns in Fig. 6, which provide two realizations of our theory. The first column evaluates the theoretical expression in Eq. 15 on the grid, and the second column presents the results of a designed model with weights set as described in Thm. 2. As predicted by our theory, a clear tradeoff is observed: for larger $D$, the minimal $N$ required for perfect recall is lower, and vice versa. This matches our $ND \gtrsim f(V, N_f)$ prediction from Eq. 17, where the RHS is constant in each MQAR setting. Most importantly, this trend also matches our empirical results in columns 3 and 4, thus bridging between the theory-based simulations and both empirical models. We note that while the accuracy values reflect the same trends, they do not align perfectly. The reasons are as follows: (i) our theory is based on a specific model construction with fixed weight assignments (column 2). As such, the proposed solution is not necessarily optimal compared to the trained variant and should be considered a lower bound rather than a direct predictor; (ii) our analysis relies on statistical bounds, which are not always tight; and (iii) the theoretical law involves several simplifications, thus should be regarded as a coarse approximation.

Lastly, the similar scaling behavior seen above can lead to the following hypothesis: both the simplified and full models depend primarily on a hash-like key-value compression mechanism (the full model additionally benefits from nonlinear operators). Therefore, the compressed dimensions $N, D$ continue to form the main bottleneck for associative recall, leading to comparable scaling laws. We leave a detailed investigation of this hypothesis to future work.

## 5.2 MULTI-LAYER AND MULTI-HEAD SCALING LAWS

In this section, we conduct empirical studies to validate our predictions for recall capabilities of multi-layer models and multi-head models (Thm. 4 and Thm. 5; see App. F for proofs).

**Multi-layer scaling laws.** To validate our theoretical analysis, we train *simplified multi-layer Mamba* models with varying states $N$ and depths $\Lambda$ on MQAR. Our findings are concluded in Fig. 7. In an accurate fit with Thm. 4, we empirically verify that recall capacity of a simplified multi-layer mamba is uniquely determined by $\Lambda N$, which we term as the model *effective state size* $N_{\text{eff}}$.

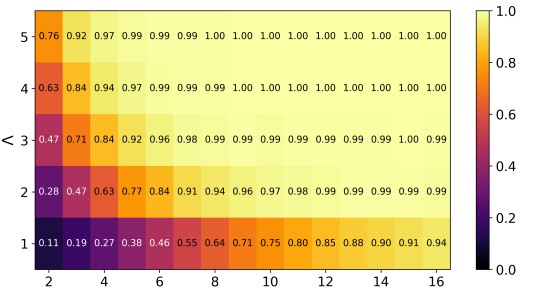 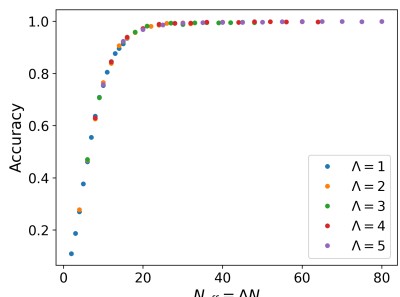

Figure 7: **Scaling laws for multi-layer recall in Mamba.** Exactly as predicted in Thm. 4 and in Rem. 2 (Appendix), the MQAR performance of a simplified multi-layer Mamba model of a state size $N$ and $\Lambda$ layers is uniquely determined by its effective state size $N_{\text{eff}} = \Lambda N$, rather than by each factor separately.

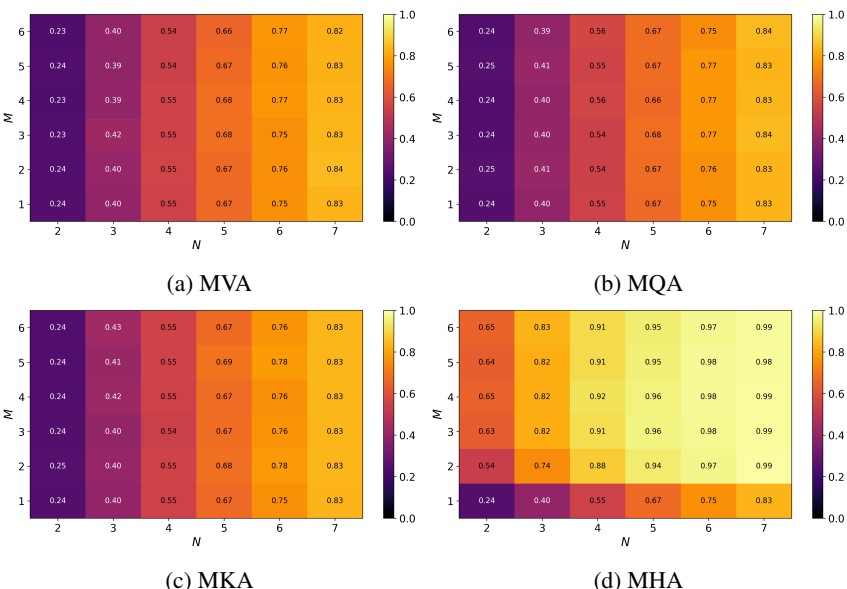

Figure 8: **Scaling laws for multi-head SSM patterns**. In order to improve recall by splitting into $M$ heads, both key and query dimensions should increase from $N$ to $MN$ (MHA, bottom right). Sharing keys (MQA), values (MKA) or both (MVA) across SSM heads implies similar accuracy as in the single-headed SSM.

**Multi-head scaling laws.** Our experiments consist of training *simplified multi-head Mamba-2* models on MQAR, varying state size $N$ and number of heads $M$ (Note that total dimension is fixed: the $D$ SSM channels are split between $M$ heads). Results are summarized in Fig. 8. Exactly as predicted in Thm. 5, we find that in order to effectively improve recall via multi-head patterns, the dimensions of both keys and queries should be increased. Among the four different head patterns, this is only available using the MHA pattern (See additional experiments in App. H.2.)

## 6 CONCLUSIONS

In this paper, we identify the neural algorithm underlying associative recall in Mamba models. We build on this understanding to develop a theoretical framework that predicts, given model dimensions, the maximum vocabulary size and number of facts over which accurate recall can be achieved in the MQAR task. Our theoretical predictions are shown to closely align with empirical results.

As with many theoretical works, our analysis relies on simplified models, and thus has certain limitations. In particular, we do not fully characterize how each component of the Mamba architecture contributes to recall performance. For example, the theoretical role of the gating branch in recall remains unclear. For future work, we aim to extend this analysis to other architectures, such as xL-STM (Beck et al., 2024), RWKV Peng et al. (2023), DeltaNet (Yang et al., 2024), with the goal of understanding how architectural modifications impact recall capabilities.

## ETHICS STATEMENT

This work analyzes the recall capabilities of Mamba models and linear RNNs, which are crucial for deploying recurrent LLMs in real-world systems. Improving memory efficiency and retrieval understanding can lead to more reliable and controllable architectures, with broad benefits for AI applications. We do not anticipate any direct negative societal impacts beyond those generally associated with LLMs.

## REPRODUCIBILITY STATEMENT

We provide the source code for the key experiments and include detailed descriptions of the full configurations for all experiments, along with instructions for setting up and evaluating the models. All datasets, models, prompts, and metrics used in each experiment are explicitly stated in the paper to ensure full transparency and reproducibility.

## THE USE OF LARGE LANGUAGE MODELS (LLMS)

ChatGPT (OpenAI, 2025) was used as a general-purpose writing and editing assistant in the preparation of this manuscript. Its role was limited to helping polish the presentation: rephrasing sentences for clarity, improving grammar and flow, suggesting alternative wording, and condensing or expanding explanations when asked. Importantly, ChatGPT was not involved in the research ideation, design, implementation, analysis, or interpretation of results. Additionally, ChatGPT was used for coding assistance.

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

# A   NOTATION AND DIMENSIONS

This appendix provides reference tables for the notation appearing in the paper, as well as tables for tensor dimensions, as used in our algorithms and analysis.

## A.1   MULTI-QUERY ASSOCIATIVE RECALL (MQAR)

Table 1 includes notation for the MQAR benchmark.

Table 1: Notation for MQAR

| Symbol | Meaning | Value | Notes |
|--------|---------|-------|-------|
| $\mathcal{V}$ | Vocabulary | - | |
| $\mathcal{V}_k$ | Key vocabulary | - | |
| $\mathcal{V}_v$ | Value vocabulary | - | |
| $V$ | Vocabulary size | $|\mathcal{V}|$ | |
| $V_k$ | Key vocabulary size | $|\mathcal{V}_k|$ | $V_k = \frac{V}{2}$, unless stated otherwise |
| $V_v$ | Value vocabulary size | $|\mathcal{V}_v|$ | $V_v = \frac{V}{2}$, unless stated otherwise |
| $N_f$ | Number of facts | - | |
| $L$ | Total context length | - | $L \geq 4N_f$, following implementation (Arora et al., 2023) |
| - | Facts section length | $2N_f$ | $N_f$ key-value token pairs |
| - | Queries section length | $L - 2N_f$ | $N_f$ query tokens, $L - 3N_f$ random padding tokens |

## A.2   MAMBA MODEL

Tables 2, 3 include detailed notation and dimensions for Mamba model and tensors.

For detailed dimensions of hidden state and key-value matrices $H_t$, $H_t'$, $H_t''$ used in Sec. 3.2 for mechanistic interpretability, see Appendix C.2.

For Mamba-2 dimensions, see Appendix F.2.

Table 2: Notation for Mamba

| Symbol | Meaning | Value | Notes |
|--------|---------|-------|-------|
| $D$ | Embedding size | - | |
| $N$ | SSM State size | - | |
| expand | SSM expansion factor | $\frac{D_{\text{in}}}{D}$ | |
| $D_{\text{in}}$ | SSM number of channels | expand $\times D$ | |
| $D_{\text{conv}}$ | Convolution kernel size | - | |
| $\Lambda$ | Number of layers | - | |

## A.3   SSM STATE UPDATE RULE

In the following Appendix, we clarify shapes and dimensions for SSM state update.

**Full Mamba model.**   For each timestep $t$, the SSM vectors $B_t, C_t \in \mathbb{R}^N$ and discrete matrices $\bar{A}_t, \bar{B}_t \in \mathbb{R}^{D_{\text{in}} \times N}$ are computed as:

$$B_t = S_B\,\hat{x}_t, \quad C_t = S_C\,\hat{x}_t, \quad \Delta_t = \text{SoftPlus}(S_\Delta \hat{x}_t), \quad \bar{A}_t = \exp(\Delta_t A), \quad \bar{B}_t = \Delta_t B_t \quad (18)$$

where $S_B, S_C, S_\Delta$ are linear projection matrices, SoftPlus is a smooth approximation of ReLU, and biases are omitted for simplicity.

Then, for any timestep $t$, each SSM channel $d = 1, \ldots, D_{\text{in}}$ has a scalar input $\hat{x}_t^d \in \mathbb{R}$ and a scalar output $\hat{y}_t^d \in \mathbb{R}$. The output is computed through a recurrent state-update step:

$$\begin{cases} \mathbf{h}_t^d = \bar{\mathbf{A}}_t^d \odot \mathbf{h}_{t-1}^d + \hat{x}_t^d\,\bar{\mathbf{B}}_t^d \\ \hat{y}_t^d = \mathbf{h}_t^d \cdot \mathbf{C}_t \end{cases} \quad (19)$$

Table 3: Dimensions for Mamba

| Symbol | Meaning | Dimension | Notes |
|---|---|---|---|
| $x$ | Model input sequence (one-hot) | $\mathbb{R}^{V \times L}$ | $x_t \in \mathbb{R}^V$ |
| $y$ | Model output sequence (one-hot) | $\mathbb{R}^{V \times L}$ | $y_t \in \mathbb{R}^V$ |
| $x^e$ | Model input sequence (embedded) | $\mathbb{R}^{D \times L}$ | $x_t^e \in \mathbb{R}^D$ |
| $\hat{x}$ | SSM input sequence | $\mathbb{R}^{D_{\text{in}} \times L}$ | $\hat{x}_t \in \mathbb{R}^{D_{\text{in}}}$ |
| $\hat{y}$ | SSM output sequence | $\mathbb{R}^{D_{\text{in}} \times L}$ | $\hat{y}_t \in \mathbb{R}^{D_{\text{in}}}$ |
| $\hat{z}$ | Gate sequence | $\mathbb{R}^{D_{\text{in}} \times L}$ | $\hat{z}_t \in \mathbb{R}^{D_{\text{in}}}$ |
| $E$ | Model embedding | $\mathbb{R}^{D \times V}$ | |
| $W_{\text{conv}}$ | Convolution kernel | $\mathbb{R}^{D_{\text{in}} \times D_{\text{conv}}}$ | |
| $P_{\text{in}}$ | Input projection | $\mathbb{R}^{D_{\text{in}} \times D}$ | |
| $P_{\text{out}}$ | Output projection | $\mathbb{R}^{D \times D_{\text{in}}}$ | |
| $S_B$ | $B$ projection | $\mathbb{R}^{N \times D_{\text{in}}}$ | |
| $S_C$ | $C$ projection | $\mathbb{R}^{N \times D_{\text{in}}}$ | |
| $A$ | SSM transition coefficient | $\mathbb{R}^N$ | |
| $B$ | SSM input coefficient | $\mathbb{R}^{N \times L}$ | $B_t \in \mathbb{R}^N$ |
| $C$ | SSM readout coefficient | $\mathbb{R}^{N \times L}$ | $C_t \in \mathbb{R}^N$ |
| $\Delta$ | SSM discretization parameter | $\mathbb{R}^{D_{\text{in}} \times L}$ | $\Delta_t \in \mathbb{R}^{D_{\text{in}}}$ |
| $\bar{A}$ | SSM Discrete transition coefficient | $\mathbb{R}^{D_{\text{in}} \times N \times L}$ | $\bar{A}_t \in \mathbb{R}^{D_{\text{in}} \times N}$ |
| $\bar{B}$ | SSM Discrete input coefficient | $\mathbb{R}^{D_{\text{in}} \times N \times L}$ | $\bar{B}_t \in \mathbb{R}^{D_{\text{in}} \times N}$ |
| $h$ | SSM hidden state | $\mathbb{R}^{D_{\text{in}} \times N \times L}$ | $h_t \in \mathbb{R}^{D_{\text{in}} \times N}$ |
| $\alpha$ | SSM equivalent attention matrix | $\mathbb{R}^{L \times L}$ | |

where $\mathbf{h}_t^d, \bar{\mathbf{A}}_t^d, \bar{\mathbf{B}}_t^d, \mathbf{C}_t \in \mathbb{R}^N$ are all $N$-vectors, $\odot$ denotes an element-wise product, and $\cdot$ a vector dot-product. (For $\hat{x}_t^d \bar{\mathbf{B}}_t^d$ we have a scalar multiplying a vector.)

Per-channel, we have a state vector $\mathbf{h}_t^d \in \mathbb{R}^N$. In total, all channels together form a state matrix $h_t \in \mathbb{R}^{D_{\text{in}} \times N}$.

**Simplified Mamba model.** In order to ease Mamba model analysis, we preset a simplified SSM, where discretization is removed, and $\bar{A}$ is ignored (by setting its values to 1). The SSM operation can be now written as:

$$\begin{cases} \mathbf{h}_t^d = \mathbf{h}_{t-1}^d + \hat{x}_t^d \, \mathbf{B}_t \\ \hat{y}_t^d = \mathbf{h}_t^d \cdot \mathbf{C}_t \end{cases} \tag{20}$$

Note that $\bar{\mathbf{B}}_t^d = \mathbf{B}_t \in \mathbb{R}^N$ is now shared between channels.

If we consider the channel dimension, we have a vector input and output $\hat{x}_t, \hat{y}_t \in \mathbb{R}^{D_{\text{in}}}$, and a state matrix $h_t \in \mathbb{R}^{D_{\text{in}} \times N}$. Equation 20 can be written in a matrix form as:

$$\begin{cases} h_t = h_{t-1} + \hat{x}_t \, B_t^\top \\ \hat{y}_t = h_t \, C_t \end{cases} \tag{21}$$

where $\hat{x}_t \, B_t^\top \in \mathbb{R}^{D_{\text{in}} \times N}$ is a vector-vector outer product.

## B  MAMBA RECALL CIRCUITS: PROOFS

**Theorem 1** (**Perfect non-compressive recall circuit**). *Given a vocabulary size $V$, a single-layer simplified Mamba with dimensions $D = V$, $N = V$, expand $= 2$, $D_{\text{conv}} = 2$ can **perfectly solve** an MQAR task, i.e. with recall probability = 1.*

*Proof*. We construct model weights as follows:

**Embeddings.**  Let us choose the trivial identity embedding $\in \mathbb{R}^{V \times V}$:

$$E = I_V \tag{22}$$

The embedded input vectors are then trivially $x_t^e = E\, x_t = x_t$.

**Projections.**  We choose:
$$P_{\text{in}} = \left(I_V \mid I_V\right)^\top$$
$$P_{\text{out}} = \left(0 \mid I_v\right) \tag{23}$$

The input projection simply duplicates the state:

$$x_t^p = P_{\text{in}}\, x_t^e = \begin{pmatrix} x_t \\ x_t \end{pmatrix} \tag{24}$$

The output projection extracts the "current" half of duplicated current-previous vectors, then projects it onto the value vocabulary space $\mathcal{V}_v$.

**Conv1D kernel.**  We choose

$$W_{\text{conv}} = \left(W_p \mid W_c\right) = \begin{pmatrix} 1_V & \mid & 0_V \\ 0_V & \mid & 1_V \end{pmatrix} \tag{25}$$

with $1_V$ and $0_V$ all-ones and all-zeros column vectors in $\mathbb{R}^V$, respectively. This implies:

$$\text{diag}\left(W_c\right) = \left(0 \mid I_V\right)$$
$$\text{diag}\left(W_p\right) = \left(I_V \mid 0\right) \tag{26}$$

This kernel applies identity and shift operations, to form a vector:

$$\hat{x}_t = \text{Conv1D}\left(x_t^p,\, W_{\text{conv}}\right) = \begin{pmatrix} x_{t-1}^e \\ x_t^e \end{pmatrix} = \begin{pmatrix} x_{t-1} \\ x_t \end{pmatrix} \tag{27}$$

This stacked vector is the input to the SSM.

**SSM projections and embeddings.**  For the SSM selection matrices, let us choose:

$$S_B = \left(I_V \mid 0\right)$$
$$S_C = \left(0 \mid I_V\right) \tag{28}$$

That is to say, $S_B$ outputs the previous token,

$$B_t = S_B\, \hat{x}_t = x_{t-1} \tag{29}$$

while $S_C$ outputs the current token:

$$C_t = S_C\, \hat{x}_t = x_t \tag{30}$$

As we will soon see, this allows $S_B$ and $S_C$ to function as key and query extractors, respectively.

**Overall mechanism.**  Let us now look at consecutive context token pairs $(x_{\tau-1}, x_\tau)$, not necessarily a key-value pair, and a current token $x_t$, not necessarily a (query) key token. Let us also denote the stacked token-pair vectors as $\xi_t \equiv \begin{pmatrix} x_{t-1} \\ x_t \end{pmatrix}$. Note that in this non-compressive scheme described, the SSM input is exactly $\hat{x}_t = \xi_t$ (see equation 27).

With the matrix construction defined above, the SSM recurrent update rule from Alg. 1 now becomes:

$$h_t - h_{t-1} = \hat{x}_t\, B_t^\top = \xi_t\, x_{t-1}^\top \tag{31}$$

Hence, assuming $h_0 = 0$, we have

$$h_t = \sum_{\tau=0}^{t} \hat{x}_t \, B_t^\top = \sum_{\tau=1}^{t} \xi_\tau \, x_{\tau-1}{}^\top$$

$$\hat{y}_t = h_t \, C_t = \sum_{\tau=1}^{t} \xi_\tau \, x_{\tau-1}{}^\top x_t \tag{32}$$

The overall model output can be now expressed as:

$$y_t = E^\top P_{\text{out}} \, \hat{y}_t = \sum_{\tau=1}^{t} (P_{\text{out}} \, \xi_\tau) \, x_{\tau-1}{}^\top x_t \tag{33}$$

However, we have constructed $P_{\text{out}}$ to extract the current token:

$$P_{\text{out}} \, \xi_\tau = \left(0 \mid I_V\right) \xi_\tau = x_\tau \tag{34}$$

We then have:

$$y_t = \sum_{\tau=1}^{t} x_\tau \, x_{\tau-1}{}^\top x_t \tag{35}$$

Hence, if we now have a *query* $x_t = q_m \in \mathcal{V}_k$, then

$$y_t = \sum_{n=1}^{N_f} x_\tau \, x_{\tau-1}{}^\top q_m \tag{36}$$

If we know the query corresponds to a *key* fact in the context $q_m = k_{n^*}$ which is a part of the a fact pair $(k_{n^*}, v_{n^*})$, then finally, since one-how keys and values are orthonormal:

$$y_t = \sum_{n=1}^{N_f} v_n \, k_n{}^\top k_{n^*} = \sum_{n=1}^{N_f} v_n \, \delta_{n,n^*} = v_{n^*} \tag{37}$$

with $\delta_{i,j}$ the Kronecker delta.

This means the model achieves perfect recall: for any query, it exactly retrieves the correct value from the context.

$\square$

**Theorem 2 (Efficient compressive recall circuit).** *Given a vocabulary size $V$, a single-layer simplified Mamba with dimensions $D, N = O(\log V)$, expand $= 2$, $D_{\text{conv}} = 2$ can solve an MQAR task **with high probability**. (See Thm. 3 for quantitive analysis.)*

*Proof.* We apply quite the same mechanism described in Theorem 1 , with modifications to enable compression and decompression.

**Projectors.**    As in the non-compressive circuit, let us define similar *previous and current* projectors, now in $\mathbb{R}^{D \times 2D}$ compressed space:

$$M_p = \left(I_D \quad 0\right)$$

$$M_c = \left(0 \quad I_D\right) \tag{38}$$

**Embeddings.**    Given a general embedding $E \in \mathbb{R}^{D \times V}$, the embedded input vectors are then $x_t^e = E \, x_t \in \mathbb{R}^D$.

**Projections.**    We choose:

$$P_{\text{in}} = M_p^\top + M_c^\top = \left(I_D \mid I_D\right)^\top$$

$$P_{\text{out}} = M_c = \left(0 \mid I_D\right) \tag{39}$$

The input projection simply duplicates the embedded state:

$$x_t^p = P_{\text{in}} \, x_t^e = \begin{pmatrix} E \, x_t \\ E \, x_t \end{pmatrix} \tag{40}$$

The output projection extracts the "current" half of duplicated current-previous vectors.

**Conv1D kernel.** We choose

$$W_{\mathrm{conv}} = \begin{pmatrix} W_p \mid W_c \end{pmatrix} = \begin{pmatrix} 1_D \mid 0_D \\ 0_D \mid 1_D \end{pmatrix} \tag{41}$$

such that

$$\begin{aligned}
\mathrm{diag}(W_c) &= \begin{pmatrix} 0 \mid I_D \end{pmatrix} \\
\mathrm{diag}(W_p) &= \begin{pmatrix} I_D \mid 0 \end{pmatrix}
\end{aligned} \tag{42}$$

This kernel applies identity and shift operations, to form

$$\hat{x}_t = \mathrm{Conv1D}\big(x_t^p,\, W_{\mathrm{conv}}\big) = \begin{pmatrix} x_{t-1}^e \\ x_t^e \end{pmatrix} = \begin{pmatrix} E\,x_{t-1} \\ E\,x_t \end{pmatrix} \tag{43}$$

**SSM projections and embeddings.** Let $F \in \mathbb{R}^{N \times D}$ be an embedding matrix. We choose:

$$\begin{aligned}
S_B &= F\,M_p = \begin{pmatrix} F \mid 0 \end{pmatrix} \\
S_C &= F\,M_c = \begin{pmatrix} 0 \mid F \end{pmatrix}
\end{aligned} \tag{44}$$

That is to say, $S_B$ outputs a compressed previous token,

$$B_t = S_B\,\hat{x}_t = F\,M_p\,\hat{x}_t = F\,x_{t-1}^e = F\,E\,x_{t-1} = \tilde{E}x_{t-1} \in \mathbb{R}^N \tag{45}$$

while $S_C$ outputs a compressed current token,

$$C_t = S_C\,\hat{x} = F\,M_c\,\hat{x}_t = F\,x_t^e = F\,E\,x_t = \tilde{E}x_t \in \mathbb{R}^N \tag{46}$$

where we have introduced $\tilde{E} \equiv FE$.

**Overall mechanism.** Given the context consecutive key-value token pairs $(x_{\tau-1}, x_\tau) \equiv (k_\tau, v_\tau)$, and a query token $x_t \equiv q_t$, we notice the matrix setup defined above describes a simple compression-decompression scheme.

The SSM hidden state and output from 32 are now expressed as:

$$\begin{aligned}
h_t &= \sum_{\tau=0}^{t} \hat{x}_t\,B_t^\top = \sum_{(\tau-1)\in\{\tau_k\}}^{t} \begin{pmatrix} E\,x_{\tau-1} \\ E\,x_\tau \end{pmatrix} (\tilde{E}\,x_{\tau-1})^\top \\
\hat{y}_t &= h_t\,C_t = \sum_{(\tau-1)\in\{\tau_k\}}^{t} \begin{pmatrix} E\,x_{\tau-1} \\ E\,x_\tau \end{pmatrix} (\tilde{E}\,x_{\tau-1})^\top (\tilde{E}\,x_t)
\end{aligned} \tag{47}$$

If we recall that $P_{\mathrm{out}} = M_c = \begin{pmatrix} 0 \mid I_D \end{pmatrix}$, the overall layer output is therefore:

$$y_t = E^\top P_{\mathrm{out}}\,\hat{y}_t \sum_{(\tau-1)\in\{\tau_k\}}^{t} E^\top (E\,x_{\tau-1})\,(\tilde{E}\,x_{\tau-1})^\top (\tilde{E}\,x_t) \tag{48}$$

This expression can be seen as simple bilinear function of the original one-hot (non-compressed) tokens:

$$y_t = \sum_{\tau=0}^{t} (E^\top E)\,x_\tau\,x_{\tau-1}{}^\top (\tilde{E}^\top \tilde{E})\,x_t \tag{49}$$

If we further look at the Gram matrices, defined as

$$\begin{aligned}
G_E &\equiv E^\top E \in \mathbb{R}^{V \times V} \\
G_{\tilde{E}} &\equiv \tilde{E}^\top \tilde{E} \in \mathbb{R}^{V \times V}
\end{aligned} \tag{50}$$

(which, in our case, corresponds to applying successive compression and decompression), the output vector can now be expressed as:

$$y_t = \sum_{\tau=0}^{t} G_E\,x_\tau\,x_{\tau-1}{}^\top\,G_{\tilde{E}}\,x_t \tag{51}$$

This would have reduced to a perfect retrieval, as described in the perfect case, only if we had perfect embeddings, satisfying $G_E = I_V$ and $G_{\tilde{E}} = I_V$, which is, of course, impossible for $N < D < V$.

However, for proper, large enough dimensions, we can approximately achieve perfect recall, as we will now state.

**Applying JL Lemma.** Using *Johnson–Lindenstrauss lemma* (Johnson et al., 1984) twice, for arbitrarily small $0 < \varepsilon_k,\ \varepsilon_v < 1$, if model dimensions are large enough, $D > \frac{c \log V}{\varepsilon_v^2}$, $N > \frac{c \log V}{\varepsilon_k^2}$ for some constant $c$, we can construct embedding matrices $E, \tilde{E}$ for which

$$\left| \langle E x_i, E x_j \rangle - \langle x_i, x_j \rangle \right| = \left| x_i^\top (E^\top E - I) x_j \right| \leq \varepsilon_v \|x_i\| \|x_j\| \tag{52}$$

and similarly for $\tilde{E}$ and $\varepsilon_k$.

Since our input vectors are all one-hot (satisfying $\|x_i\| = 1$):

$$\left| \langle E x_i, E x_j \rangle - \langle x_i, x_j \rangle \right| \leq \varepsilon_v \tag{53}$$

For such embeddings, we can further bound the inner products for one-hot vectors, by:

$$x_i^\top G_E x_j = (E x_i)^\top (E x_j) \in \begin{cases} (1 - \varepsilon_v,\ 1 + \varepsilon_v) & \text{if } i = j \\ (-\varepsilon_v,\ \varepsilon_v) & \text{if } i \neq j \end{cases} \tag{54}$$

and similarly:

$$x_i^\top G_{\tilde{E}} x_j = (\tilde{E} x_i)^\top (\tilde{E} x_j) \in \begin{cases} (1 - \varepsilon_k,\ 1 + \varepsilon_k) & \text{if } i = j \\ (-\varepsilon_k,\ \varepsilon_k) & \text{if } i \neq j \end{cases} \tag{55}$$

Even without an exact analysis (which is further developed later in Lemma 1 and Theorem 3), we claim that for small enough $\varepsilon_v, \varepsilon_k$, almost-perfect recall is achievable. To see why, let us examine the output vector $y_t$.

**Entry-wise analysis.** As before, let us assume the context contains a single correct fact pair with a key $m$ and a value $i$, both one-hot encoded by token vectors $(k_m, v_i)$.

The vector $y_t$ has $V$ entries in total, divided into:

- 1 *correct* value entry $i$, corresponds to the correct fact value $v_i$.

- $N_f - 1$ *wrong* value entries, matching $N_f - 1$ wrong fact values.

- $V_v - N_f$ *empty* value entries, matching values that do not appear in the context.

- $V_k$ *key* entries.

For any vector entry $p$, using the one-hot basis vector $e_p$, we can express the entry scalar value using the following bilinear forms:

$$y_t^p = e_p^\top y_t = \sum_{\tau=0}^{t} \left(e_p^\top G_E x_\tau\right) \left(x_{\tau-1}^\top G_{\tilde{E}} k_m\right) \tag{56}$$

However, since $E, \tilde{E}$ are JL embeddings, we can state (not quantitatively for now; see Lemma 1 and Theorem 3):

$$e_p^\top G_E x_\tau \approx \begin{cases} 1 & x_\tau = e_p \\ 0 & x_\tau \neq e_p \end{cases}, \qquad x_{\tau-1}^\top G_{\tilde{E}} k_m \approx \begin{cases} 1 & x_{\tau-1} = k_m \\ 0 & x_{\tau-1} \neq k_m \end{cases} \tag{57}$$

Note that this approximation becomes more exact as the model dimensions grow. Since in MQAR the tokens do not repeat, we have both terms $\approx 1$ only if $(x_{\tau-1}, x_\tau) = (e_p, k_m)$. However, this can only be found in the single fact where $k_m$ appears; namely, $(x_{\tau-1}, x_\tau) = (v_i, k_m)$. This occurs only for this single specific entry $i$, while for all other entries, we have at least one of the terms $\approx 0$. Therefore:

$$y_t^p \approx \begin{cases} 1 & p = i \\ 0 & p \neq i \end{cases} \tag{58}$$

or, more simply, $y_t \approx v_i$.

That is to say, for small enough $\varepsilon_v$, $\varepsilon_k$, or equivalently for large enough $D$, $N$, the simplified linear Mamba model with weights as constructed above can achieve almost-perfect recall.

$\square$

## C MECHANISTIC INTERPRETABILITY OF THE RECALL CIRCUIT: SUPPLEMENTARY MATERIAL

### C.1 INVARIANT OPERATORS

$G_{kq}$ is a bilinear operator, operates on token pairs: $\xi_t = \left(\begin{smallmatrix} x_{t-1} \\ x_t \end{smallmatrix}\right)$ and $\xi_\tau = \left(\begin{smallmatrix} x_{\tau-1} \\ x_\tau \end{smallmatrix}\right)$. Therefore, it consists of 4 blocks, corresponding to the 4 combinations of $x_t$, $x_{t-1}$ parts and $x_\tau$, $x_{\tau-1}$ parts.

As predicted by theory and observed in Fig. 9, $\xi_\tau^\top G_{kq} \xi_t$ is nonzero only for the $(\tau-1, t)$ block. This corresponds to the mechanism of matching between a query $q_t = x_t$ and its corresponding key $k_t = x_{\tau-1}$. Note that $x_\tau$ and $x_{t-1}$ are completely ignored by $G_{kq}$, which makes sense: they are unrelated to the query-key matching procedure.

Similarly, $G_{vv}$ is thought as a matching between token pairs $\xi_\tau = \left(\begin{smallmatrix} x_{\tau-1} \\ x_\tau \end{smallmatrix}\right)$ and output tokens $y_t$. Hence, it consists of 2 blocks, corresponding the $x_\tau$, $x_{\tau-1}$ parts and the $y_t$ part.

With a fit to our theory, Fig. 9, shows that $y_t^\top G_{vv} \xi_t$ is nonzero only for the $(t, \tau)$ block. This corresponds to the mechanism of retrieving $y_t = x_\tau$, which is the correct value given a query-key match $x_t = x_{\tau-1}$. Also note that $G_{vv}$ ignores $x_{\tau-1}$ (the key token): only the value $x_\tau$ matters.

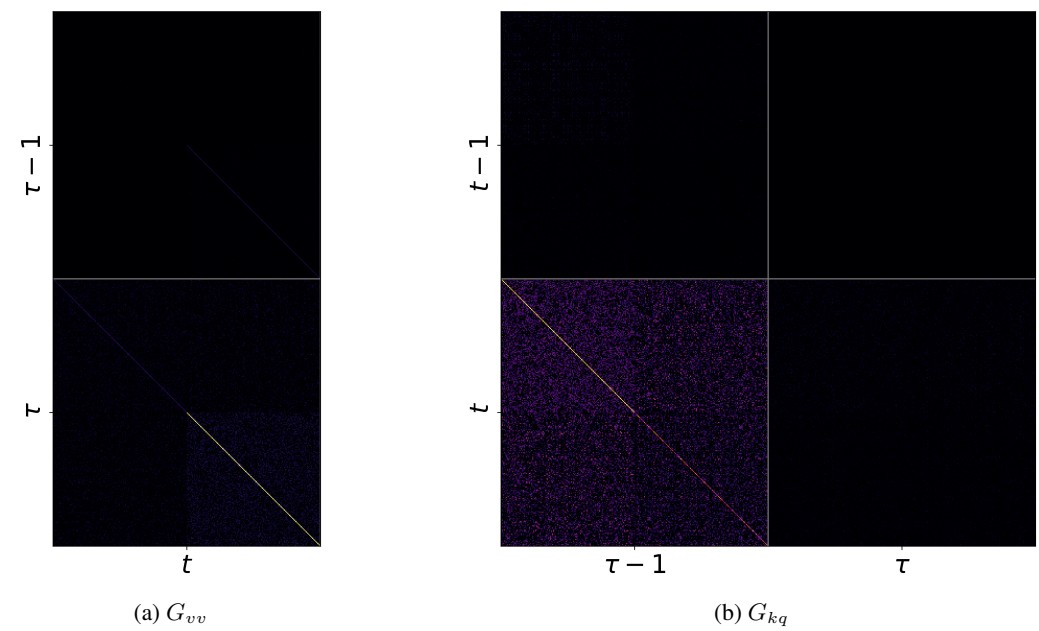

(a) $G_{vv}$        (b) $G_{kq}$

Figure 9: **Invariant operators.** The model's internal mechanism is revealed when applying the appropriate transformations. As predicted by Eq. 10, the core operation compares a query $q_t$ with a key $k_{\tau-1}$, then outputs the corresponding value $v_\tau$.

### C.2 SSM HIDDEN STATE AND KEY-VALUE TABLES

We shortly present and discuss the definitions for $H_t$, $H_t'$ and $H_t''$.

For clarity, let us denote the variables from Thm. 1 as *ideal*, and the variables from Thm. 2 as *noisy*.

Table 4: Dimensions for hidden state and key-value tables

| Symbol | Meaning | Dimension | Notes |
|---|---|---|---|
| $h_t^{\text{ideal}}$ | Hidden state, non-compressive circuit | $\mathbb{R}^{2V \times V}$ | As constructed in Thm. 1 |
| $h_t^{\text{noisy}}$ | Hidden state, compressive circuit | $\mathbb{R}^{2D \times N}$ | As constructed in Thm. 2 |
| $H_t$ | Projected hidden state, non-compressive circuit | $\mathbb{R}^{V \times V}$ | $= (0 \mid I_V)\, h_t^{\text{ideal}}$ |
| $H_t'$ | Projected hidden state, compressive circuit | $\mathbb{R}^{D \times N}$ | $= (0 \mid I_D)\, h_t^{\text{noisy}}$ |
| $H_t''$ | Decompressed hidden state, compressive circuit | $\mathbb{R}^{V \times V}$ | $= E^\top H_t' \tilde{E}$ |

For the two circuits, ideal and compressive, we now define:

$$H_t \equiv P_{\text{out}}\, h_t^{\text{ideal}} = \sum_{n=1}^{N_f} v_n k_n^\top \tag{59}$$

$$H_t' \equiv P_{\text{out}}\, h_t^{\text{noisy}} = \sum_{n=1}^{N_f} v_n' {k_n'}^\top \tag{60}$$

$$H_t'' \equiv E^\top H_t' \tilde{E} = \sum_{n=1}^{N_f} v_n'' {k_n''}^\top \tag{61}$$

where the *compressed* tokens are $v_n' = E\, v_n$, $k_n' = \tilde{E}\, k_n$, $q_t' = \tilde{E}\, q_t$, and the *decompressed* tokens are $v_n'' = E^\top E\, v_n \approx v_n$ and $k_n'' = \tilde{E}^\top \tilde{E}\, k_n \approx k_n$. (Note that $E \in \mathbb{R}^{D \times V}$ and $\tilde{E} \in \mathbb{R}^{N \times V}$ are JL matrices, thus achieving approximately perfect compression and decompression, given sufficiently large $D, N$.)

The model output for each of the circuits can be now written as:

$$y_t^{\text{ideal}} = H_t\, q_t = \sum_{n=1}^{N_f} v_n k_n^\top q_t \tag{62}$$

$$y_t^{\text{noisy}} = E^\top H_t'\, q_t' = E^\top \sum_{n=1}^{N_f} v_n' {k_n'}^\top q_t' \tag{63}$$

$$y_t^{\text{noisy}} = H_t''\, q_t = \sum_{n=1}^{N_f} v_n'' {k_n''}^\top q_t \tag{64}$$

where the latter two equations are equivalent.

Throughout the manuscript, the *projected hidden states* $H_t$, $H_t'$, $H_t''$ are sometimes referred to as *hidden states* as well. The reason is that for MQAR, the *informative* parts of $h_t^{\text{ideal}}$ and $h_t^{\text{noisy}}$ (which contain all key-value information) entirely lie in the $H_t$, $H_t'$ halves, respectively.

### C.3 INSPECTING THE HASH TABLE

The starting point of our analysis is the *non-rotated* recall circuit, as constructed in Theorem 2, where $P_{\text{in}}$, $P_{\text{out}}$, and $W_{\text{conv}}$ are carefully designed with identity and zero matrices and vectors. In this setup, $S_B$ and $S_C$ must have a structure $S_B = (F \mid 0)$ and $S_C = (0 \mid F)$ in order to solve MQAR. If we fix $P_{\text{in}}$, $P_{\text{out}}$, and $W_{\text{conv}}$ as in the construction, then indeed $F$ can be extracted from learned model weights, allows to compute $\tilde{E}$, then to compute Equation 13 and $H'' = E^\top H' \tilde{E}$.

However, in the general *rotated* case, where $P_{\text{in}}$, $P_{\text{out}}$, and $W_{\text{conv}}$ are not constrained, the operator $F$ does not exist as-is in any of the model weights: $S_B$ and $S_C$ do not have the simple structure described in Theorem 2, since the key and query compression operations $F$ are distributed across

a mixture of linear operators $S_B$, $S_C$, $P_{\text{in}}$, and $W_{\text{conv}}$. Therefore, we must use the net operations $\Pi_{k,\text{in}}$ and $\Pi_{q,\text{in}}$ (see Equation 8), which correctly combine all these linear transformations. Decompression via $k_n'' = \tilde{E}^\top \tilde{E} k_n$ is a special case, which only applies for the *non-rotated* circuit; its general *rotation-invariant* form is $k_n'' = G_{kq}^\top k_n$. Similarly, $H_t'' = E^\top H_t' \tilde{E}$ is a special *non-rotated* case; its general invariant form is $H_t'' = \Pi_{v,\text{out}} H_t' \Pi_{q,\text{in}}$, which is derived by replacing $E^\top$ and $\tilde{E}$ with their general equivalent operators from Equation 8.

# D    MAMBA RECALL SCALING LAWS: PROOFS

**Key-value selection mechanism.**    Let us consider the following insight. If a realistic, compressive model is able to perfectly ignore non-key-value $(x_{\tau-1}, x_\tau)$ pairs, then its recall abilities depends only on $N_f$ and not on context length $L$. Such non-key-value pairs include, i.e. value-key pairs between context facts, and key-key or value-value *random non-query* token pairs between queries.

Such selectivity is achievable if the learned embeddings $E_k$ and $E_v$ span orthogonal subspaces (which we denote shortly as $E_k \perp E_v$), and similarly for $\tilde{E}_k$ and $\tilde{E}_v$. Let us analyze it case by case, considering a query $x_t = k_m \in \mathcal{V}_k$.

- If $\tilde{E}_k \perp \tilde{E}_v$, then for all $(v_i, x_\tau)$ pairs in the context, we have $v_i^\top G_{\tilde{E}}\, k_m = 0$, which means these pairs are not written into the hidden state $h_t$. This implies all value-key and value-value pairs are ignored.
- Key-key pairs, unfortunately, are written into $h_t$. However, if $E_k \perp E_v$, then for a key-key pair $(k_l, k_n)$, given a value entry $p$ where $e_p \in \mathcal{V}_v$, we have $e_p^\top G_E\, k_n = 0$. This implies that even though written into $h_t$, this pair has zero contribution when retrieved, thus the "value" $k_n$ would not affect the output $y_t$.

The overall consequence of this selection mechanism is that only key-value pairs contribute to $y_t$, resulting in recall capacity determined only by $N_f$ and not $L$.

From a mechanistic interpretability perspective, Fig. 2 tells us that, at least empirically, the model indeed learns such orthogonal embeddings, which justifies our conclusion.

**Phase transitions.**    It may be briefly noted that, from a statistical physics point of view, Theorem 3 describes a phase transition behavior: for large $V$, the $(\Phi(\dots))^{V_v}$ term in Eq. 78 demonstrate a sharp, step-like transition. The resulting prediction is that, plotting MQAR accuracy in dimension space $D, N$, we expect to see a large domain with accuracy $\approx 1$ and a large domain with accuracy $\approx 0$, with a relatively small domain of transition between them.

**Lemma 1** (**JL-Based AR Scaling Law**). *Given a vocabulary size $V$, a single-layer simplified Mamba with dimensions $N < D < V$, $\mathrm{expand} = 2$, $D_{\mathrm{conv}} = 2$ can **perfectly solve** an MQAR task if model dimensions satisfy:*

$$\varepsilon_v < 1, \quad \varepsilon_k < 1, \quad \varepsilon_v + 2\,\varepsilon_k + 2N_f\,\varepsilon_v\,\varepsilon_k < 1 \tag{65}$$

*with $\varepsilon_v = \sqrt{\frac{c \log V}{D}}$, $\varepsilon_k = \sqrt{\frac{c \log V}{N}}$, $c = 4$.*

**Proof.** We use the lower and upper bounds derived from JL lemma to separate the correct value output entry from all other entries.

**JL bounds.**    Assume the model weights are constructed as described in Theorem 2. Given a query $x_t = q_t = k_m$, the model output vector $y_t$ has value entries given by: (since $v_p$ are one-hot vectors)

$$y_t^p = v_p^\top y_t = \sum_{\tau=0}^t \left(v_p^\top G_E\, x_\tau\right) \left(x_{\tau-1}^\top G_{\tilde{E}}\, k_m\right) \tag{66}$$

Since $E, \tilde{E}$ are both JL embeddings, we have:

$$v_p^\top G_E\, x_\tau \in \begin{cases} (1 - \varepsilon_v,\, 1 + \varepsilon_v) & x_\tau = v_p \\ (-\varepsilon_v,\, \varepsilon_v) & x_\tau \neq v_p \end{cases} \tag{67}$$

$$x_{\tau-1}^\top G_{\tilde{E}}\, k_m \in \begin{cases} (1 - \varepsilon_k,\, 1 + \varepsilon_k) & x_{\tau-1} = k_m \\ (-\varepsilon_k,\, \varepsilon_k) & x_{\tau-1} \neq k_m \end{cases} \tag{68}$$

For simplicity, let us first examine a single-query retrieval. We assume the context contains $L$ tokens previous to the query, of which $2N_f$ tokens are key-value pairs.

**Entry-wise analysis.**    We start with the output vector $y_t$ decomposition. For each entry $p$, we analyze the contribution of each token pair in the context to the sum in Eq. 66. For brevity, we denote

Table 5: Analysis of token pairs contribution per output entry. For each token pair, each cell value corresponds to match (✔), mismatch (✗) or zero contribution (0) with the first and second token $(x_{\tau-1}, x_\tau)$, respectively. Note that all non-fact pairs have zero contribution, due to key-value embeddings orthogonality. (For instance, $(v_a, v_b)$ pairs have no contribution since $v_a^\top G_{\tilde E} k_m = 0$, etc.)

| Token Pair | | **Fact** $(k, v)$ | | | | | | **Non-Fact** | | | | | |
| | | Correct $(k_m, v_i)$ | | Wrong $(k_n, v_j)$ | | Wrong $(k_a, v_b)$ | | $(k, k)$ | | $(v, v)$ | | $(v, k)$ | |
| **Entry** | Correct $i$ | ✔ | ✔ | ✗ | ✗ | ✗ | ✗ | - | 0 | 0 | - | 0 | 0 |
| | Wrong $j$ | ✔ | ✗ | ✗ | ✔ | ✗ | ✗ | - | 0 | 0 | - | 0 | 0 |
| | Empty $l$ | ✔ | ✗ | ✗ | ✗ | ✗ | ✗ | - | 0 | 0 | - | 0 | 0 |

$g_\tau = v_p^\top G_E x_\tau$ and $\tilde g_\tau = x_{\tau-1}^\top G_{\tilde E} k_m$. Given a value entry $p$, for each token pair $(x_{\tau-1}, x_\tau)$, we say we have a *value match* if $x_\tau = v_p$ (thus $g_\tau \approx 1$) and a *key match* if $x_{\tau-1} = k_m$ (such that $\tilde g_\tau \approx 1$). As discussed in D, we assume key and value embeddings are orthogonal.

Also see Table 5.

For the **correct** entry $i$, the total $L$ context token pairs are divided into:

- $N_f$ *fact* pairs:

    - 1 *correct* fact pair: for this pair, both key and value match, $(x_{\tau-1}, x_\tau) = (k_m, v_i)$. Hence, applying JL lower bound, $g_\tau \tilde g_\tau > (1 - \varepsilon_v)(1 - \varepsilon_k)$.

    - $N_f - 1$ *wrong* fact pairs: since key and values are unique (and $k_m, v_i$ have already appeared in the correct fact), for each of these pairs we have $x_{\tau-1} \neq k_m$ and $x_\tau \neq v_i$. Therefore, using JL bound, $g_\tau \tilde g_\tau > -\varepsilon_v \varepsilon_k$.

- $L - N_f$ *non-fact* pairs:

    - These are value-key or value-value pairs. Since value and key embeddings are orthogonal, and $x_{\tau-1}$ is a value, we have for these pairs $\tilde g_\tau = x_{\tau-1}^\top G_{\tilde E} k_m = 0$. Hence, they have no contribution to the sum.

Applying JL lemma lower bound, the contributions add up to:

$$y_t^i > (1 - \varepsilon_v)(1 - \varepsilon_k) - (N_f - 1)\varepsilon_v \varepsilon_k \approx 1 - \varepsilon_v - \varepsilon_k - N_f \varepsilon_v \varepsilon_k \tag{69}$$

For each of the $N_f - 1$ **wrong** entries $j$, we have:

- $N_f$ *fact* pairs:

    - 1 *correct* fact: for this fact, we have a key match and a value mismatch, since $x_{\tau-1} = k_m$ and $x_\tau = v_i \neq v_j$. Using JL upper bound, this fact contributes $g_\tau \tilde g_\tau < \varepsilon_v(1 + \varepsilon_k)$.

    - 1 *wrong* fact with $x_\tau = v_j$: we have a key mismatch, since $x_{\tau-1} = k_n \neq k_m$, and a value match, since $x_\tau = v_j$ match entry index $j$. Hence, applying JL bound, this fact contributes $g_\tau \tilde g_\tau < (1 + \varepsilon_v)\varepsilon_k$.
    - $N_f - 2$ *wrong* facts with $x_\tau \neq v_j$: as before, we have key mismatch, and now also a value mismatch, since $x_\tau \neq v_j$. These facts contribute $g_\tau \tilde g_\tau < \varepsilon_v \varepsilon_k$.

- $L - N_f$ *non-fact* pairs:

    - Same as above: all of these pairs have $\tilde g_\tau = 0$ due to key-value orthogonality, hence do not contribute to the sum.

Applying JL lemma upper bound, we have:

$$y_t^j < \varepsilon_v(1 + \varepsilon_k) + (1 + \varepsilon_v)\varepsilon_k + (N_f - 2)\varepsilon_v \varepsilon_k = \varepsilon_v + \varepsilon_k + N_f \varepsilon_v \varepsilon_k \tag{70}$$

For each of the rest $V_v - N_f$ **empty** value entries $l$, corresponding to values $v_l$ which do no appear in any context fact, we simply have:

- $N_f$ *fact* pairs:
    - 1 *correct* fact: as above, a key match and a value mismatch.
    - $N_f - 1$ *wrong* fact: a key mismatch and a value mismatch.
- $L - N_f$ *non-fact* pairs:
    - All have zero contribution due to orthogonality.

Applying JL lemma upper bound, we have:

$$y_t^l \; < \; \varepsilon_v(1 + \varepsilon_k) + (N_f - 1)\,\varepsilon_v\varepsilon_k = \varepsilon_v + N_f\,\varepsilon_v\varepsilon_k \tag{71}$$

**Perfect recall condition.** For perfect retrieval, we want the correct-value entry $i$ to be maximal:

$$y_t^i = \max_p \; y_t^p \tag{72}$$

Using our JL-based bounds, $y_t^i > y_t^{p \neq i}$ is achieved when

$$1 - \varepsilon_v - \varepsilon_k - N_f\varepsilon_v\varepsilon_k > \varepsilon_v + \varepsilon_k + N_f\,\varepsilon_v\varepsilon_k \tag{73}$$

or, simpler, if

$$\varepsilon_v + \varepsilon_k + N_f\varepsilon_k\varepsilon_v < \frac{1}{2} \tag{74}$$

**From JL bound to a scaling law.** We can now think of $\varepsilon_v$, $\varepsilon_k$ as of new, "natural" dimension variables, replacing the original dimensions as in a coordinate transformation:

$$\begin{aligned} \varepsilon_v &\equiv \sqrt{\frac{c \log V}{D}} \\ \varepsilon_k &\equiv \sqrt{\frac{c \log V}{N}} \end{aligned} \tag{75}$$

JL lemma generally requires $\varepsilon_v, \varepsilon_k < 1$, or equivalently, $D, N > c \log V$. To these requirements we add the specific MQAR condition given in equation 74, which can now be seen as a *scaling law*:

$$\sqrt{\frac{c \log V}{D}} + \sqrt{\frac{c \log V}{N}} + N_f \frac{c \log V}{\sqrt{ND}} < \frac{1}{2} \tag{76}$$

A simplified linear Mamba model, with weights set as in Theorem 2 and dimensions $D, N$ satisfying the above condition, is **guaranteed to achieve perfect recall** on MQAR.

$\square$

**Theorem 3** (**Probabilistic AR Scaling Law**). *A single-layer simplified Mamba can solve an MQAR task with a probability approximated by:*

$$p_{\text{success}} \approx \left( \Phi \left( \frac{1}{\sqrt{\frac{3}{D} + \frac{3}{N} + \frac{2N_f}{ND}}} \right) \right)^{N_f - 1} \left( \Phi \left( \frac{1}{\sqrt{\frac{3}{D} + \frac{2}{N} + \frac{2N_f}{ND}}} \right) \right)^{\frac{V}{2} - N_f} \tag{77}$$

*which, for $N_f >> N, D$, becomes*

$$p_{\text{success}} \approx \left( \Phi \left( \sqrt{\frac{2ND}{N_f}} \right) \right)^{\frac{V}{2}} \tag{78}$$

*with $\Phi$ the standard Normal CDF.*

***Proof.*** We view the distortion terms in Eq. 14 as random variables rather than tight bounds. We analyze their distribution, and, using CLT, derive a corresponding probability of success.

**From JL bounds to distributions.** As in Lemma 1, we want to evaluate the output vector entries:

$$y_t^p = v_p^\top y_t = \sum_{\tau=0}^{t} \left(v_p^\top E^\top E\, x_\tau\right)\left(x_{\tau-1}^\top \tilde{E}^\top \tilde{E}\, k_m\right) \tag{79}$$

Instead of the JL bound,

$$x_i^\top E^\top E\, x_j \;\in\; \begin{cases} (1-\varepsilon_v,\, 1+\varepsilon_v) & x_i = x_j \\ (-\varepsilon_v,\, \varepsilon_v) & x_i \neq x_j \end{cases} \tag{80}$$

let us now observe this value has a distribution (over choices of $x_i$, $x_j$, with $E$ fixed). Defining $Z_{ii}^d$ and $Z_{ij}^o$ random variables for diagonal and off-diagonal entries of $E^\top E - I$, we have:

$$x_i^\top (E^\top E - I)\, x_j \;=\; \begin{cases} Z_{ii}^d & x_i = x_j \\ Z_{ij}^o & x_i \neq x_j \end{cases} \tag{81}$$

and similarly for $\tilde{E}$, defining $\tilde{Z}^d$, $\tilde{Z}^o$:

$$x_i^\top (\tilde{E}^\top \tilde{E} - I)\, x_j \;=\; \begin{cases} \tilde{Z}_{ii}^d & x_i = x_j \\ \tilde{Z}_{ij}^o & x_i \neq x_j \end{cases} \tag{82}$$

Usually, for standard JL embedding constructions, if the columns of $E$, $\tilde{E}$ are further normalized (e.g. by their mean), we have all $Z^d$, $Z^o$, $\tilde{Z}^d$, $\tilde{Z}^o$ zero mean, for any $i, j$. Also, for such a standard JL matrices, it can be shown that the variances satisfy

$$\sigma_d^2 = \mathrm{Var}(Z^d) = \frac{2}{D}$$
$$\sigma_o^2 = \mathrm{Var}(Z^o) = \frac{1}{D}$$
$$\tilde{\sigma}_d^2 = \mathrm{Var}(\tilde{Z}^d) = \frac{2}{N} \tag{83}$$
$$\tilde{\sigma}_o^2 = \mathrm{Var}(\tilde{Z}^o) = \frac{1}{N}$$

**Entry-wise analysis.** Equipped with this perspective, the per-entry values become, for *correct i*, *wrong j* and *empty l* entries (see equations 69, 70, 71):

$$y_t^i = 1 + Z_{ii}^d + \tilde{Z}_{mm}^d + \sum_{p}^{N_f} Z_p^o \tilde{Z}_p^o \tag{84}$$

$$y_t^j = Z_{ji}^o + \tilde{Z}_{mn}^o + \sum_{p}^{N_f} Z_p^o \tilde{Z}_p^o \tag{85}$$

$$y_t^l = Z_{li}^o + \sum_{p}^{N_f} Z_p^o \tilde{Z}_p^o \tag{86}$$

However, for large enough $N_f$, using Central Limit Theorem, the sums distribute normally:

$$\mathrm{Var}(Z_p^o \tilde{Z}_p^o) = \sigma_o^2\, \tilde{\sigma}_o^2 \tag{87}$$

$$\sum_{p}^{L} Z_p^o \tilde{Z}_p^o \sim \mathcal{N}(0,\, N_f\, \sigma_o^2\, \tilde{\sigma}_o^2) \tag{88}$$

If we further approximate $Z$, $\tilde{Z}$ themselves to be normal, i.e. $Z_{ii}^d \sim \mathcal{N}(0, \sigma_d^2)$ and so on, we can finally get, approximately:

$$y_t^i \sim \mathcal{N}(1,\, \sigma_d^2 + \tilde{\sigma}_d^2 + N_f\, \sigma_o^2\, \tilde{\sigma}_o^2)$$
$$y_t^j \sim \mathcal{N}(0,\, \sigma_o^2 + \tilde{\sigma}_d^2 + N_f\, \sigma_o^2\, \tilde{\sigma}_o^2) \tag{89}$$
$$y_t^l \sim \mathcal{N}(0,\, \sigma_o^2 + N_f\, \sigma_o^2\, \tilde{\sigma}_o^2)$$

Then, the difference between the correct $i$ entry to the other entries are distributed as:

$$y_t^i - y_t^j \sim \mathcal{N}(1, \ \sigma_{wrong}^2)$$
$$y_t^i - y_t^l \sim \mathcal{N}(1, \ \sigma_{empty}^2)$$

(90)

with

$$\sigma_{wrong}^2 = \sigma_o^2 + \sigma_d^2 + \tilde{\sigma}_d^2 + \tilde{\sigma}_o^2 + 2\,N_f\,\sigma_o^2\,\tilde{\sigma}_o^2$$
$$\sigma_{empty}^2 = \sigma_o^2 + \sigma_d^2 + \tilde{\sigma}_o^2 + 2\,N_f\,\sigma_o^2\,\tilde{\sigma}_o^2$$

(91)

**Probability of success.** Now we are able to evaluate success probability on associative recall. The probability that $y_t^i > y_t^j$ for a single wrong entry $j$ is given by

$$p_w = \Phi\left(\frac{1}{\sigma_{wrong}}\right)$$

(92)

with $\Phi$ the standard Normal CDF. Similarly, the probability for $y_t^i > y_t^l$ is

$$p_n = \Phi\left(\frac{1}{\sigma_{empty}}\right)$$

(93)

Since $y_t$ has $(N_f - 1)$ wrong entries and $(V_v - N_f)$ empty coordinates, the overall success probability for a single query is:

$$p_{\text{success}} = p_w^{N_f - 1} p_n^{V_v - N_f}$$

(94)

Plugging Eq. 83 back in, we have

$$\sigma_{wrong}^2 = \frac{3}{D} + \frac{3}{N} + \frac{2N_f}{ND}$$

(95)

$$\sigma_{empty}^2 = \frac{3}{D} + \frac{2}{N} + \frac{2N_f}{ND}$$

(96)

Hence, the overall success probability is given by

$$p_{\text{success}} = \left(\Phi\left(\sqrt{\frac{1}{\frac{3}{D} + \frac{3}{N} + \frac{2N_f}{ND}}}\right)\right)^{N_f - 1} \left(\Phi\left(\sqrt{\frac{1}{\frac{3}{D} + \frac{2}{N} + \frac{2N_f}{ND}}}\right)\right)^{\frac{V}{2} - N_f}$$

(97)

where we assume a standard $V_v = \frac{V}{2}$. For $N_f >> N, D$, the previous result becomes:

$$p_{\text{success}} \approx \left(\Phi\left(\sqrt{\frac{2ND}{N_f}}\right)\right)^{\frac{V}{2}}$$

(98)

$\square$

# E    MULTI-LAYER AND MULTI-HEAD RECALL CIRCUITS: THEOREMS

In this section, we extend our results to the multi-layer and multi-head setting.

## E.1    MULTI-LAYER MAMBA: A RECALL CIRCUIT AND SCALING LAWS

Given the recall circuit in Thm. 2 and the scaling laws established in Thm. 3 and Eq. 17, it is natural to ask whether these findings extend to deeper architectures. Specifically, we ask what mechanism enables associative recall in a multi-layer Mamba model, and quantify its recall capacity. The following theorem aims to answer these question. Also see App. F for details.

**Theorem 4** (**Multi-layer recall circuit**). *Given a vocabulary size $V$, a simplified multi-layer Mamba with $\Lambda$ layers can solve MQAR if model dimensions are $D = O(\log V)$, $N = O(\frac{\log V}{\Lambda})$. (Full proof can be found in App. F.1.)*

**Multi-layer recall circuit.**    Let us briefly describe the mechanism. The residual connections between layers allow each Mamba layer $l$ to receive, as an input $x^l$, a linear combination of the original sequence $Ex$ and the current cumulative recall output $\sum_k^{l-1} y_{AR}^k$ as an input. Each layer then adds its own independent associative recall result $y_{AR}^l = Ex\,\alpha_{AR}^l$, which is then similarly passed through a residual connection to the next layer, such that model output is $y = \sum_k^{\Lambda} y_{AR}^k$.

**Remark 2** (**AR scaling law for multi-layer Mamba**). Combining the results of Thm. 3 and Thm. 4, we conclude: a *multi-layer linear Mamba* with $\Lambda$ layers can solve an MQAR task with a probability approximated by $p_{\text{success}} \approx \left( \Phi \left( \sqrt{\frac{2\Lambda ND}{N_f}} \right) \right)^{\frac{V}{2}}$.

We further validate the scaling law results through experiments in Sec. 5.2, and found it to be accurate and predictive.

## E.2    MAMBA-2: MULTI-HEAD RECALL CIRCUITS

Among other architectural modifications, a significant concept introduced in Mamba-2 (Dao & Gu, 2024) over Mamba is its ability to process the input sequence through multiple SSM heads in parallel. Equivalently to parameter sharing across attention heads in Transformers, the authors present several ***head patterns*** for multi-head SSM (see Tab. 8, Appendix). In each head pattern, a different combination of $\hat{x}$, $B$ and $C$ is shared across SSM heads, correspondingly to sharing $V$, $K$ and $Q$ between attention heads in a Transformer. Specifically, MVA shares $B, C$ between SSM heads, MKA shares $C$, MQA shares $B$, and MHA shares none.

**Multi-head recall circuits.**    Upon defining the multi-head SSM, one may ask whether it is beneficial in terms of associative recall capabilities, and if so, which head pattern should be chosen. In Alg. 3 (Appendix), we present a simplified Mamba-2 model, which follows the Mamba-2 design but keeps critical components only. In particular, it allows examination of the four head patterns.

Let us now analyze the associative recall capacity of each head pattern.

**Theorem 5** (Multi-head recall circuits). *Given fixed model dimension $D$, $N$, a single-layer simplified Mamba-2 model with $M$ heads, using MVA, MKA or MQA head patterns, is equivalent, in term of recall capacity, to a single-head model of same dimensions $D' = D$, $N' = N$. If MHA pattern is used, then the model is equivalent to a single-head model with $D' = D$, $N' = MN$, thus achieving an improved recall capacity. (See App. F.2.2 for proofs.)*

A more detailed discussion can be found in Sec. 5.2, where we experimentally verify these theoretical results.

# F   MULTI-LAYER AND MULTI-HEAD RECALL CIRCUITS: PROOFS

## F.1   MULTI-LAYER MAMBA: A RECALL CIRCUIT AND SCALING LAWS

In favor of the analysis of a multi-layer Mamba model, we present the following simplified model. As in Alg. 1, we assume gating, discretization, nonlinearities, biases and normalization layers are all removed. Note, however, the insertion of residual connections, as they are important for multi-layer recall, as we soon observe.

---

**Algorithm 2:** Simplified Multi-Layer Mamba

**Input:** $x \in \mathbb{R}^{V \times L}$ ,   **Output:** $y \in \mathbb{R}^{V \times L}$

Parameters: $E$, $\{P_{\text{in}}^l,\ P_{\text{out}}^l,\ W_{\text{conv}}^l,\ S_B^l,\ S_C^l \mid l = 1, \ldots, \Lambda\}$

$x_{\text{e}} \leftarrow E\, x$
$x^1 \leftarrow x_{\text{e}}$
**for** $l \leftarrow 1$ **to** $\Lambda$ **do**

    **Mixer:**

        $x_{\text{in}}^l \leftarrow P_{\text{in}}^l\, x^l$
        $\hat{x}^l \leftarrow \text{Conv1D}(x_{\text{in}}^l, W_{\text{conv}}^l)$
        $B^l,\ C^l \leftarrow S_B^l\, \hat{x}^l,\ S_C^l\, \hat{x}^l$
        $\hat{y}^l \leftarrow \text{SSM}(\hat{x}^l, B^l, C^l)$
        $y_{\text{out}}^l \leftarrow P_{\text{out}}^l\, \hat{y}^l$

    $y^l \leftarrow x^l + y_{\text{out}}^l$
    $x^{l+1} \leftarrow y^l$

$y \leftarrow E^\top\, y^\Lambda$
**return** $y$

---

**Theorem 6** (**Multi-layer recall circuit**). *Given a vocabulary size $V$, a simplified multi-layer Mamba with $\Lambda$ layers and dimensions $N < D < V$, $\text{expand} = 2$, $D_{\text{conv}} = 2$ can solve an MQAR task with the same probability as a single-layer model with the same dimensions and state size $N^{\text{eff}} = \Lambda N$.*

*In particular, the model can solve MQAR if its dimensions are $D = O(\log V)$, $N = O(\frac{\log V}{\Lambda})$.*

*Proof.* Let us construct the multi-layer Mamba solution to MQAR step by step.

**Step 1: Single-layer Mamba**

First, we show a single-layer Mamba with residual connection can solve MQAR. We slightly modify the weights construction of Theorem 2, such that residual connection does not affect the recall circuit output.

**Weights.**   Given embedding matrices $E \in \mathbb{R}^{D \times V}$ and $F^1 \in \mathbb{R}^{N \times D}$, and given a parameter $a^1 \in \mathbb{R}$, we choose:

$$P_{\text{in}}^1 = \begin{pmatrix} I_D \\ I_D \end{pmatrix}, \quad P_{\text{out}}^1 = \begin{pmatrix} 0 \mid I_D \end{pmatrix},$$
$$W_{\text{conv}}^1 = \begin{pmatrix} 1_D \mid 0_D \\ 0_D \mid 1_D \end{pmatrix}, \quad S_B^1 = \begin{pmatrix} F^1 \mid a^1 F^1 \end{pmatrix}, \quad S_C^1 = \begin{pmatrix} 0 \mid F^1 \end{pmatrix} \tag{99}$$

As before, let us define $\tilde{E}^1 \equiv F^1 E \in \mathbb{R}^{N \times V}$. Inside the SSM, we now have $\hat{x}^1 \in \mathbb{R}^{D_{\text{in}} \times L}$ such that $\hat{x}_t^1 = \begin{pmatrix} E\, x_{t-1} \\ E\, x_t \end{pmatrix}$, $C_t^1 = \tilde{E}^1 x_t$, and most importantly $B_t^1 = \tilde{E}^1 x_{t-1} + a^1 \tilde{E}^1 x_t$.

**Attention view.**   We now leverage *attention* formulation to view SSM operation. Namely, the SSM attention matrix $\alpha^1 \in \mathbb{R}^{L \times L}$ should satisfy $\hat{y}_t^1 = \sum_{\tau=0}^t \hat{x}_\tau^1 \alpha_{\tau t}^1$, or in matrix form $\hat{y}^1 = \hat{x}^1 \alpha^1 \in \mathbb{R}^{D_{\text{in}} \times L}$. (Note the right-side matrix multiplication, which operates along time axis rather than dimension axis). Following Ali et al. (2024); Dao & Gu (2024), the attention matrix for our simplified SSM is:

$$\alpha_{\tau t}^1 = B_\tau^{1 \top} C_t = (\tilde{E}^1 x_{\tau-1} + a^1 \tilde{E}^1 x_\tau)^\top (\tilde{E}^1 x_t) \tag{100}$$

As in Theorem 2, we choose $E, F^1$ such that $\tilde{E}^1 \in \mathbb{R}^{N \times V}$ is a JL embedding matrix w.r.t some small $\varepsilon_k > 0$. Defining $G^1 \equiv (\tilde{E}^1)^\top \tilde{E}^1$ for short, we have:

$$\alpha_{\tau t}^1 = x_{\tau-1}^\top G^1 x_t + a^1 x_\tau^\top G^1 x_t \tag{101}$$

Alternatively, minding that for a JL matrix, $G^1 = I_V + O(\varepsilon_k)$,

$$\alpha^1_{\tau t} = x^\top_{\tau-1} G^1 x_t + a^1 x^\top_\tau x_t + O(\varepsilon_k) \tag{102}$$

which yields the $L \times L$ matrix form

$$\alpha^1 = \alpha^1_{AR} + a^1 I_L + O(\varepsilon_k) \tag{103}$$

where $(\alpha^1_{AR})_{\tau t} = x^\top_{\tau-1} G^1 x_t$ is the compressive-recall attention matrix equivalent of Theorem 1 (which performs recall by matching MQAR queries with keys), and $I_L \in \mathbb{R}^{L \times L}$ is the identity matrix in sequence space. We denote $\alpha^1_{AR}$ this way to avoid confusion with the overall SSM attention matrix $\alpha^1$.

**Layer output.** Given the input sequence $x^1 = Ex \in \mathbb{R}^{D \times L}$, the SSM output is:

$$\hat{y}^1 = \hat{x}^1 \alpha^1 = P^1_{\text{in}} Ex(\alpha^1_{AR} + a^1 I_L) = P^1_{\text{in}} Ex\alpha^1_{AR} + a^1 P^1_{\text{in}} Ex \tag{104}$$

When projected out, since $P^1_{\text{in}} = \begin{pmatrix} I_D \\ I_D \end{pmatrix}, P^1_{\text{out}} = \begin{pmatrix} 0 \mid I_D \end{pmatrix}$, we have:

$$y^1_{\text{out}} = P^1_{\text{out}} \hat{y}^1 = \hat{x}^1 \alpha^1 = Ex\alpha^1_{AR} + a^1 Ex \tag{105}$$

The residual connection now implies:

$$y^1 = x^1 + y^1_{\text{out}} = Ex\alpha^1_{AR} + (a^1 + 1)Ex \tag{106}$$

We are now able to choose $a^1 = -1$, such that

$$y^1 = Ex\alpha^1_{AR} \tag{107}$$

and model output is

$$y = E^\top y^1 = E^\top Ex\alpha^1_{AR} \tag{108}$$

which is, by design, the same solution to MQAR as in Theorem 2.

**Step 2: Two-layer Mamba**

Building upon previous step, we now describe a two-layer Mamba solution to MQAR.

Assume a *simplified multi-layer Mamba* with $\Lambda = 2$. We set first layer ($l = 1$) weights as in Step 1, but this time without specifying $a^1 = -1$.

**Layer input.** Importantly, the second layer ($l = 2$) input is:

$$x^2 = y^1 = x^1 + y^1_{\text{out}} = Ex\alpha^1_{AR} + (a^1 + 1)Ex \tag{109}$$

For clarity, let us denote $y^1_{AR} = Ex\alpha^1_{AR}$ and $b = a^1 + 1$. Layer input now reads:

$$x^2 = y^1_{AR} + bEx \tag{110}$$

**Weights.** Given a matrix $F^2 \in \mathbb{R}^{N \times D}$ and a parameter $a^2 \in \mathbb{R}$, let us set layer weights as follows:

$$P^2_{\text{in}} = \begin{pmatrix} I_D \\ I_D \end{pmatrix}, \quad P^2_{\text{out}} = \begin{pmatrix} 0 \mid I_D \end{pmatrix},$$
$$W^2_{\text{conv}} = \begin{pmatrix} 1_D \mid 0_D \\ 0_D \mid 1_D \end{pmatrix}, \quad S^2_B = \frac{1}{b}\begin{pmatrix} F^2 \mid a^2 F^2 \end{pmatrix}, \quad S^2_C = \frac{1}{b}\begin{pmatrix} 0 \mid F^2 \end{pmatrix} \tag{111}$$

**Key and value subspaces.** Inside layer 2 SSM, after input projections, we have $\hat{x}^2_t = \begin{pmatrix} bE\,x_{t-1} + (y^1_{AR})_{t-1} \\ bE\,x_t + (y^1_{AR})_t \end{pmatrix}$. Importantly, note that $\mathbb{R}^D$ can be decomposed into two orthogonal key and value subspaces, namely $\mathbb{R}^D = U \oplus W$, such that $E = E_k + E_v$ and $(E_k)_i \perp (E_v)_j$ for any $i, j$. Consequently, we can choose $F^2$ which acts only on embedded key vectors $Ek_i \in U$ and zeros out embedded value vectors $Ev_j \in W$: for any $j$, $F^2 Ev_j = 0$.

Notice that, as a result of Theorem 2, any $(y^1_{AR})_t$ is approximately a *value* vector, up to a small $O(\varepsilon_k + \varepsilon_v)$ error. (This can be further guaranteed by modifying layer 1 output projection $P^1_{\text{out}}$ to zero out the key subspace $U$.) This implies $F(y^1_{AR})_t \approx 0$, for any $t$. Then, upon SSM projections,

we have, approximately, $C_t^2 = \tilde{E}^2 x_t$ and $B_t^2 = \tilde{E}^2 x_{t-1} + a^2 \tilde{E}^2 x_t$, without any $y_{AR}^1$ terms, where we have defined $\tilde{E}^2 = F^2 E$.

**Attention view.** Replicating the attention matrix computation from Step 1, and similarly defining $G^2 \equiv (\tilde{E}^2)^\top \tilde{E}^2$, we now have for layer 2 SSM:

$$\alpha^2 = \alpha_{AR}^2 + a^2 I_L + O(\varepsilon_k) \tag{112}$$

where $(\alpha_{AR}^2)_{\tau t} = x_{\tau-1}^\top G^2 x_t$ is a compressive-recall attention matrix. Note that $G^2 \neq G^1$ in general, since generally $F^2 \neq F^1$.

**Layer output.** Following Step 1, we compute the the SSM output:

$$\hat{y}_{\text{out}}^2 = P_{\text{out}}^2 \hat{y}^2 = P_{\text{out}}^2 \hat{x}^2 \alpha^2 = P_{\text{out}}^2 P_{\text{in}}^2 x^2 (\alpha_{AR}^2 + a^2 I_L) \tag{113}$$

But, since projections are trivial:

$$y_{\text{out}}^2 = x^2 (\alpha_{AR}^2 + a^1 I_L) \tag{114}$$

Considering residual connection, the layer output is:

$$y^2 = x^2 + y_{\text{out}}^2 = x^2 (\alpha_{AR}^2 + b^2 I_L) \tag{115}$$

where $b^2 = a^2 + 1$. Recall that $x^2 = y_{AR}^1 + b^1 E x$. Therefore,

$$y^2 = (y_{AR}^1 + b^1 E x)(\alpha_{AR}^2 + b^2 I_L) \tag{116}$$

We denote $y_{AR}^2 = E x \alpha_{AR}^2$, since it is a compressive-recall output, similarly to these in Step 1 and in Theorem 2. This yields:

$$y^2 = y_{AR}^1 \alpha_{AR}^2 + b^1 y_{AR}^2 + b^2 y_{AR}^1 + b^1 b^2 E x \tag{117}$$

Note that we have additional degrees of freedom by choosing the scales of $b^1, b^2$ and the norms of $E_k, E_v, F^1, F^2$, w.r.t to some arbitrary scale factor $\varepsilon$. This can be done such that:

$$y^2 = b^1 y_{AR}^2 + b^2 y_{AR}^1 + O(\varepsilon) \tag{118}$$

This, of course, does not mean the $O(\varepsilon)$ terms are negligible, but rather they have weaker contribution to model output, hance are less probable to change the maximal entry of retrieved value.

**Effective state size.** Importantly, both $y_{AR}^1$ and $y_{AR}^2$ are linear mixtures of the input sequence $E x$, namely, $y_{AR}^l = E x \alpha_{AR}^l$. Let us absorb the constants $b^1, b^2$ into $\alpha_{AR}^2, \alpha_{AR}^1$, and omit the error term:

$$y^2 \approx E x (\alpha_{AR}^1 + \alpha_{AR}^2) \equiv E x \, \alpha^{\text{eff}} \tag{119}$$

Moreover, the layers attention matrices are of a similar form, $(\alpha_{AR}^l)_{\tau t} = x_{\tau-1}^\top G^l x_t$. Hence:

$$\alpha_{\tau t}^{\text{eff}} = (\alpha_{AR}^1 + \alpha_{AR}^2)_{\tau t} = x_{\tau-1}^\top (G^1 + G^2) x_t \tag{120}$$

Since $G^l = \tilde{E}^{l^\top} \tilde{E}^l = E^\top F^{l^\top} F^l E$, we conclude:

$$\alpha_{\tau t}^{\text{eff}} = x_{\tau-1}^\top E^\top (F^{1^\top} F^1 + F^{2^\top} F^2) E \, x_t \tag{121}$$

This motivates us to define an effective matrix,

$$F^{\text{eff}} \equiv \begin{pmatrix} F^1 \\ F^2 \end{pmatrix} \in \mathbb{R}^{2N \times D} \tag{122}$$

and effective net embedding:

$$\tilde{E}^{\text{eff}} \equiv F^{\text{eff}} E \in \mathbb{R}^{2N \times V} \tag{123}$$

Finally, the model output now becomes:

$$y_t = E^\top y_t^2 = \sum_{\tau=0}^{t} E^\top (E \, x_\tau) (\tilde{E}^{\text{eff}} x_{\tau-1})^\top (\tilde{E}^{\text{eff}} x_t) \tag{124}$$

Hence, our multi-layer model with $\Lambda = 2$ is mathematically equivalent to a single-layer model with larger state size, $N^{\text{eff}} = 2N = \Lambda N$.

**Step 3: Multi-layer Mamba**

Statement is trivially proven by induction. We use $\Lambda = 1$ (Step 1) as a *base step*, and a similar claim as done for $\Lambda = 2$ (Step 2) as and *induction step*.

More intuitively: Step 2 shows that a Mamba layer $l$ which receives a linear combination $x^l = \sum_k^{l-1} a^k y_{AR}^k + b^l Ex$ as an input, can output a similar (independent) associative-recall result $y_{AR}^l = Ex \, \alpha_{AR}^l$, which is then similarly passed through a residual connection to the next layer, as $x^{l+1} = y_{AR}^l + \sum_k^{l-1} a^k y_{AR}^k + b^l Ex$.

Conclusion is, a simplified multi-layer Mamba model with $\Lambda$ layers can solve MQAR equivalently to a single-layer model with a larger state size $N^{\text{eff}} = \Lambda N$. $\qquad\square$

## F.2 MAMBA-2: MULTI-HEAD RECALL CIRCUITS

### F.2.1 NOTATION AND DIMENSIONS

We present here the notation and dimensions to be used throughout the analysis of associative recall in Mamba-2. Several differences from Mamba (see Table 3) are worthy of explanation.

Firstly, note that we begin already with a simplified model, hence omit state update matrix $A$, which is assumed an identity operation.

Secondly, mind the different pattern of projections and convolutions used in Mamba-2 to produce SSM sequences $\hat{x}$, $B$ and $C$: no expansion is used for input projection (thus $D_{\text{in}}$ is not used), but rather three different projections and convolution kernels, one per $\hat{x}, B, C$. (In implementation, these are usually fused into larger, unified projection and kernel; here we separate notation for clarity.)

Table 6: Notation for Mamba-2

| Symbol | Meaning | Value | Notes |
|--------|---------|-------|-------|
| $D$ | Embedding size | = Number of SSM channels | |
| $N$ | SSM State size | - | |
| $D_{\text{conv}}$ | Convolution kernel size | - | |
| $\Lambda$ | Number of layers | - | |
| $M$ | Number of SSM heads | - | |
| $P$ | SSM head size | $\frac{D}{M}$ | |
| $p$ | Head pattern | MHA, MKA, MQA or MVA | |

Table 7: Dimensions for Mamba-2

| Symbol | Meaning | Dimension | Notes |
|--------|---------|-----------|-------|
| $x$ | Model input sequence (one-hot) | $\mathbb{R}^{V \times L}$ | $x_t \in \mathbb{R}^V$ |
| $y$ | Model output sequence (one-hot) | $\mathbb{R}^{V \times L}$ | $y_t \in \mathbb{R}^V$ |
| $x^e$ | Model input sequence (embedded) | $\mathbb{R}^{D \times L}$ | $x_t^e \in \mathbb{R}^D$ |
| $\hat{x}$ | SSM input sequence | $\mathbb{R}^{D \times L}$ | $\hat{x}_t \in \mathbb{R}^D$ |
| $\hat{y}$ | SSM output sequence | $\mathbb{R}^{D \times L}$ | $\hat{y}_t \in \mathbb{R}^D$ |
| $\hat{z}$ | Model gate sequence | $\mathbb{R}^{D \times L}$ | $\hat{z}_t \in \mathbb{R}^D$ |
| $E$ | Model embedding | $\mathbb{R}^{D \times V}$ | |
| $P_{\text{in}}^{\hat{x},B,C}$ | Input projections | $\mathbb{R}^{D \times D}$ | Per $\hat{x}, B, C$ |
| $W_{\text{conv}}^{\hat{x},B,C}$ | Convolution kernels | $\mathbb{R}^{D \times D_{\text{conv}}}$ | Per $\hat{x}, B, C$ |
| $S_{B,C}$ | $B$, $C$ projections | $\mathbb{R}^{N^* \times D}$ | $N^*$ depends on head pattern |
| $P_{\text{out}}$ | Output projection | $\mathbb{R}^{D \times D}$ | |
| $B$ | SSM input coefficient | $\mathbb{R}^{N \times L}$ | $B_t \in \mathbb{R}^N$ |
| $C$ | SSM readout coefficient | $\mathbb{R}^{N \times L}$ | $C_t \in \mathbb{R}^N$ |

### F.2.2 MAMBA-2 AND MULTI-HEAD PATTERNS

One of the major concepts introduced in Mamba-2 (Dao & Gu, 2024) over Mamba is the ability to process the input sequence through multiple SSM heads in parallel. Equivalently to parameter sharing across attention heads in Transformers, the authors present several ***head patterns*** for multi-head SSM (see Tab. 8). In each head pattern, a different combination of $\hat{x}$, $B$ and $C$ is shared across SSM heads, correspondingly to sharing $V$, $K$ and $Q$ between attention heads in a Transformer.

It is assumed that the number of heads $M$ divides model dimension $D$, such that head dimension is $P = \frac{D}{M}$. Also note that as done in Eq. 3 and Alg. 1, we assume SSM matrix $A$ is an identity operation, hence omit it.

Table 8: Multi-Head Patterns in Mamba-2

| | **Multi-Head SSM** Multi-Head Attn. | **Multi-Contract SSM** Multi-Query Attn. | **Multi-Expand SSM** Multi-Key Attn. | **Multi-Input SSM** Multi-Value Attn. |
|---|---|---|---|---|
| $\hat{x}$ | $(L, M, P)$ | $(L, 1, P)$ | $(L, 1, P)$ | $(L, M, P)$ |
| $B$ | $(L, M, N)$ | $(L, 1, N)$ | $(L, M, N)$ | $(L, 1, N)$ |
| $C$ | $(L, M, N)$ | $(L, M, N)$ | $(L, 1, N)$ | $(L, 1, N)$ |

Importantly, the head pattern actually *used* in Mamba-2 in practice is MVA. Here, however, we study all patterns, in order to better understand recall capacity in Mamba-2.

Following these head patterns, we now generalize our simplified model to match Mamba-2 multi-head SSM patterns. Also notice the different structure of projections and convolutions (compared to Mamba) used to produce $\hat{x}$, $B$ and $C$ from input sequence $Ex$. See Table 6 and Table 7 for notation and dimensions.

---

**Algorithm 3:** Simplified Single-Layer Mamba-2

**Input:** $x$ $(L, V)$, **Output:** $y$ $(L, V)$

Parameters: $E$, $P_{\text{in}}$, $P_{\text{out}}$, $W_{\text{conv}}$, $W_x$, $W_y$

Head Pattern: $p \in \{\text{MHA, MQA, MKA, MVA}\}$

$x_{\text{e}} \leftarrow E\,x$ $(L, D)$

**Mixer:**

$\hat{x} \leftarrow \text{Conv1D}(P_{\text{in}}^{\hat{x}}\, x_{\text{e}}, W_{\text{conv}}^{\hat{x}})$

$B, C \leftarrow S_{B,C}\, \text{Conv1D}(P_{\text{in}}^{B,C}\, x_{\text{e}}, W_{\text{conv}}^{B,C})$

$\hat{x} :$ $(L, D)$

$B : \begin{cases} (L, N) & \text{if } p \in \{\text{MQA, MVA}\} \\ (L, N, M) & \text{if } p \in \{\text{MHA, MKA}\} \end{cases}$

$C : \begin{cases} (L, N) & \text{if } p \in \{\text{MKA, MVA}\} \\ (L, N, M) & \text{if } p \in \{\text{MHA, MQA}\} \end{cases}$

$\hat{x}_{\text{heads}} \leftarrow \text{split}(\hat{x}, M)$ $(L, P, M)$

$\hat{x}_{\text{shared}} \leftarrow \text{contract}(\hat{x}_{\text{heads}}, W_x)$ $(L, P)$

**for** $h \leftarrow 1$ **to** $M$ **do**

$\quad \hat{x}^h \leftarrow \begin{cases} \hat{x}_{\text{heads}}^h & \text{if } p \in \{\text{MHA, MVA}\} \\ \hat{x}_{\text{shared}} & \text{if } p \in \{\text{MQA, MKA}\} \end{cases}$ $(L, P)$

$\quad \hat{y}^h \leftarrow \text{SSM}(\hat{x}^h, B^h, C^h)$ $(L, P)$

$\hat{y} \leftarrow \text{aggregate}(\{\hat{y}^h\}, W_y)$ $(L, D)$

$y_{\text{out}} \leftarrow P_{\text{out}}\, \hat{y}$ $(L, D)$

$y \leftarrow E^{\top}\, y_{\text{out}}$ $(L, V)$

**return** $y$

---

It should be noted that *single-layer simplified Mamba* is roughly a single-headed redundant case of Alg. 3, with $M = 1$, $P = D_{\text{in}}$, and $p = \text{MVA}$.

**Multi-head recall circuits.** Upon defining the multi-head SSM, one may ask whether it is beneficial in terms of associative recall capabilities, and if so, which head pattern should be chosen. Equipped with the simplified Mamba-2 model, let us now analyze the associative recall capacity of each head pattern.

**Lemma 2** (**MVA recall circuit**). *A single-layer simplified Mamba-2 model with $M$ heads using MVA pattern and dimensions $D, N = O(\log V)$ (with* $\text{expand} = 2$, $D_{\text{conv}} = 2$, *as in Thm. 2) is* **equivalent***, in term of associative recall, to a similar single-headed model of the same dimensions $D' = D$, $N' = N$.*

*Namely, for MVA pattern, splitting into heads does not improve recall performance.*

*Proof.* We use the compressed recall weights as in Theorem 2, with slight adaptations to Mamba-2. Given a matrix $F \in \mathbb{R}^{N \times D}$, we choose weights $W_{\text{conv}}^{\hat{x}} = \begin{pmatrix} 0_D & | & 1_D \end{pmatrix}$, $W_{\text{conv}}^B = \begin{pmatrix} 1_D & | & 0_D \end{pmatrix}$,

$W_{\text{conv}}^C = \begin{pmatrix} 0_D \mid 1_D \end{pmatrix}$ and $S_B = S_C = F$, such that values are $\hat{x}_t = E\, x_t$, keys are $B_t = FE\, x_{t-1}$ and queries are $C_t = FE\, x_t$. The value $\hat{x}_t$ is then split into $M$ heads $\hat{x}_t^h$:

$$\hat{x}_t = \begin{pmatrix} \hat{x}_t^1 \\ \vdots \\ \hat{x}_t^M \end{pmatrix} \tag{125}$$

In each SSM head, we have:

$$\hat{y}_t^h = \sum_\tau^t \hat{x}_\tau^h B_\tau^\top C_t \tag{126}$$

Assume, for simplicity, heads aggregation has trivial weights $W_y^h = 1$ (see Alg. 3); the SSM output is thus $\{\hat{y}_t^h\}$ stacked:

$$\hat{y}_t = \begin{pmatrix} \hat{y}_t^1 \\ \vdots \\ \hat{y}_t^M \end{pmatrix} = \begin{pmatrix} \sum_\tau^t \hat{x}_\tau^1 B_\tau^\top C_t \\ \vdots \\ \sum_\tau^t \hat{x}_\tau^M B_\tau^\top C_t \end{pmatrix} = \sum_\tau^t \begin{pmatrix} \hat{x}_\tau^1 \\ \vdots \\ \hat{x}_\tau^M \end{pmatrix} B_\tau^\top C_t = \sum_\tau^t \hat{x}_\tau B_\tau^\top C_t \tag{127}$$

which is exactly the output we have for a single-headed model of the same dimensions $D, N$.

$\square$

**Lemma 3** (**MQA recall circuit**). *A single-layer simplified Mamba-2 model with $M$ heads using MQA pattern and dimensions $D, N = O(\log V)$ is **equivalent**, in term of associative recall, to a similar single-headed model of the same dimensions $D' = D,\, N' = N$.*

*Namely, for MQA pattern, splitting into heads does not improve recall performance, despite using a larger query dimension MN.*

**Proof sketch**. We choose convolution kernels as done in Lem. F.2.2, and fix $S_B = F$ for $F \in \mathbb{R}^{N \times D}$. We now have $M$ distinct projections $S_C^h = F^h \in \mathbb{R}^{N \times D}$, for generally different matrices $\{F^h\}$. As before, we have values $\hat{x}_t = Ex_t$, shared keys $B_t = FE\, x_{t-1}$, but now distinct queries $C_t^h = F^h E\, x_t$.

**Attention view.** Recall that each SSM head can be equivalently thought as of an attention head (see i.e. Ali et al. (2024); Dao & Gu (2024)), with attention matrix $\alpha_{\tau t}^h = {B_\tau^h}^\top C_t^h$. In our case:

$$\alpha_{\tau t}^h = B_\tau^\top C_t^h = x_{\tau-1}^\top E^\top F^\top F^h E\, x_t \tag{128}$$

The ideal attention matrix for associative recall (see Thm. 1) is $\alpha_{\tau t}^h = x_{\tau-1}^\top x_t$, which matches queries to keys. As in Thm. 2, this attention matrix can be approximated using JL matrices for compression. In our case, we should require $E^\top F^\top F^h E \approx I_V$. This can be achieved if $\tilde{E} \equiv FE$ is a JL matrix and also $F^h = F$ for all heads $h$. Only this way, we can achieve $\alpha_{\tau t}^h \approx x_{\tau-1}^\top x_t$. Note, however, this implies our queries are not independent anymore, as now $C_t^h = FEx_t$ are in fact shared between all heads; the additional weights become redundant. This means the recall circuit is equivalent to the one in Lem. F.2.2, thus performs equivalently to a single-headed model with $N' = N$.

$\square$

**Lemma 4** (**MKA recall circuit**). *A single-layer simplified Mamba-2 model with $M$ heads using MKA pattern and dimensions $D, N = O(\log V)$ is **equivalent**, in term of associative recall, to a similar single-headed model of the same dimensions $D' = D,\, N' = N$.*

*Namely, for MKA pattern, splitting into heads does not improve recall performance, despite using a larger key dimension MN.*

**Proof sketch**. Similarly to Lem. 3, where now $B_\tau^h = F^h Ex_{\tau-1}$ are distinct and $C_t = FEx_t$ is shared. Same constraint, forcing $F^h = F$, applies here, in order to achieve recall. $\square$

**Lemma 5** (**MHA recall circuit**). *A single-layer simplified Mamba-2 model with $M$ heads using MHA pattern and dimensions $D, N = O(\log V)$ is **equivalent**, in term of associative recall, to a similar single-headed model of dimensions $D' = D$, $N' = MN$. This implies an improved recall performance.*

*Proof.* We prove the lemma using two alternative explanations: effective state size, and statistical error analysis.

**Effective state size.** Let us carefully examine the model output computation. For simplicity, let us rely on the weights from Lem. F.2.2. We choose head-dependent key and query weights, $S_B^h = S_C^h = F^h$, where $F^h \in \mathbb{R}^{N \times D}$. Each head output is then

$$\hat{y}_t^h = \sum_{\tau=0}^{t} \hat{x}_\tau \, (F^h \, \hat{x}_{\tau-1})^\top (F^h \, \hat{x}_t), \tag{129}$$

with $\hat{x}_t \equiv E \, x_t$. The overall SSM output is, hence, aggregation of $\{\hat{y}_t^h\}$:

$$\hat{y}_t = \sum_{h=1}^{M} W_y^h \sum_{\tau=0}^{t} \hat{x}_\tau \, (F^h \, \hat{x}_{\tau-1})^\top (F^h \, \hat{x}_t), \tag{130}$$

which is, since $W_y^h$ and $F^h$ are all independent, equivalent to:

$$\hat{y} = \sum_{\tau=0}^{t} \hat{x}_\tau \, (F' \, \hat{x}_{\tau-1})^\top (F' \, \hat{x}_t), \tag{131}$$

where $F' \in \mathbb{R}^{(MN) \times D} \equiv \mathbb{R}^{N' \times D}$. This implies that effectively, MHA patterns utilizes a larger state size $N' = MN$.

**Statistical error analysis.** Alternatively, let us quantify how the additional heads shrink the recall error. We have shown in Theorem 3 that the *correct*, *wrong* and *empty* entries of the output vector $y_t$, namely $i, j, l$ respectively, are distributed as: (see Eq. 89)

$$\begin{aligned}
y_t^i &\sim \mathcal{N}(1, \ \sigma_d^2 + \tilde{\sigma}_d^2 + N_f \, \sigma_o^2 \, \tilde{\sigma}_o^2) \\
y_t^j &\sim \mathcal{N}(0, \ \sigma_o^2 + \tilde{\sigma}_o^2 + N_f \, \sigma_o^2 \, \tilde{\sigma}_o^2) \\
y_t^l &\sim \mathcal{N}(0, \ \sigma_o^2 + N_f \, \sigma_o^2 \, \tilde{\sigma}_o^2)
\end{aligned} \tag{132}$$

The $\sigma_o^2$, $\sigma_d^2$ terms originate from embedding compression noise, via $\hat{x}_i^\top \hat{x}_j = x_i^\top E^\top E \, x_j \sim \mathcal{N}(\delta_{ij}, \frac{c}{D})$, with $c$ a constant. This variance terms $\sigma_o^2$, $\sigma_d^2 = O(\frac{1}{D})$ cannot be reduced using additional heads, since they arise from input and output embedding $E$ compression and decompression. However, the splitting into multiple SSM heads does reduce the variance. The $\tilde{\sigma}_o^2$, $\tilde{\sigma}_d^2$ terms arise from SSM compression noise, originally via $(F^h \hat{x}_i)^\top (F^h \hat{x}_j) = x_i^\top E^\top F^{h\top} F^h E \, x_j \sim \mathcal{N}(\delta_{ij}, \frac{c}{N})$. Now since we have $M$ independent noise terms, each with variance $(\tilde{\sigma}^h)^2 = O(\frac{1}{N})$, their weighted average have a smaller variance, reduced by a factor of $M$, namely $\tilde{\sigma}^2 = O(\frac{1}{MN})$. Therefore, we can effectively think of this model as of a single-headed model with state size $N' = MN$. $\qquad\square$

## G  MQAR WITH REPEATING KEYS AND QUERIES

In this appendix we study two more general variants of the classical MQAR benchmark: *MQAR with repeating keys* and *MQAR with repeating queries*. We define each of the variants, and present our experimental results.

### G.1  DEFINITIONS

In the original MQAR task (also see full definition here), for each query $q_i$, the model's task is to retrieve and output the value $v^*$ corresponding to the fact $(k^*, v^*)$ with a matching key $k^* = q_i$. An example input $x$ and ground-truth output $y$ sequences can be found below, with highlighted key, value, query and padding tokens:

$$x \quad \text{A 6 B 3 C 7 B 2 5 0 9 C 4 A 8 1}$$
$$y \quad \star \ \star \ \star \ \star \ \star \ \star \ \star \ 3 \ \star \ \star \ \star \ \star \ 7 \ \star \ 6 \ \star \ \star$$

where $\star$ denotes an ignored output token (whose classification does not contribute to the loss).

We define **MQAR with repeating keys** as a generalization of MQAR where the context contains $N_k$ repetitions of each key, and the model has to retrieve the value associated with the *last* appearance of the key, as in the following highlighted example ($N_k = 2$):

$$x \quad \text{A 2 B 9 C 4 A 6 B 3 C 7 B 2 5 0 9 C 4 A 8 1}$$
$$y \quad \star \ \star \ \star \ \star \ \star \ \star \ \star \ \star \ \star \ \star \ \star \ \star \ 3 \ \star \ \star \ \star \ \star \ 7 \ \star \ 6 \ \star \ \star$$

This task is harder, in terms of in-context learning, compared to the original MQAR, since the model must update its internal key-value associations while processing the context.

In contrast, we define **MQAR with repeating queries** as a generalization of MQAR where each query appears $N_q$ times, and the model should retrieve a single value associated with the key multiple times. An example is found below ($N_q = 2$):

$$x \quad \text{A 6 B 3 C 7 B 2 5 0 9 C 4 A 8 1 A 3 1 B 7 6 2 C 5}$$
$$y \quad \star \ \star \ \star \ \star \ \star \ \star \ \star \ 3 \ \star \ \star \ \star \ \star \ 7 \ \star \ 6 \ \star \ \star \ 6 \ \star \ \star \ 3 \ \star \ \star \ \star \ 7 \ \star$$

We denote by $N_q$ the number of repetitions. This task is also challenging in terms of in-context learning. The model has to ignore the random values that appear after the queries, to avoid creating new associations.

### G.2  MQAR WITH REPEATING KEYS

Let us begin with the following informal insight, which demonstrates the challenging nature of the task by analyzing the failure of our existing solution described in Theorem 2.

**Remark 3.** A *single-layer simplified Mamba* with dimensions $D, N = O(\log V)$ and weights designed as specified in Theorem 2 **fails** to solve *MQAR with repeating keys*.

**Explanation.**  For simplicity, we consider the following sequence, where $N_k = 2$: $x = (k, v_1, k, v_2, \ldots, k)$, where the last token is a query $q_t = k$. Let us examine the MQAR solution from Theorem 2 (see full proof for notation). The key $k$ appears twice in the context; the model encodes it using $k' = \tilde{E}k$. Each of the associated values $v_1$ and $v_2$ is compressed by the model into $v_1' = Ev_1$ and $v_2' = Ev_2$, respectively. While processing the key-value pairs in the context, both associations are written into the SSM hidden state. Using Eq. 12, we have $H_t' = \sum_{n=1}^{N_f} v_n' k_n'^{\top} = (v_1' + v_2')k'^{\top} = E(v_1 + v_2)k\tilde{E}^{\top}$. For the query $q_t = k$, we have an encoded representation $q_t' = \tilde{E}k$. Following the retrieval formulation in Eq. 12, the model output is $y_t = E^{\top}\sum_{n=1}^{N_f} v_n' k_n'^{\top} q_t' = E^{\top}E(v_1 + v_2)k\tilde{E}^{\top}\tilde{E}k \approx v_1 + v_2$. Namely, the model output consists of both $v_1$ and $v_2$, both having an equal weight. Hence, output value token is 1 or 2 with an approximately equal probability of $\frac{1}{2}$.

The picture is different, however, for a Mamba model with multiple layers. In the following remark, we brief an intuitive, informal explanation why a two-layer model better solves the task.

**Remark 4.** A *multi-layer simplified Mamba* with $\Lambda = 2$ layers and dimensions $D, N = O(\log V)$ is able to solve an *MQAR with repeating keys* task of $N_k = 2$.

**Intuition.** In order to associate the *last* key-value pair only, the required state update step is the following:

$$H'_t = H'_{t-1} + v'_t {k'_t}^\top - v'_\tau {k'_\tau}^\top \tag{133}$$

where $v'_t$ is the new value and $v'_\tau$ is the stored value. This way, if $k'_t = k'_\tau = k'$, the update state becomes $H'_t = H'_{t-1} + (v'_t - v'_\tau){k'}^\top$. Given an example sequence $x = (k, v_1, k, v_2, \ldots, k)$, this implies that at the first fact $(k, v_2)$ the value $v'_1$ is stored (since $H'_t$ is initialized with 0, thus $v'_t - 0 = v'_1$). In the next appearance $(k, v_2)$, the stored value $v'_1$ is replaced with $v'_2$, since $v'_1 + (v'_2 - v'_1) = v'_2$. Then, given a query $q_t = k$, the retrieved value is $v_2$, as required. Such a mechanism cannot be learned in a single-layer Mamba model, since $v'_\tau$ is not accessible as an input value, but is rather stored at the hidden state. However, in a two-layer model, layer 1 output *reads* from the hidden state, $y^{(1)}_{t,\text{out}} \approx v'_\tau$. Hence, layer 1 can pass the value $v'_\tau$ to layer 2. This way, $v'_t - v'_\tau$ can serve as the *input* value to the SSM, thus implement the update rule discussed above.

**Experimental results.** We train single-layer and multi-layer simplified Mamba models on *MQAR with repeating keys*. Our results are presented in Figure 10. Strikingly, for $N_k = 2$, the 1-layer models fail the task, while the 2-layer models perfectly succeeds it. The 3-layer model does not show significantly better results than achieved with 2 layers. (For $N_k = 3$, the 2-layer model does not achieve perfect performance. This is, however, beyond the scope of the current study.) Interestingly, for $N_k = 2$ the single-layer model accuracy is approximately bounded by $\frac{1}{2}$; for $N_k = 3$, the bound is roughly $\frac{1}{3}$. This aligns with our intuition (see Remark 3) of "guessing" one out of $N_k$ options, when the 1-layer model can only access a linear combination of the $N_k$ stored values.

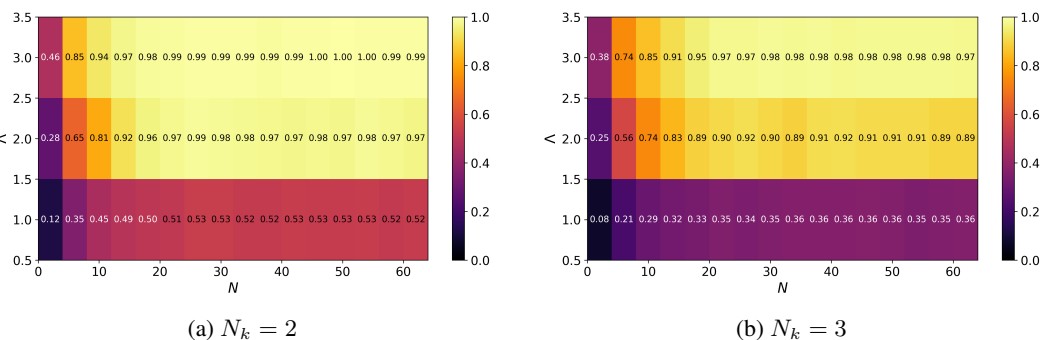

(a) $N_k = 2$           (b) $N_k = 3$

Figure 10: **MQAR with repeating keys.** Left: while a 1-layer model fails the $N_k = 2$ repeating keys task, independently of state size $N$, a 2-layer model perfectly solve it. Right: for $N_k = 3$ repeating keys, a 3-layers model is required for perfect recall.

### G.3 MQAR WITH REPEATING QUERIES

We start with a similar informal remark, explaining how our existing solution described in Theorem 2 fails to solve the *repeating queries* variant of MQAR.

**Remark 5.** A *single-layer simplified Mamba* with dimensions $D, N = O(\log V)$ and weights designed as specified in Theorem 2 **fails** to solve *MQAR with repeating queries*.

**Explanation.** Let us consider the following sequence, $x = (k, v_f, \ldots, k, v_n, \ldots, k)$, for which $N_q = 2$. The context *fact* $(k, v_f)$ contains the fact value $v_f$ that should be retrieved. There are two queries, which we denote $q_1 = q_2 = k$. The first query $q_1$ is followed by a random value $v_n$, considered a *noise* value. We analyze the MQAR solution from Theorem 2. The model encodes $k$ via $k' = \tilde{E}k$. The values $v_f$ and $v_n$ are compressed into $v'_f = Ev_f$ and $v'_n = Ev_n$, respectively. Upon processing the context, both values are written into the SSM hidden state, since the model does not distinguish between a fact and a query-noise pair. Using Eq. 12, we have $H'_t = (v'_f + v'_n){k'}^\top = E(v_1 + v_n)k\tilde{E}^\top$. For the first query $q_1 = k$, we have a correct retrieval $y_1 \approx v$, as guaranteed from Theorem 2. However, for the second query, we have $y_t = E^\top H'_t q'_2 = E^\top E(v_f + v_n)k\tilde{E}^\top \tilde{E}k \approx$

$v_f + v_n$. The model now outputs the $f$ and $n$ tokens with an approximately equal probability of $\frac{1}{2}$. Overall, model accuracy is roughly $\frac{3}{4}$ (since first query always succeeds).

Similarly to Remark 4, we intuitivaly claim that unlike the single-layer simplified Mamba model, a two-layer model is able to solve MQAR with repeating queries.

**Remark 6.** A *multi-layer simplified Mamba* with $\Lambda = 2$ layers and dimensions $D, N = O(\log V)$ is able to solve an *MQAR with repeating queries* task of $N_k = 2$.

**Intuition.** In order to associate the *first* key-value pair only, a possible state update step is:

$$H'_t = H'_{t-1} + 2\,v'_\tau k'_\tau{}^\top + \frac{1}{2}v'_t k'_t{}^\top \tag{134}$$

where $v'_t$ is the new value and $v'_\tau$ is the stored value. If $k'_t = k'_\tau = k'$, the update state becomes $H'_t = H'_{t-1} + (2\,v'_\tau + \frac{1}{2}v'_t)k'^\top$. Assume, for instance, we have a sequence $x = (k, v_f, \ldots k, v_n, \ldots, k)$. For the fact $(k, v_f)$, the model stores the value $\frac{1}{2}v'_f$ is stored (since $H'_t$ is initialized with 0). In the first query $(k, v_n)$, the stored value is updated to $\frac{1}{2}v'_f + (2\left(\frac{1}{2}v'_f\right) + \frac{1}{2}v'_n) = \frac{3}{2}v'_f + \frac{1}{2}v'_n$. Then, for the second query $q_2 = k$, the retrieved value is $y_t = \frac{3}{2}v_f + \frac{1}{2}v_n$, which means the $f$ token is chosen over $n$, as required. As in Remark 4, such a mechanism cannot be learned in a single-layer Mamba model, but is possible for a two-layer model.

**Experimental results.** We train single-layer and multi-layer simplified Mamba models on *MQAR with repeating queries*. Our results are presented in Figure 11. As in the *repeating keys* case, the 1-layer cannot solve it perfectly, while the 2-layer models achieves perfect recall. The 3-layer model performs almost similarly to the 2-layer model. Note that for $N_q = 2$, the single-layer model achieves an accuracy quite similar to our $\frac{3}{4}$ prediction from Remark 6.

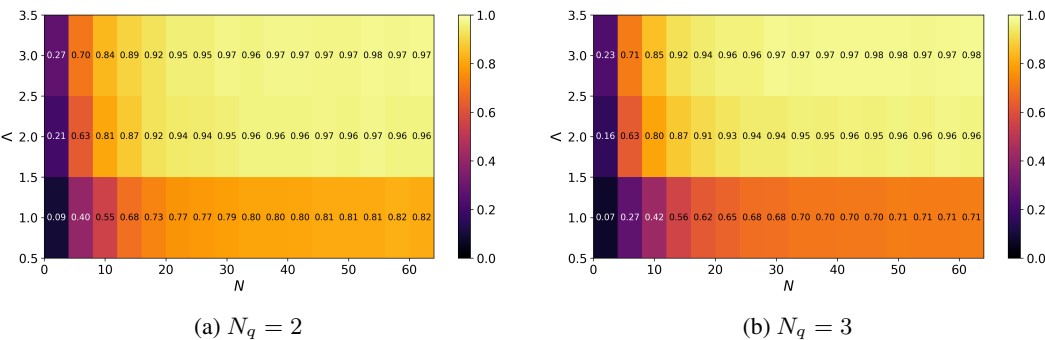

(a) $N_q = 2$                (b) $N_q = 3$

Figure 11: **MQAR with repeating queries.** While 1-layer model fails to achieve perfect recall in *repeating queries* task of $N_q = 2$, $N_q = 3$, the 2-layer and 3-models models solve it perfectly, almost independently of state dimension $N$.

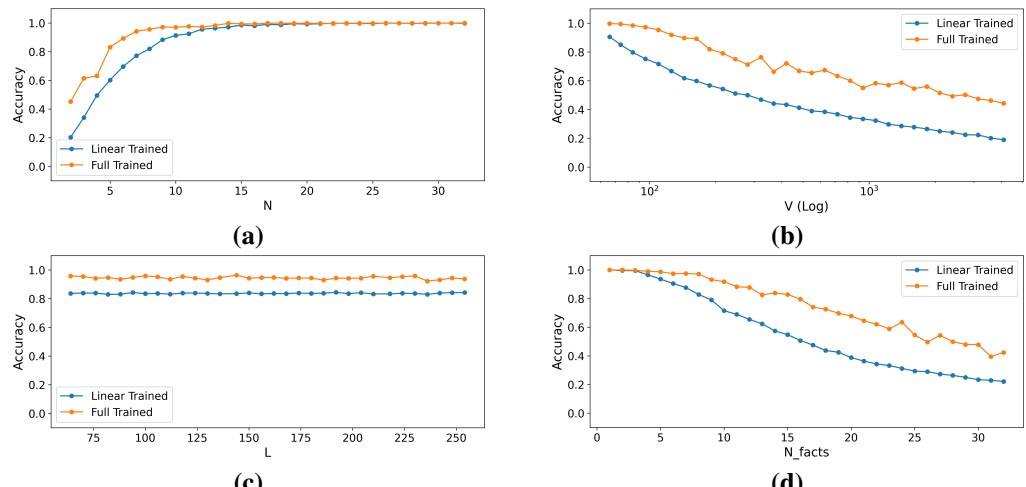

Figure 12: **Full model versus simplified model.** We compare MQAR performance of a full single-layer Mamba model to our simplified linear model. It can be seen that full model always perform better, but the scaling trends are similar to these of the simplified model. In each figure, the other model dimensions and MQAR parameters are fixed.

## H    SCALING LAWS EXPERIMENTS: SUPPLEMENTARY MATERIAL

### H.1    SIMPLIFIED VERSUS FULL MODEL

To validate that the scaling trend of the simplified model is similar to that of the original full model, we conducted additional experiments, as reported in Figure 12. As can be seen, the scaling trends are relatively similar, demonstrating the relevance of our theory to real models.

## H.2 MULTI-HEAD RECALL SCALING LAWS

In this appendix we provide additional experiments for the multi-head Mamba recall circuits, validating the theoretical results established in App. E and App. F.

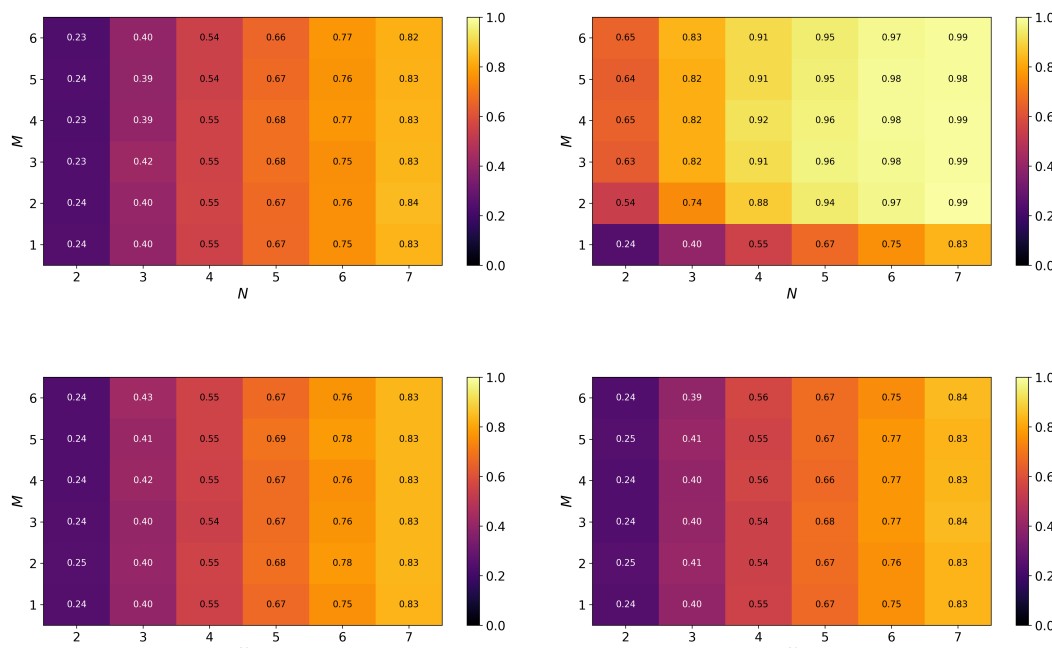

Figure 13: **Scaling laws for multi-head SSM patterns**. In order to improve recall by splitting into $M$ heads, both key and query dimensions should increase from $N$ to $MN$ (MHA, bottom right). Sharing keys (MQA), values (MKA) or both (MVA) across SSM heads implies similar accuracy as in the single-headed SSM.

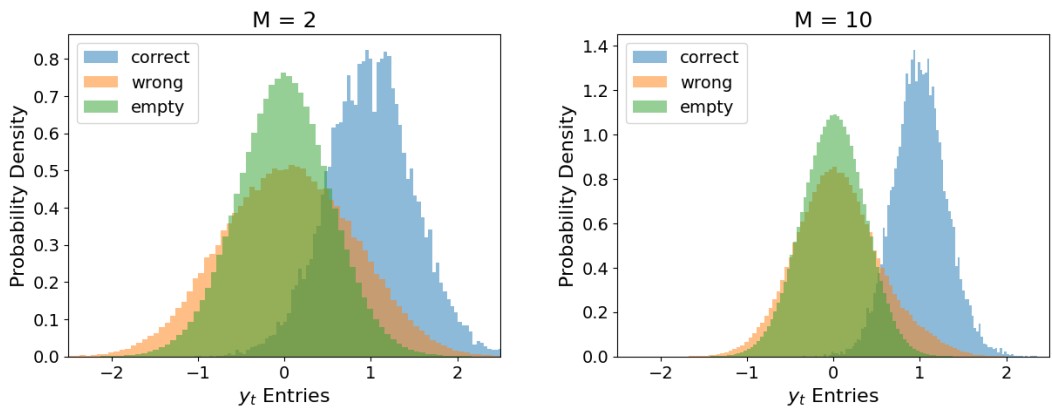

Figure 14: **Head-pattern effect on model output distribution (visualization).** As predicted in Lem. 5, for the MHA pattern, where SSM heads are independent, splitting into more heads shrinks model output distribution variance, which leads to better recall performance.

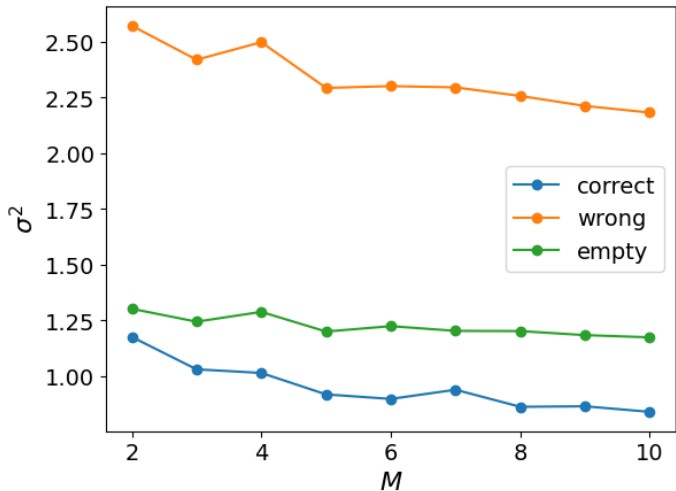

Figure 15: **Head-pattern effect on model output distribution.** For the MHA pattern, output entries variance is reduced upon addition of SSM heads.

# I SIMILARITIES AND DIFFERENCES WITH *Understanding Input Selectivity in Mamba*

This section provides a methodological and technical comparison between our work and Huang et al. (2025).

Starting with a bird-eye view, both works focus on Mamba model variants and their ability to perform associative recall. In addition, both works offer a detailed theoretical perspective, supported by experiments. Our study, however, proves *tighter bounds* for the dimensions required for the Mamba recall circuit. These tighter bounds enable both *mechanistic investigation* and theoretical derivation of highly-predictive AR *scaling laws* for Mamba, which are *empirically verified* - capabilities not possible with the bounds established by Huang et al. (2025).

The main points of difference lie in both contribution and methodology:

1. We theoretically show that Mamba can learn a parameter-efficient solution for MQAR, i.e. a solution which requires feasible model dimensions $D, N = O(\log V)$. These are **tighter bounds** compared to the bounds proved by Huang et al. (2025) for Mamba, which are $D = O(V_k + \log V_v)$ and $N = V_k$.

2. Our suggested recall circuit is validated by **mechanistic interpretability** experiments, based on trained model weights and activations. In other words, our interest lies in the mechanism the model *truly learns* during training, as opposed to those it *could, in principle, learn* based solely on its architectural capacity.

3. Our contribution includes the theoretical derivation of **scaling laws** for recall in Mamba, as well as empirical validation. The exact scaling laws (Theorem 3 and Remark 1), which align with experimental result, cannot be derived from Huang et al. (2025) Theorem 1, since it is based on a different circuit.

4. In addition to single-layer models, our study generalizes to **multi-layer** Mamba models, aiming to contribute a further understanding of multi-layer recall circuits (Theorem 4). Our conclusions are validated through empirical examination (Section 5.2).

5. While Huang et al. (2025) focuses on single-head SSM mixers for Mamba-2, our work further contributes the study of **multi-head** recall circuits, as enabled by the Mamba-2 architecture (Theorem 5). The theoretical insights are further verified by experimental analysis.

6. By methodology, we seek a **minimal recall circuit**, in order to isolate factors and find the most significant component for recall (see Appendix K). Upon removal of model components, we identify linear compression limits as the main factor, represented by the model dimensions $D$ and $N$. Neither gating nor nonlinearities drastically overcome this inherent bottleneck (as shown in Figure 6).

Let us now carefully dive into a technical comparison between the works.

(Note that by *Mamba*, we specifically refer to *"Mamba-1"* architecture, as opposed to *Mamba-2*.)

**Non-compressive recall circuits.** Lemma 3 in Huang et al. (2025) proves that a 1-layer *Mamba-$\Delta^\top$* model can solve the Induction Heads task, if model dimensions are $D, N = O(V)$. Theorem 1 in our work leverages a similar construction for a 1-layer *Mamba*, showing it can solve MQAR, given $D, N = O(V)$. Importantly, however, this Theorem is not the key novelty of our work, but rather serves as an intermediate logical step towards Theorem 2, which addresses a realistic scenario where $D, N = O(\log V) < O(V)$, thus applicable to actual MQAR tasks.

**Compressive recall circuits.** Theorem 2 in Huang et al. (2025) shows that a 1-layer *Mamba* with dimensions $D = O(V_k + \log V_v)$ and $N = V_k$ can solve MQAR, by applying JL lemma to the values (but not to the keys, which remain uncompressed).

Firstly, Theorem 2 in our work improves the **efficiency** of the solution, by showing that a 1-layer *Mamba* model can solve MQAR with smaller, realistic dimensions $D, N = O(\log V)$, by using a different construction, which allows us to apply JL lemma *twice*, to both values and keys. (Technically, we show *Mamba* can compress its inputs twice: first compression via embedding into

$Ex_t \in \mathbb{R}^D$, used for values; then a second compression via $S_B, S_C$ into $FEx_t \in \mathbb{R}^N$, used for keys and queries.) This difference is of major importance for realistic Mamba models and MQAR tasks, where $N = V_k$ and $D = O(V_k)$ are impractical and cannot be used due to memory limitation. A parameter-efficient solution is critical in practice; see, for instance, Figure 6, row (iii), where we empirically show that a 1-layer Mamba can perfectly solve an MQAR task of $V_k = \frac{V}{2} = 512$ given relatively small dimensions $N = 32 << V_k$ and $D = 64 << V_k$.

Secondly and most importantly, our **mechanistic analysis** in Section 3.2 (also see Figure 3) highlights the difference between the two constructions. The solution suggested by Huang et al. (2025) implies that *each (hidden state) column corresponds to a specific key*: each embedded value $Ev_\tau$ is stored in a **single**, distinct key coordinate of the hidden state $h_t$ (equivalent to $H'_t$ in Section 3.2), out of $N = V_k$ coordinates in total. In contrast, our parameter-efficient construction suggests each embedded value $Ev_\tau$ is stored in a **distributed** manner, across a linear mixture of **all** $N = O(\log V_k)$ key coordinates. The mechanistic experiments we conduct in Section 3.2 support our suggested circuit. It shows that *Mamba* actually learns a solution where writing a single key-value pair into $H'_t$ affects **all** key coordinates, leading to the pattern observed for $H'_t$, where all entries are nonzero (Figure 3). If, conversely, the learned mechanism would have used a **single** key entry per value $Ev_\tau$, one could have inspected the hidden state by decompressing value coordinates **only**, via $H''_t = E^\top H'_t$, since key coordinates would not interfere with one another. However, we experimentally show that a **double**-decompression is rather required, $H''_t = E^\top H'_t \tilde{E}$ (see Equation 13 and Figure 3), supporting our theory.

Thirdly, our theoretical and empirical **scaling laws** (Theorem 3 and Figure 6 respectively) indicate that *Mamba* recall capacity strongly depends on both dimensions $N$ and $D$. The exact form of this dependence, where performance improves upon increasing $\frac{ND}{N_f}$, arises due to *noisy retrieval* of values out of the hidden state $H'_t$. Importantly, while the $\frac{1}{D}$ factor is caused by value reconstruction noise $v''_i = E^\top Ev_i + O(\frac{1}{D})$, the $\frac{N_f}{N}$ factor arises from noisy query-key matching $\sum_p^{N_f} (\tilde{E}k_p)^\top (\tilde{E}k_m) = O(\frac{N_f}{N})$. This is only the case if values are stored as $H'_t = \sum_p^{N_f} (Ev_p)(\tilde{E}k_p)^\top$, where keys are compressed into $\tilde{E}k_p \in \mathbb{R}^N$. Without relying on such a *double-compressive* circuit (as in Theorem 2, as opposed to Theorem 1 in Huang et al. (2025)), the derived scaling law would lack this important $\frac{N_f}{N}$ factor, which is clearly demonstrated in practice (Figure 6).

**Minimal recall circuits.** Our work focuses on finding a **minimal** Mamba model that can solve MQAR, in order to isolate the contributing factors. In our perspective, this approach is important in order to understand potential **bottlenecks** in Mamba architecture. Since Mamba have a variety of components (including discretization, gating, nonlinearities, normalization layers, input selectivity, etc.), there may *exist* many possible constructions combining them that can *possibly* enable Mamba to solve MQAR. Minimality is required if we aim to find a critical component.

Firstly, our ablation study (Appendix K; see table entry B) clearly shows that a Mamba model without gating is still able to achieve perfect accuracy on MQAR. This hints that, while probably participating in the recall process, the gate-based mechanism is empirically not a bottleneck for Mamba ability to perform recall.

This insight is further reinforced by the results presented in Figure 6. The conducted experiment compares between our gate-free simplified model and the full gated model, both trained on MQAR. Our findings show the models perform quite similarly on MQAR. This demonstrates that the main bottleneck in Mamba recall capacity is the model dimensions $D$ and $N$ (responsible for reliable compression-decompression schemes of keys and values), rather than the existence of a gate.

## J  TRAINING AND IMPLEMENTATION DETAILS

This appendix provides all training and implementation details needed to reproduce our results. We use PyTorch (Paszke et al., 2019) for all experiments.

### J.1  DATASETS

We use the original MQAR dataset as implementaed by Arora et al. (2023). To avoid generalization error and overfit phenomena, instead generating large training datasets statically, we sample new batches from MQAR in any training step.

### J.2  MODELS

As for the full Mamba model, we use `mamba-ssm` by (Gu & Dao, 2023), unmodified. Our simplified linear Mamba model is implemented based on `mamba-tiny` open-source Github repo.

### J.3  TRAINING HYPERPARAMETERS

Unless stated otherwise, we use the hyperparameters in Table 9.

Table 9: Default training hyperparameters.

| | |
|---|---|
| Batch size | 128 |
| Optimizer | AdamW |
| Base learning rate | $1 \times 10^{-2}$ |
| LR schedule | 500-steps warmup, linear decay by 10 in 15000 steps |
| Weight decay | 0.1 |
| Max. step | 3000 for simplified model, 10000 for full model |
| Label smoothing | 0.1 |
| Gradient clipping | 0.75 (global norm) |

### J.4  COMPUTE AND ENVIRONMENT

All experiments were run on *8×RTX 11GB* GPUs with CUDA 12.4 and PyTorch 2.6.0. We fix `torch`, `numpy`, and `python` versions in a `requirements.txt`. Wall-clock for a 10000-steps run for $32 \times 32$ parameters grid with 10 seeds is $\sim 24$ hours.

### J.5  SEEDS AGGREGATION CONSIDERATIONS

We briefly describe our considerations choosing best-of-seeds accuracy. In this work we investigate Mamba's recall *capacity*, therefore retaining the maximal result achieved by each configuration is the most relevant quantity for our analysis. In other words, we are interested in the *existence* of a recall solution, not the *probability* of achieving it by optimization. Therefore, it makes more sense to take into account only the trials where optimization succeeds.

# K    MQAR Ablation Study

We use a vocabulary size of $V = 128$. Each sample contains 16 key-value pairs and is $L = 64$ tokens long. The sample consists of 32 context tokens and 32 additional tokens for the queries and answers. The training set includes 200K samples, and the test set contains 2K samples. To augment the training data, for each sampled sequence we shuffle the order of the queries four times while keeping the context fixed. Therefore, if we sample $S$ sequences from the dataset, the effective batch size is $4S$. For settings Base and A we use $S = 112$, and for settings B, C, D, and E, we use $S = 224$. We train using a learning rate of 0.01 and weight decay of 0.1 for 10 epochs. Each model is trained with 3 different seeds, and the results are averaged. The model has a single layer with $D = 64$ channels and a state dimension of $N = 16$. We use tied input and output embeddings. As in the original Mamba block, the value of $D_{\text{conv}} = 4$ unless stated otherwise.

Table 10: **MQAR Ablation.** To find the critical components of Mamba's recall circuit we ablate the components of a 1-layer Mamba model in the MQAR task. Each row in the table removes an additional component from the Mamba block. The *MQAR Accuracy* column shows the performance of the model, averaged after training over 3 different seeds. Full implementation details can be found in Sec. K. Besides the SSM itself, we find that the convolution block is the most critical component of Mamba's recall circuit.

| Model ID | Description | MQAR Accuracy |
|----------|-------------|---------------|
| Base | 1-Layer Mamba model w/o layer normalization | $\mathbf{0.99} \pm 0.01$ |
| A | Base model with $\bar{A}_t = I$ | $\mathbf{1.00} \pm 0.00$ |
| B | Model A w/o gate | $\mathbf{0.98} \pm 0.01$ |
| C | Model B w/o activation function (post-Conv1D) | $\mathbf{0.99} \pm 0.01$ |
| D | Model C with $d_{\text{conv}} = 2$ | $\mathbf{0.96} \pm 0.05$ |
| E | Model C w/o Conv1D | $\mathbf{0.00} \pm 0.00$ |

