# OpenReview forum: "On the Recall Scaling Laws in Mamba: A Theoretical and Mechanistic Study via Hashing"
_ICLR.cc/2026/Conference — Submitted to ICLR 2026_

### Official Review · Reviewer_jebP · 2025-10-31

**Soundness:** 3
**Presentation:** 1
**Contribution:** 1
**Rating:** 2
**Confidence:** 4

**Summary:**

This paper studies the associative recall mechanism in Mamba architecture. The first part proves that Mamba models can solve the multi-query associative recall task and demonstrates that the trained Mamba architecture closely follows the construction in the proof (up to equivalent orthogonal transformations). The second part builds on the construction to derive the scaling law for associative recall and empirically validates the law.

**Strengths:**

1. This paper addresses an important problem of how state space models retrieve information from their context.

2. The paper validates its theoretical constructions through empirical results.

**Weaknesses:**

1. The theoretical construction of Mamba architecture, solving multi-query associative recall, has already been discovered in the previous work (Huang et al. 2025). Empirical observations and scaling laws are novel, but these results lack connections to practical settings (since they are restricted to single-layer models) and quite straightforward corollaries from the construction.

2. The presentation of theoretical results can be improved. There are occasional notations without proper definitions (e.g., Dtilde in L167, N in 174), inconsistent variable names (e.g., different names between Figure 1 and Algorithm 1), inconsistent usage of notations (e.g., P_in and P_out use the same concatenation but should be along different dimensions), typos in equations (e.g., Equation 7 has unclear commas, wrong ordering of E and P). In general, these problems challenge understanding the ideas. The paper appears to have significant room for improvement by checking for consistency and refining the writing.

**Questions:**

1. How is the construction in this paper different from that in Huang et al. 2025?

2. Would the empirical observations (mechanistic and scaling laws) still be valid in multi-layer settings?

---

> ### Author Response · Authors · 2025-12-03
> **Response to Reviewer jebP (Part 1/2)**
>
> We thank the reviewer for the effort and the time you have dedicated to reviewing our submission. Below are our responses.
>
> .
> > **W.1.1.** Empirical observations and scaling laws are novel but these results lack connections to practical settings (since they are restricted to single-layer models).
>
>
> We thank you for pointing this out. To address the reviewer's concern, we have extended our work beyond single-layer models. Specifically, in the revised paper, we have added a further theoretical analysis of multi-layer recall in Mamba (Theorem 4), supported by additional experiments (Section 5.2).
>
> Briefly summarizing our results, we re-use our construction from the single-layer case and find that a Mamba model with $\Lambda$ layers can perfectly perform associative recall relying on smaller dimensions $D=O(\log V)$, $N=O(\frac{\log V}{\Lambda})$.
>
> .
> > **W.1.2.** “The theoretical construction of Mamba architecture, solving multi-query associative recall, has already been discovered in the previous work (Huang et al. 2025).”
>
>
> The reviewer raised a concern regarding the similarities between our work and Huang et al. (2025). We respond as follows:
>
>
> Indeed, both works focus on Mamba model variants and their ability to perform associative recall. In addition, both works offer a detailed theoretical perspective, supported by experiments. \
> However, our work improves upon Huang et al. (2025)’s in several aspects: (1) as detailed below, our work proves **tighter bounds** for the dimensions required for the Mamba recall circuit. (2) These tighter bounds enable both **mechanistic investigation** and theoretical derivation of **highly-predictive AR scaling laws** for Mamba, which are **empirically verified**. These results could not have been achieved with the bounds established by Huang et al. (2025).
>
>
> Before diving into the technical details, let us brief the different contributions of our work. \
> In addition, we refer the reviewer to the revised manuscript, which now includes a detailed comparison, attributing both similarities and differences (Appendix I):
> 1. We theoretically show that Mamba **can learn a parameter-efficient solution for MQAR**, i.e. a solution which requires feasible model dimensions $D,N=O(\log{(V_k + V_v)})$. These are **tighter bounds** compared to the bounds proved by Huang et al. for Mamba, which are $D=O(V_k+\log{V_v})$ and $N=V_k$.
> 2. Our suggested recall circuit is validated by **mechanistic interpretability** experiments, based on trained model weights and activations.
> 3. Our contribution includes the theoretical derivation of **scaling laws** for recall in Mamba, which cannot be derived from Theorem 1 in Huang et al., since it is based on a different circuit.
> 4. In addition to single-layer models, our study generalizes to **multi-layer** Mamba models (Theorem 4).
> 5. We further contribute the study of **multi-head recall circuits for Mamba-2 models** (Theorem 5).
> 6. Our study shows the suggested recall circuit is **minimal**, which is important for understanding the bottlenecks for recall in Mamba.
>
> \
> Let us carefully consider the differences between the constructions:
>
>
> Theorem 2 in Huang et al. shows that a 1-layer Mamba with dimensions $D=O(V_k+\log{V_v})$ and $N=V_k$ can solve MQAR, by applying JL lemma to the values.
>
> Firstly, Theorem 2 in our work improves the **efficiency** of the solution, by showing that a 1-layer Mamba model can solve MQAR with smaller, realistic dimensions $D,N=O(\log{V})$, by using a different construction, which allows us to apply JL lemma **twice**, to both values and keys. (Technically, we show Mamba can compress its inputs twice: first compression via embedding into $ Ex_t\in \mathbb{R}^{D} $, used for values; then a second compression via $S_B,S_C$ into $FEx_t \in \mathbb{R}^{N}$, used for keys and queries.) This difference is of major importance in practice: see Figure 6, where we empirically show Mamba can solve an MQAR task of $V_k=\frac{V}{2}=512$ given dimensions $N=32<<V_k$ and $D=64<<V_k$.
>
> Secondly, our **mechanistic analysis** (also see Figure 3) highlights the difference between the two constructions. The solution suggested by Huang et al. implies that **each column corresponds to a specific key**: each embedded value $Ev_\tau$ is stored in a **single** distinct key coordinate of the hidden state $h_t$, out of $N=V_k$ coordinates in total. In contrast, our parameter-efficient construction suggests each embedded value is stored in a **distributed** manner, across a linear mixture of *all* $N=O(\log{V_k)}$ key coordinates. The mechanistic experiments we conduct in Section 3.2 support our suggested circuit.
>
> Thirdly, the fact Theorem 2 in our work uses JL compression twice allows a derivation of the correct scaling law, which depends on both model dimensions $N$ and $D$ (see Theorem 3). Theorem 2 at Huang et al. uses a different circuit, hence would have resulted in a different scaling law. (See detailed explanation in Appendix I.)

---

> ### Author Response · Authors · 2025-12-03
> **Response to Reviewer jebP (Part 2/2)**
>
> > **W.1.3.** Empirical observations [...] are novel but [...] quite straightforward corollaries from the construction.
>
> We believe that the mechanistic-interpretability empirical observations (section 3.2) provided in our work are not straightforward given the construction, and that our methodology to address observation difficulties is novel.
>
>
> The circuit constructions (both in our work and in Huang et al.) describe ideally-formed model weights - projections, convolution kernel, etc. One might think that observing the suggested circuit by examination of model is immediate upon extracting model weights (see i.e. our answer to Q.2. by Reviewer 1qe9). However, we believe it is nontrivial, and requires several innovative steps, which are novel in SSM research as far as we are concerned:
>
> Identification and construction of the invariant operators, $G_{vv}$, $G_{kq}$ (see equations 8, 9).
>
> Using net operators to inspect and interpret model hidden dimensions, via $H_t^{\prime\prime}=\Pi_{v,\mathrm{out}} H_t^{\prime} \Pi_{q,\mathrm{in}}$.
>
> Using net operators to reveal copy and shift patterns in SSM, which are of major importance in any of the theoretical constructions, and would otherwise remain a conjecture.
>
> .
> > **W.1.4.** Scaling laws are novel but [...] quite straightforward corollaries from the construction.
>
> We believe that scaling laws derivation is not an immediate corollary given the construction. Several further nontrivial conceptual steps are required:
>
> (a) Switching to a probabilistic approach (rather than a bound-based perspective), (b) applying CLT to the $y_t$ output vector entries, (c) analyzing all cross interactions - all different token contributions (correct / wrong / non- facts) to all types of $y_t$ value entries (correct / wrong / empty, see Appendix D and Table 5 in the revised manuscript), (d) theoretically deriving variance dependencies on both model dimensions $N$, $D$ and problem parameters $V$, $L$, $N_f$, and (e) addressing key and value embeddings orthogonality, leading to independence on $L$. (See Appendix D.)
>
> .
> >**W.2.** The presentation of theoretical results can be improved. [...]
>
> We are grateful for your detailed comment, highlighting several places where typos and mistakes have occurred. Our revised version fixes all typos, inconsistencies and mismatches you have mentioned. We highly appreciate your attention to details, which is helpful for improving the presentation of our study.
>
> More specifically:
> * Notations without proper definitions (e.g., $\tilde{D}$ in L167, N in 174) -> Fixed everywhere.
> * Inconsistent variable names (e.g., different names between Figure 1 and Algorithm 1) -> Fixed. We have updated Figure 1 to match Algorithm 1 and Theorems 1, 2.
> * Inconsistent usage of notations (e.g., $P_{in}$ and $P_{out}$ use the same concatenation but should be along different dimensions) -> Fixed everywhere.
> * Typos in equations (e.g., Equation 7 has unclear commas, wrong ordering of E and P). -> Fixed everywhere.
>
> \
> **Questions:**
>
> > **Q.1.** How is the construction in this paper different from that in Huang et al. 2025?
>
> Please see our detailed answer to **W.1.2** above.
>
> We included the full details in the revised manuscript (Appendix I).
>
> .
> > **Q.2.** Would the empirical observations (mechanistic and scaling laws) still be valid in multi-layer settings?
>
>
> We thank you for pointing out this concern. \
> We have extended our work beyond single-layer models. In the revised paper, we have added a further theoretical analysis of multi-layer recall in Mamba (Theorem 4), supported by additional experiments (Section 5.2).
>
> Briefly summarizing our results, we find that a Mamba model with $\Lambda$ layers can perfectly perform associative recall relying on smaller dimensions $D=O(\log V)$, $N=O(\frac{\log V}{\Lambda})$.

---

### Official Review · Reviewer_1qe9 · 2025-10-31

**Soundness:** 3
**Presentation:** 3
**Contribution:** 2
**Rating:** 4
**Confidence:** 4

**Summary:**

The paper performs a theoretical and mechanistic study of how Mamba solves associative recall, in particular the MQAR task. Focused on a simplified model (one-layer linear Mamba), the authors identity a solution for solving MQAR and verify such solutions present in the learned model weights. The authors then use JL lemma to theoretically show the required model size for solving associative recall for given task parameters. Finally, the theoretical claims are supported via empirical validations on both the simplified model and the original model.

**Strengths:**

1. The mechanistic interpretability analysis on validating the simplified Mamba recall circuit is interesting and nicely done (taking in account suitable invariances).

2. The quantitative analysis of how Mamba solves associative recall is thorough and empirically validated.

**Weaknesses:**

1. The theoretical construction of simplified one-layer linear Mamba largely follows from Huang et al. (2025), without providing enough remarks or discussions. Specifically, Theorem 1 uses the construction of Lemma 3 in Huang et al., and Theorem 2 largely reuses the idea of Theorem 2 in Huang et al. (i.e. using JL lemma to reduce dimensionality). Moreover, Theorem 4 in Huang et al. shows that using convolution and gating (together with a S4D-SSM) can solve MQAR, which takes a step towards understanding the role of gating, a direction that the authors remark for future work. The authors only highlight the difference with the study of Huang et al. in Sec 2, without properly attributing the similarities.

2. The mechanistic and theoretical study focus on a particular task (MQAR) to identify a recall circuit. However, it is unclear how generalizable this circuit is applicable to other related recall tasks (e.g., MQAR with repeating keys, induction head where keys and values are from the same vocabulary).

**Questions:**

1. In most of the theoretical constructions, the hidden state is treated as a matrix with size (N, d) (e.g., eqn 3). However, in equation 2, the hidden state $h_t$ is treated as a vector of size N. Can the authors clarify the purpose of eqn 2 and discuss its relations to the rest of the paper?

2. Fig.3 provides a very interesting inspection of the hidden state. I want to better understand the computation of $H''$: in lines 286-296, the authors comment $H''$ in eqn.13 does not exist as-is, but has to use $H'' = \Pi_{v, out} H_t' \Pi_{q, in}$. But it seems possible to compute eqn.13 from the learned embedding. Can the authors clarify?

3. How do the mechanistic and theoretical findings generalize to Mamba2?

4. The proof of Lemma 1: while the proof steps look correct to me, some wording and analysis look confusing. In line 1028-1030, the non-matching wrong fact pairs and non-matching empty pairs look the same from their mathematical definitions ($x_{\tau-1} \neq k_m, x_{\tau} \neq v_i$). Do the authors mean non-matching wrong fact as $x_{\tau-1} = k \neq k_m$ being a non-matching key, and non-matching empty pair as $x_{\tau-1} = v$ being an arbitrary value?

5. The proof of Thm 3, eqn (98): shouldn't it be $p_{\text{success}} = p_w^{N_f - 1} p_n^{V_v - N_f}$? can the authors double check the rest of the equations are correct?

Minor:

a. The notation $N_f$ in proof sketch of Lemma 1 and Theorem 3 means the same as $N_{\text{facts}}$?

b. What does the pixel color in Fig.6 mean (accuracy)? Also, the color bar is missing.

---

> ### Author Response · Authors · 2025-12-03
> **Response to Reviewer 1qe9 (Part 1/4)**
>
> We thank the reviewer for the valuable, highly detailed comments and important questions.
>
> .
> > **W.1.** Similarities with Huang et al. (2025)
>
> The reviewer raised a concern regarding the similarities between our work and Huang et al. (2025). \
> We respond as follows:
>
>
> Indeed, both works focus on Mamba model variants and their ability to perform associative recall. In addition, both works offer a detailed theoretical perspective, supported by experiments. \
> However, our work improves upon Huang et al. (2025)’s in several aspects: (1) as detailed below, our work proves **tighter bounds** for the dimensions required for the Mamba recall circuit. (2) These tighter bounds enable both **mechanistic investigation** and theoretical derivation of **highly-predictive AR scaling laws** for Mamba, which are **empirically verified**. These results could not have been achieved with the bounds established by Huang et al. (2025).
>
>
> Before diving into the technical details, let us brief the different contributions of our work.
> In addition, we refer the reviewer to the revised manuscript, which now includes a detailed comparison, attributing both similarities and differences (Appendix I):
> 1. We theoretically show that Mamba can learn a parameter-efficient solution for MQAR, i.e. a solution which requires feasible model dimensions $ D,N=O(\log{(V_k+V_v)})$. These are **tighter bounds** compared to the bounds proved by Huang et al. for Mamba, which are $ D=O(V_k+\log{V_v})$ and $N=V_k$.
> 2. Our suggested recall circuit is validated by **mechanistic interpretability** experiments, based on trained model weights and activations.
> 3. Our contribution includes the theoretical derivation of **scaling laws** for recall in Mamba, which cannot be derived from Theorem 1 in Huang et al., since it is based on a different circuit.
> 4. In addition to single-layer models, our study generalizes to **multi-layer** Mamba models (Theorem 4).
> 5. We further contribute the study of **multi-head recall circuits for Mamba-2 models** (Theorem 5).
> 6. Our study shows the suggested recall circuit is **minimal**, which is important for understanding the bottlenecks for recall in Mamba.
>
> \
> We turn to address each specific concern directly:
>
> .
> > **W.1.1.** Theorem 1 uses the construction of Lemma 3 in Huang et al.
>
> Lemma 3 in Huang et al. proves that a 1-layer Mamba-$\Delta^\top$ model can solve the Induction Heads task, if model dimensions are $D,N=O(V)$. Theorem 1 in our work leverages a similar construction for a 1-layer Mamba, showing it can solve MQAR, given $D,N=O(V)$. Importantly, however, Theorem 1 is not the key novelty of our work, but rather serves as an intermediate logical step towards Theorem 2, which addresses a realistic scenario where $D,N=O(\log{V})<O(V)$, thus applicable to actual MQAR tasks.
>
> .
> > **W.1.2.** Theorem 2 largely reuses the idea of Theorem 2 in Huang et al. (i.e. using JL lemma to reduce dimensionality).
>
> Theorem 2 in Huang et al. shows that a 1-layer Mamba with dimensions $D=O(V_k+\log{V_v})$ and $N=V_k$ can solve MQAR, by applying JL lemma to the values.
>
>
> Firstly, Theorem 2 in our work improves the **efficiency** of the solution, by showing that a 1-layer Mamba model can solve MQAR with smaller, realistic dimensions $D,N=O(\log{V})$, by using a different construction, which allows us to apply JL lemma *twice*, to both values and keys. (Technically, we show Mamba can compress its inputs twice: first compression via embedding into $Ex_t\in \mathbb{R}^{D}$, used for values; then a second compression via $S_B,S_C$ into $FEx_t \in \mathbb{R}^{N}$, used for keys and queries.) This difference is of major importance in practice: see Figure 3, where we empirically show Mamba can solve an MQAR task of $V_k=\frac{V}{2}=512$ given dimensions $N=32<<V_k$ and $D=64<<V_k$.
>
>
> Secondly, our **mechanistic analysis** (also see Figure 3) highlights the difference between the two constructions. The solution suggested by Huang et al. implies that **each column corresponds to a specific key**: each embedded value $Ev_\tau$ is stored in a **single** distinct key coordinate of the hidden state $h_t$, out of $N=V_k$ coordinates in total. In contrast, our parameter-efficient construction suggests each embedded value is stored in a *distributed* manner, across a linear mixture of *all* $N=O(\log{V_k)}$ key coordinates. The mechanistic experiments we conduct in Section 3.2 support our suggested circuit.
>
>
> Thirdly, the fact Theorem 2 in our work uses JL compression twice allows a derivation of the correct scaling law, which depends on both model dimensions $N$ and $D$ (see Theorem 3). Theorem 2 at Huang et al. uses a different circuit, hence would have resulted in a different scaling law. (See detailed explanation in Appendix I.)

---

> > ### Author Response · Authors · 2025-12-03
> > **Response to Reviewer 1qe9 (Part 2/4)**
> >
> > > **W.1.3.** Theorem 4 in Huang et al. shows that using convolution and gating [...] can solve MQAR, which takes a step towards understanding the role of gating, a direction that the authors remark for future work.
> >
> > Our work focuses on identifying a minimal Mamba circuit that can solve MQAR, allowing us to isolate the key mechanisms contributing to performance.
> >
> >
> > Additionally, our ablation study (Appendix K; Table 10) demonstrates that a Mamba model without gating continues to perform well on MQAR. This suggests that while gating may contribute to recall in some SSM variants, here, its absence does not produce a fundamental degradation, and therefore it should not be considered an essential part of the recall circuit.
> >
> > This insight is further reinforced by the results presented in Figure 6. The conducted experiment compares between our gate-free simplified model and the full gated model, both trained on MQAR. Our findings show the models perform quite similarly on MQAR. This demonstrates that the main bottleneck in Mamba recall capacity is the model dimensions $D$ and $N$, rather than the existence of a gate.
> >
> > .
> > > **W.2.** Applicability to other recall tasks - “MQAR with repeating keys, induction heads where keys and values are from the same vocabulary”.
> >
> > Thank you for this important concern. In the revised manuscript, we have added an Appendix (G) discussing the applicability of our framework for two variants of **MQAR with repeating keys** (where keys repeat either among the context facts or within the queries part of the context). We provide both theoretical insights and experimental validation for each of the variants. \
> > Interestingly, we find that while single-layer Mamba models struggle with these harder tasks, two-layer models can successfully solve them.
> >
> > \
> > **Questions:**
> >
> > > **Q.1.** Equation 2 purpose and clarification about hidden state dimensions.
> >
> > Thank you for pointing this out. Indeed, some details were missing, as are now clarified and corrected in the revised manuscript. (Also see Appendix A.3.)
> >
> > Equation 2 describes the SSM recurrent state-update as appears in the original, full Mamba model (selective SSM, or S6). This equation is applied per-channel, where there are $ D_\mathrm{in}$ channels. However, equation 2 is only a reference point for the original model; along the paper, equation 3 is always used - it simplifies equation 3 to match the simplified (trivially-discretized) model we use.
> >
> > Regarding hidden state dimension in equation 2: \
> > Generally, $h_t \in \mathbb{R}^{N \times D}$. Hence, per each of the $D_\mathrm{in}$ channels, the per-channel state is a vector $h_t^{d} \in \mathbb{R}^N$.
> > The discretized matrix $ \bar{A_t}\, \in \mathbb{R}^{N \times D_\mathrm{in}}$, thus per channel $\bar{A}_t^{d} \in \mathbb{R}^{N}$.
> > Per channel, we have a vector-elementwise product $\bar{A}_t^{d} \odot h_t^{d}$.
> >
> >
> > Regarding the SSM output (second part of equation 2): per channel, $\hat{y}_t^{d} \in \mathbb{R}$ is the scalar result of a vector dot-product: $\hat{y}_t^{d} = h_t^{d} \cdot C_t$ (note that $C_t \in \mathbb{R}^N$ does not have a per-channel discretization). Hence, overall, $\hat{y}_t  \in \mathbb{R}^D$ is the vector result of a matrix-vector multiplication $\hat{y}_t = h_t \cdot C_t$.

---

> > > ### Author Response · Authors · 2025-12-03
> > > **Response to Reviewer 1qe9 (Part 3/4)**
> > >
> > > > **Q.2.** I want to better understand the computation of $H’’$: [...] The authors comment $H’’$ in eqn.13 does not exist as-is [...] But it seems possible to compute eqn.13 from the learned embedding. Can the authors clarify?”
> > >
> > > Thank you for your excellent question, which indicates a deep understanding of our work. It is interesting indeed, and a proper clarification has been added in the revised version (see Appendix C.3).
> > >
> > > The starting point of the paragraph “Hidden state as a hash table” is the “non-rotated” recall circuit, as constructed in Theorem 2, where $P_{in}$, $P_{out}$, and $W_{conv}$ are carefully designed with identity and zero matrices and vectors. In this setup, $S_B$ and $S_C$ must have a structure $S_B=(F\mid 0)$ and $S_C=(0\mid F)\$ in order to solve MQAR. If we fix $P_{in}$, $P_{out}$, and $W_{conv}$ as in the construction, then indeed $F$ can be extracted from learned model weights, allows to compute $\tilde{E}$, then to compute equation 13 and $H''=E^\top H' \tilde{E}$.
> > >
> > > However, in the general case, where $P_{in}$, $P_{out}$, and $W_{conv}$ are not constrained, the operator $F$ does not exist in model weights: $S_B$ and $S_C$ do not have the simple structure described in Theorem 2, since the key and query compression operations $F$ are distributed across a mixture of linear operators $S_B$, $S_C$, $P_{in}$, and $W_{conv}$. Therefore, we must use the net operations $\Pi_{k,\mathrm{in}}$ and $\Pi_{q,\mathrm{in}}$ (see equation 8), which correctly combine all these linear transformations. Decompression via $k_n^{\prime\prime}=\tilde{E}^\top \tilde{E} k_n$ is a special case, which only applies for the “non-rotated” circuit; its general “rotation-invariant” form is $k_n^{\prime\prime} = G_{kq}^\top k_n$.  Similarly, $H_t^{\prime\prime} = E^\top H_t^{\prime} \tilde{E}$ is a special “non-rotated” case; its general invariant form is $H_t^{\prime\prime}=\Pi_{v,\mathrm{out}}\, H_t^{\prime} \Pi_{q,\mathrm{in}}$, which is derived by replacing $E^\top$ and $\tilde{E}$ with their general equivalent operators from equation 8.
> > >
> > >
> > > We understand why these insights are non-trivial and require further explanation. A proper explanation has been added to the revised manuscript.
> > >
> > > .
> > > > **Q.3.** How do the mechanistic and theoretical findings generalize to Mamba2?
> > >
> > > We thank you for highlighting this important concern.
> > > Our revised manuscript applies our results to Mamba-2. Specifically, Mamba-2 leverages a quite different structure of convolutions and projections, and further enables a multi-head SSM usage. We study associative recall circuits for these architectures, as can be found in Sections 4.3 and 5.2.
> > >
> > > Let us briefly summarize our conclusions.
> > > Firstly, we show the core recall mechanism is similar to that of Mamba (see i.e. App. F.2, Lemma 2), consisting of values compression into $\mathbb{R}^D$, keys and queries compression into $ \mathbb{R}^N $, query-key matching via SSM, and value decompression.
> > > Secondly, we study multi-head patterns in Mamba-2. We show both theoretically and empirically that MHA pattern benefits an improved recall capacity, while other patterns (MVA, MQA, MKA) do not improve recall. (See Lemmas 2,3,4,5 in App. F.2.)
> > >
> > > .
> > > > **Q.4.** The proof of Lemma 1: while the proof steps look correct to me, some wording and analysis look confusing. [...]
> > >
> > > Thank you for your helpful question.
> > > In the revised manuscript, we have added a table (see Appendix D, Table 5) to better explain these proof steps. Wording has also been improved, thanks to your comment.
> > >
> > > Our intention by these notations is as follows (the wording is different from the one originally used): \
> > > Assume we have an existing fact $(k_m, v_i)$ and a query $q_t=k_i$.  The total $V_v$ value entries of the output vector $y_t$ are divided into: \
> > > (a) a single “correct” entry $i$, which corresponds to the ground-truth value $v_i$ the model should retrieve, \
> > > (b) other $N_f - 1$ value entries corresponding to all other fact values $v_j \neq v_i$ which appear the context, thus “wrong”, and \
> > > (c) all other value entries, which are “empty” since their associated values don’t appear in any facts in the context.
> > >
> > > Next, considering each *output entry* $ y_t^{p} $ by type (a) (b) and (c), we analyze the contribution caused by different types of *token pairs* in the context, defined as:
> > > (1) ‘correct fact’ token pair - where $x_{\tau-1} = k_m$,
> > > (2) ‘wrong fact’ token pair - where $x_{\tau-1} \neq k_m$, and
> > > (3) ‘non-fact’ token pair - all other context tokens.
> > >
> > > Then, “matching” and “non-matching” is not a definition but a property of a token pair, measuring the match between key and query ($x_{\tau-1}$ and $k_m$) and between entry and value ($v_p$ or $e_p$ and $x_\tau$).
> > >
> > > We hope the improved wording in the revised paper, along with Table 5, are helpful for better readability of the proof.

---

> > > > ### Author Response · Authors · 2025-12-03
> > > > **Response to Reviewer 1qe9 (Part 4/4)**
> > > >
> > > > > **Q.5.** Correctness and typos in Theorem 3 full proof (appendix).
> > > >
> > > > Thank you for the insightful question and your careful attention to the details. Indeed, we found several typos in the proof. All typos have been corrected in the paper revision. We have further detected and corrected additional algebraic errors, which, however, do not influence the derived scaling law in the large $N_f$ regime (Equation 98 in the revised paper).
> > > >
> > > > \
> > > > **Minor:**
> > > >
> > > > > **Q.a.** The notation $N_f$ in proof sketch of Lemma 1 and Theorem 3 means the same as $N_\mathrm{facts}$?
> > > >
> > > > We thank you for pointing this out. Indeed $N_f$ means the same as $N_\mathrm{facts}$. Notations are unified in the revised version.
> > > >
> > > > .
> > > > > **Q.b.** What does the pixel color in Fig.6 mean (accuracy)? Also, the color bar is missing.
> > > >
> > > > We thank you for your question. Indeed, pixel color in Figure 6 means model measured accuracy on MQAR. The corresponding color bar has been included in the revised version.

---

### Official Review · Reviewer_dQhk · 2025-11-09

**Soundness:** 3
**Presentation:** 2
**Contribution:** 2
**Rating:** 4
**Confidence:** 2

**Summary:**

This paper studies the Associative Recall (AR) capabilities of Mamba models (a Selective State Space Model, SSM) from a mechanistic interpretability perspective. The main contributions of this paper includes:

1. Reverse-engineer an underlying structure that enables Mamba to solve MQAR (Multi-Query Associative Recall). The peper first identifies a perfect non-compressive recall circuit with large state and model dimensions (Theorem 1), then relax the requirement for state and model dimensions through claiming a structure that solve MQAR with high probability (Theorem 2). The construction is verified through experiments. The paper further interprets Mamba’s recall as a linear similarity-preserving hash function, linking it to the Johnson–Lindenstrauss (JL) lemma.

2. Develop analytic AR scaling laws characterizing the success rate of a single-layer linear Mamba on solving the MQAR task, as a function of the state and model dimensions (N, D), vocabulary size V and number of facts $N_{\text{facts}}$ (Theorem 3). Further it derives the dimensions (N, D) required for solving an MQAR task with high probability, which are in the logarithm order of V. These theoretical results are validated empirically through experiments demonstrating matching scaling behavior.

**Strengths:**

This paper theoretically elucidates how the Mamba model performs associative recall, combining mechanism analysis, probabilistic modeling, and empirical verification.

$\textbf{Originality}$: The work introduces a novel mechanistic reconstruction of Mamba’s recall circuitry and interprets recall as a linear similarity-preserving hash, grounded in the Johnson–Lindenstrauss lemma. This creates an elegant conceptual link between state-space modeling and random projection theory.

$\textbf{Quality}$: The theoretical derivations are supported by empirical results. The accompanying simulations closely align with the analytical predictions, reinforcing the validity of the proposed framework.

$\textbf{Significance}$: This study offers the first theoretical scaling law for associative recall in Mamba models, offering quantitative insight into memory capacity and architectural trade-offs. These contributions advance theoretical understandings on the capability of selective SSMs.

**Weaknesses:**

As SSMs are not my primary area of expertise, my comments focus on readability and clarity. My main concern is that the paper is difficult to follow, even for readers with some familiarity with state-space models. To make the work more accessible to a broader audience, I encourage the authors to provide sufficient details when introducing new concepts or notations. Specifically, the following points could be clarified:

1. It would be beneficial to elaborate the example of MQAR (lines 094-100). What are keys, values, and queries in this context exactly?

2. What are the dimensions of each Mamba components ($\bar{A}_t,\bar{B}_t,C_t, \hat{x}_t, h_t, \hat{y}_t)$? Generally, when a matrix is first introduced, please specify its dimension.

3. In Figure 1, what is $\hat{0}$? Also what is the circle product notation? If it is element-wise product, please be consistent with the $\odot$ notation. Please include sufficient details in the caption to explain how the circuit works.

4. In Theorem 1 and Theorem 2, please be specific what does $\textsf{expand}$ refer to.

5. If there are any typos in equations (5) and (7), please correct it.

6. In the definition of $\hat{E}_{in}$ (line 231), what are $W_0, W_1$?

7. In Figure 2, please explain what the axis labels t and $\tau$ refer to.

8. There appear to be inconsistencies in notation, e.g., $G_{kq}$ and $G_{qk}$ (in page 5), $N_{facts}$ and $N_f$ (in page 7).

9. In Figure 6,  it would be helpful to add a colorbar to indicate the numerical values represented by each color.

Overall, defining all variables clearly, avoiding notation inconsistency, and providing brief explanations when introducing new symbols or concepts would greatly enhance the paper’s readability and accessibility.

**Questions:**

1. Please refer to the weakness section.

2. In equation (2) (line 133), I assume $h_t, \hat{x}_t$ are vectors and $\bar{A_t}, \bar{B}_t$ are discretized matrices. How did element-wise multiplication applied here?

3. In the scaling law experiments, the authors report the maximum accuracy across trials. Could the authors clarify this choice? Intuitively, averaging results over multiple runs (or mean ± standard deviation) would provide a more robust and statistically meaningful comparison.

4. In this paper, all theoretical results are based on single-layer linear model. It would be valuable to discuss how these results provide insight into multi-layer SSM architectures, or to clarify what key challenges prevent extending the analysis to deeper Mamba models.

---

> ### Author Response · Authors · 2025-12-03
> **Response to Reviewer dQhk (Part 1/2)**
>
> Thank you for the comprehensive review and the constructive feedback.
>
> .
> > **W.1.** Clarity of MQAR example: “What are keys, values, and queries in this context exactly?”
>
> A proper clarification (with color highlighting) is added in the revised manuscript (see L98-L100).
>
> .
> > **W.2.** Dimensions of Mamba components: “when a matrix is first introduced, please specify its dimension.”
>
> Thank you for this point. We have added an Appendix with dimension details to the revised manuscript (see Appendix A.2), as well as dimension specification throughout the paper.
>
> .
> > **W.3.** Clarity of notations and symbols in Figure 1.
>
> We appreciate your detailed comment. We have updated Figure 1 correspondingly in the revised manuscript, to better match the circuit in Theorem 2. In addition, we have included a more detailed caption, to ease understanding.
> In the original figure, we intended $\hat{0}$ to represent the zero vector in $\mathbb{R^V}$, and the “circle” notation to represent concatenation. Indeed, we see why these symbols might have been confusing. We hope the updated figure and the extended caption better explain the circuit and fix the issue of consistency with the paper notations.
>
> .
> > **W.4.** In Theorem 1 and Theorem 2, please be specific what does expand refer to.
>
> Thank you for noticing this. In the revised manuscript, *expand* is presented earlier. *expand* is the SSM projection expansion factor, such that $D_{\text{in}}=\mathrm{expand}\times D$. This clarification has been added to the revised paper.
>
> .
> > **W.5.** Fixing typos in equations (5) and (7).
>
> We apologize for the typos. Indeed, in equation (7), $P_\mathrm{out}$ and $E_\mathrm{out}$ are accidentally switched, and the commas ‘,’ are typos. Equation (5) contains no typos. All typos are fixed in the revised manuscript.
>
> .
> > **W.6.** Definition of $W_0$ and $W_1$.
>
> Thank you for pointing this out; we have clarified the notation in the revised paper. By these notations we mean $W_\mathrm{conv}=\bigl(W_0\mid W_1)$, i.e. $W_0$ and $W_1$ are column vectors in $ \mathbb{R}^{D_{\text{in}}}$.
>
> .
> > **W.7.** In Figure 2, please explain what the axis labels t and $\tau$ refer to.
>
> We are sorry for the missing explanation; it has been added in the revised manuscript (see Appendix C.1). \
> Our intention is as follows. $G_{kq}$ is a bilinear operator operates on token pairs $\xi_t = (x_{t-1}, x_{t})$ and $\xi_{\tau} = (x_{\tau-1}, x_{\tau})$. Therefore, it has 4 blocks, corresponding to the 4 combinations of $x_t, x_{t-1}$ parts and $x_{\tau}, x_{\tau-1}$ parts. Similarly, $G_{vv}$ is thought as a matching between token pairs $\xi_{\tau}$ and output tokens $y_t$. Hence, it has 2 blocks, corresponding to the $x_{\tau}, x_{\tau-1}$ parts and the $x_t$ part. In the revised paper, the figure now refers to a detailed explanation.
>
> .
> > **W.8.** Notation inconsistencies.
>
> Thank you for your careful reading and attention to details. These typos are fixed in the revised paper. Indeed, $N_f$ and $N_\mathrm{facts}$ are the same, and $G_{qk}$ is a typo for $G_{kq}$.
>
> .
> > **W.9.** Adding a colorbar to Figure 6.
>
> We appreciate your helpful comment. A colorbar to indicate numerical values for the color axis (accuracy) has been added in the revised manuscript.
>
> \
> **Questions:**
>
> > **Q.2.** Dimensions clarification for equation (2)
>
> We appreciate your on-point question, as indeed, further clarification is required. We have included a detailed explanation in the revised manuscript (see Appendix A.3). \
> In equation (2), the proper dimensions are as follows.
>
>
> Generally, $ h_t \in \mathbb{R}^{N \times D_\mathrm{in}} $, and $\hat{x_t} \in \mathbb{R}^{D_{\text{in}}} $. Hence, per each of the $ D_\mathrm{in}$ channels, the per-channel state is a vector $ h_t^{d} \in \mathbb{R}^N$ and the input is a scalar $ \hat{x_t}^{d} \in \mathbb{R}$.  \
> The discretized matrices are $ \bar{A_{t}}, \bar{B_t} \in \mathbb{R}^{N \times D_\mathrm{in}} $ , thus per channel $ \bar{A}_t^{d}, \bar{B}_t^{d} \in \mathbb{R}^{N} $ .  \
> Per channel, we have a vector-elementwise product $ \bar{A}_t^{d} \odot h_t^{d}$ and a scalar-vector product $ \hat{x}_t^{d} \bar{B}_t^{d}$. \
> Therefore, the first outer-product symbol is correct, but the second one is indeed misleading. The revised paper fixes this problem, using proper, careful notations.

---

> > ### Author Response · Authors · 2025-12-03
> > **Response to Reviewer dQhk (Part 2/2)**
> >
> > > **Q.3.** Clarification of the choice to report the maximum accuracy across trials.
> >
> > Thank you for your important question. In this work we investigate Mamba’s maximal recall **capacity**, therefore retaining the maximal result achieved by each configuration is the most relevant quantity for our analysis. In other words,  we are interested in the **existence** of a recall solution, not the **probability** of achieving it by optimization. Therefore, it makes more sense to take into account only the trials where optimization succeeds. \
> > A similar explanation has been added to the revised paper (see Appendix J.5).
> >
> > .
> > > **Q.4.** It would be valuable to discuss how these results provide insight into multi-layer SSM architectures.
> >
> > We thank you for your insightful comment. In the revised paper, we have added a further theoretical analysis of multi-layer recall in Mamba (Theorem 4), supported by additional experiments (Section 5.2). \
> > Briefly summarizing our results, we find that a Mamba model with $\Lambda$ layers can perfectly perform associative recall relying on smaller dimensions $D=O(\log V)$, $N=O(\frac{\log V}{\Lambda})$.

---

### Author Response · Authors · 2025-12-03
**Summary for AC**

Dear AC,

We thank you for taking the time to review our work.

First, we are encouraged that the reviewers found that our paper (1) addresses an important problem [jebP], (2) introduces a comprehensive mechanistic interpretability analysis [1qe9, dQhk] (3) provides the first theoretical scaling laws for associative recall in Mamba models [1qe9, dQhk], and (4) backs all theoretical claims with empirical experiments which reinforce their validity [1qe9, jebP, dQhk].

The remaining feedback includes three main concerns which emerged across reviews: (1) relation to prior work (Huang et al., 2025), (2) generality of our results, and (3) clarity/presentation issues. \
We highlight that, to the best of our understanding, **we have addressed all three concerns thoroughly** in the new revision (changes are in blue). The details of our main claims are provided below. The full details can be found in the response to each reviewer.

.
> **1. Relation to prior work**

We provide a detailed comparison (Appendix I) clarifying both similarities and key differences. Our contributions go substantially beyond Huang et al. (2025):

* We establish **tighter, practically realistic bounds** for Mamba’s recall capacity, showing MQAR is solvable with model dimensions $N,D=O(\log{⁡⁡(V_k + V_v)})$ rather than $D=O(V_k + \log{V_v})$, $N=O(V_k)$
* These tighter constructions enable the first **scaling laws** for associative recall in Mamba (Theorem 3), which match empirical behavior and could not be derived from Huang et al.’s circuit.
* We provide the first **mechanistic interpretability analysis** confirming that trained Mamba models implement our parameter-efficient recall circuit (up to invariances).
* We further extend our results to **multi-layer Mamba (Theorem 4) and Mamba-2 multi-head SSMs (Theorem 5).**

.
> **2. Generality and empirical validity**

Reviewers asked whether our findings extend beyond a single-layer linear model and whether they apply to other tasks and architectures. The revision now includes:
* **Multi-layer** recall analysis (App. E.1, Thm. 4) and experiments (Sec. 5.2).
* **Mamba-2 multi-head** circuit analysis (App. E.2, Thm. 5) and experiments (Sec. 5.2).
* Applicability to broader recall tasks: MQAR with **repeating keys** (App. G)
* Clarifications on how our mechanistic operators generalize (App. C)

.
> **3. Presentation improvements.**

The reviewers noted several clarity issues. We have carefully revised the manuscript to enhance its readability:
* All symbols, dimensions, and operations are now explicitly defined; figures and captions have been reworked; all inconsistent notations and typos (including Eq. 7, Fig. 1, Fig. 6) are corrected.
* A new Appendix (A) consolidates dimensions and SSM details; Appendix C clarifies operators and invariances; Appendix D provides clearer proof structure (i.e. a new Table 5, precise definitions).
* Additionally, we clarified several technical points raised in the questions, including SSM dimensions, equation consistency, and the rationale for reporting max-accuracy.

\
Overall, we believe the revised version significantly strengthens both the technical contributions and clarity, and we hope the AC finds the improvements satisfactory.

---

### Meta-Review · Area_Chair_RJeP · 2026-01-06

**Summary:**

This paper studies the theoretical power of Mamba for MQAR. There were two major concerns raised by reviewers. One is the relationship with the prior work (Huang et al., 2025) and the other is about the presentation of this submission. While the authors clarified the difference between their result and (Huang et al., 2025), I found that the revised draft is still hard to follow and not ready for publication. I believe that this draft requires careful revision, including precise definitions of the target model (e.g., SSR, Mamba) and theorem statements.

**Reviewer Concerns:**

All reviewers raised concerns/questions about the presentation (e.g., hard to follow) and typos in this submission. While the authors tried to address this issue in their revision, I found that the revised manuscript is still hard to follow and not mathematically rigorous. For example, in the definition of Mamba, the authors did not precisely define what SelectiveSSR, Linear, Conv1D are, and some notations are written in bold/non-bold simultaneously (e.g., Eqs. (1) and (2)).

Some theorems are not precisely stated. For example, in Theorem 2, what is the precise definition of the MQAR task? what is the precise success probability? and what is hidden in the big-O notation? Theorem 2 refers to Theorem 3 for the quantitative analysis; however, its statement does not seem to be precise as well.

**Reviewer Scores:**

I think reviewers would not change their scores.

---

### Decision · Program_Chairs · 2026-01-26

Reject